# Improving Linear System Solvers for Hyperparameter Optimisation in Iterative Gaussian Processes

**Jihao Andreas Lin**[1,2]     **Shreyas Padhy**[1]     **Bruno Mlodozeniec**[1,2]
**Javier Antorán**[1,3]     **José Miguel Hernández-Lobato**[1,3]
[1]University of Cambridge    [2]MPI for Intelligent Systems, Tübingen    [3]Ångström AI

## Abstract

Scaling hyperparameter optimisation to very large datasets remains an open problem in the Gaussian process community. This paper focuses on iterative methods, which use linear system solvers, like conjugate gradients, alternating projections or stochastic gradient descent, to construct an estimate of the marginal likelihood gradient. We discuss three key improvements which are applicable across solvers: (i) a pathwise gradient estimator, which reduces the required number of solver iterations and amortises the computational cost of making predictions, (ii) warm starting linear system solvers with the solution from the previous step, which leads to faster solver convergence at the cost of negligible bias, (iii) early stopping linear system solvers after a limited computational budget, which synergises with warm starting, allowing solver progress to accumulate over multiple marginal likelihood steps. These techniques provide speed-ups of up to $72\times$ when solving to tolerance, and decrease the average residual norm by up to $7\times$ when stopping early.

## 1   Introduction

Gaussian processes [22] (GPs) are a versatile class of probabilistic machine learning models which are used widely for Bayesian optimisation of black-box functions [24], climate and earth sciences [10, 26], and data-efficient learning in robotics and control [6]. However, their effectiveness depends on good estimates of hyperparameters, such as kernel length scales and observation noise. These quantities are typically learned by maximising the marginal likelihood, which balances model complexity with training data fit. In general, the marginal likelihood is a non-convex function of the hyperparameters and evaluating its gradient requires inverting the kernel matrix. Using direct methods, this requires compute and memory resources which are respectively cubic and quadratic in the number of training examples. This is intractable when dealing with large datasets of modern interest.

Methods to improve the scalability of Gaussian processes can roughly be grouped into two categories. Sparse methods [20, 27, 12] approximate the kernel matrix with a low-rank surrogate, which is cheaper to invert. This reduced flexibility may result in failure to properly fit increasingly large or sufficiently complex data [15]. On the other hand, iterative methods [11] express GP computations in terms of systems of linear equations. The solution to these linear systems is approximated up to a specified numerical precision with linear system solvers, such as conjugate gradients (CG) [11, 9, 30], alternating projections (AP) [23, 28, 33], or stochastic gradient descent (SGD) [15, 16]. These methods allow for a trade-off between compute time and accuracy. However, convergence can be slow in the large data regime, where system conditioning is often poor.

In this paper, we focus on iterative GPs and identify techniques, which were important to the success of previously proposed methods, but did not receive special attention in the literature. Many of these amount to amortisations which leverage previous computations to accelerate subsequent ones. We analyse and adapt these techniques, and show that they can be applied to accelerate different linear solvers, obtaining speed-ups of up to $72\times$ without sacrificing predictive performance (see Figure 1).

38th Conference on Neural Information Processing Systems (NeurIPS 2024).

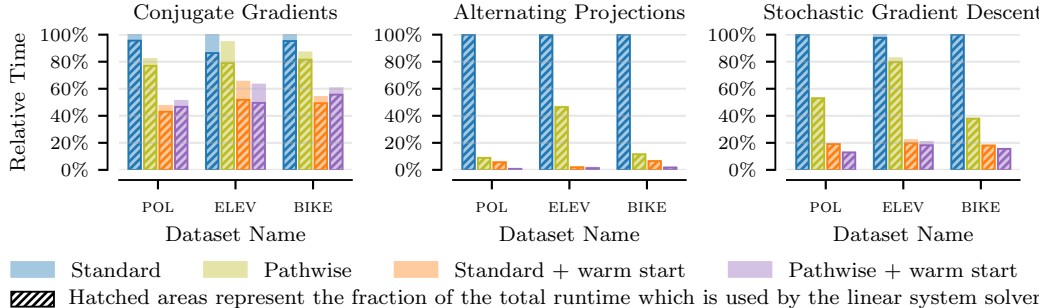

Figure 1: Comparison of relative runtimes for different methods, linear system solvers, and datasets. The linear system solver (hatched areas) dominates the total training time (coloured patches). The pathwise gradient estimator requires less time than the standard estimator. Initialising at the previous solution (warm start) further reduces the runtime of the linear system solver for both estimators.

In the following, we summarise our contributions:

- We introduce a **pathwise estimator** of the marginal likelihood gradient and show that, under real-world conditions, the solutions to the linear systems required by this estimator are closer to the origin than those of the standard estimator, allowing our solvers to converge faster. Additionally, these solutions transform into samples from the GP posterior without further matrix inversions, amortising the computational costs of predictive posterior inference.

- We propose to **warm start** linear system solvers throughout marginal likelihood optimisation by reusing linear system solutions to initialise the solver in the subsequent step. This results in faster convergence. Although this technically introduces bias into the optimisation, we show that, theoretically and empirically, the optimisation quality does not suffer.

- We investigate the behaviour of linear system solvers on a **limited compute budget**, such that reaching the specified tolerance is not guaranteed. Here, warm starting allows the linear system solver to accumulate solver progress across marginal likelihood steps, progressively improving the solution quality of the linear system solver despite early stopping.

- We demonstrate empirically that the methods above either reduce the required number of iterations until convergence without sacrificing performance or improve the performance if a limited compute budget hinders convergence. Across different UCI regression datasets and linear system solvers, we observe average **speed-ups of up to 72×** when solving until the tolerance is reached, and **increased performance** when the compute budget is limited.

Source code available at: `https://github.com/jandylin/iterative-gaussian-processes`

## 2   Gaussian Process Regression and Marginal Likelihood Optimisation

Formally, a GP is a stochastic process $f : \mathcal{X} \to \mathbb{R}$, such that, for any finite subset $\{x_i\}_{i=1}^n \subset \mathcal{X}$, the set of random variables $\{f(x_i)\}_{i=1}^n$ is jointly Gaussian. In particular, $f$ is uniquely identified by a mean function $\mu(\cdot) = \mathbb{E}[f(\cdot)]$ and a positive-definite kernel function $k(\cdot, \cdot'; \boldsymbol{\vartheta}) = \mathrm{Cov}(f(\cdot), f(\cdot'))$ with kernel hyperparameters $\boldsymbol{\vartheta}$. We use a Matérn-$3/2$ kernel with length scales per dimension and a scalar signal scale and write $f \sim \mathrm{GP}(\mu, k)$ to express that $f$ is a GP with mean $\mu$ and kernel $k$.

For GP regression, let the training data consist of $n$ inputs $\boldsymbol{x} \subset \mathcal{X}$ and targets $\boldsymbol{y} \in \mathbb{R}^n$. We consider the Bayesian model $y_i = f(x_i) + \epsilon_i$, where each $\epsilon_i \sim \mathcal{N}(0, \sigma^2)$ i.i.d. and $f \sim \mathrm{GP}(\mu, k)$. We assume $\mu = 0$ without loss of generality. The posterior of this model is $f | \boldsymbol{y} \sim \mathrm{GP}(\mu_{f|\boldsymbol{y}}, k_{f|\boldsymbol{y}})$, with

$$\mu_{f|\boldsymbol{y}}(\cdot) = k(\cdot, \boldsymbol{x}; \boldsymbol{\vartheta})(k(\boldsymbol{x}, \boldsymbol{x}; \boldsymbol{\vartheta}) + \sigma^2 \mathbf{I})^{-1} \boldsymbol{y}, \tag{1}$$

$$k_{f|\boldsymbol{y}}(\cdot, \cdot') = k(\cdot, \cdot'; \boldsymbol{\vartheta}) - k(\cdot, \boldsymbol{x}; \boldsymbol{\vartheta})(k(\boldsymbol{x}, \boldsymbol{x}; \boldsymbol{\vartheta}) + \sigma^2 \mathbf{I})^{-1} k(\boldsymbol{x}, \cdot'; \boldsymbol{\vartheta}), \tag{2}$$

where $k(\cdot, \boldsymbol{x}; \boldsymbol{\vartheta})$, $k(\boldsymbol{x}, \cdot; \boldsymbol{\vartheta})$ and $k(\boldsymbol{x}, \boldsymbol{x}; \boldsymbol{\vartheta})$ refer to pairwise evaluations, resulting in a $1 \times n$ row vector, a $n \times 1$ column vector and a $n \times n$ matrix respectively.

**Pathwise Conditioning**   Wilson et al. [31, 32] express a GP posterior sample as a random function

$$(f|\boldsymbol{y})(\cdot) = f(\cdot) + k(\cdot, \boldsymbol{x}; \boldsymbol{\vartheta})(k(\boldsymbol{x}, \boldsymbol{x}; \boldsymbol{\vartheta}) + \sigma^2 \mathbf{I})^{-1}(\boldsymbol{y} - (f(\boldsymbol{x}) + \boldsymbol{\epsilon})), \tag{3}$$

where $\epsilon \sim \mathcal{N}(\mathbf{0}, \sigma^2 \mathbf{I})$ is a random vector, $f \sim \mathrm{GP}(0, k)$ is a zero-mean prior function sample, and $f(\boldsymbol{x})$ is its evaluation at the training data. Following previous work [31, 32, 15, 16], we efficiently approximate the prior function sample using random features [21, 25] (see Appendix B for details). Using pathwise conditioning, a single linear solve suffices to evaluate a posterior function sample at arbitrary locations without further linear solves. In Section 3, we amortise the cost of this single linear solve during marginal likelihood optimisation to obtain posterior samples efficiently.

**The Marginal Likelihood and Its Gradient** With hyperparameters $\boldsymbol{\theta} = \{\boldsymbol{\vartheta}, \sigma\}$ and regularised kernel matrix $\mathbf{H}_{\boldsymbol{\theta}} = k(\boldsymbol{x}, \boldsymbol{x}; \boldsymbol{\vartheta}) + \sigma^2 \mathbf{I} \in \mathbb{R}^{n \times n}$, the marginal likelihood $\mathcal{L}$ as a function of $\boldsymbol{\theta}$ and its gradient $\nabla_{\theta_k} \mathcal{L}$ with respect to $\theta_k$ can be expressed as

$$\mathcal{L}(\boldsymbol{\theta}) = -\frac{1}{2} \boldsymbol{y}^{\mathsf{T}} \mathbf{H}_{\boldsymbol{\theta}}^{-1} \boldsymbol{y} - \frac{1}{2} \log \det \mathbf{H}_{\boldsymbol{\theta}} - \frac{n}{2} \log 2\pi, \tag{4}$$

$$\nabla_{\theta_k} \mathcal{L}(\boldsymbol{\theta}) = \frac{1}{2} (\mathbf{H}_{\boldsymbol{\theta}}^{-1} \boldsymbol{y})^{\mathsf{T}} \frac{\partial \mathbf{H}_{\boldsymbol{\theta}}}{\partial \theta_k} \mathbf{H}_{\boldsymbol{\theta}}^{-1} \boldsymbol{y} - \frac{1}{2} \mathrm{tr} \left( \mathbf{H}_{\boldsymbol{\theta}}^{-1} \frac{\partial \mathbf{H}_{\boldsymbol{\theta}}}{\partial \theta_k} \right), \tag{5}$$

where the partial derivative of $\mathbf{H}_{\boldsymbol{\theta}}$ with respect to $\theta_k$ is a $n \times n$ matrix. We assume $n$ is too large to compute the inverse or log-determinant of $\mathbf{H}_{\boldsymbol{\theta}}$ and iterative methods are used instead.

## 2.1 Hierarchical View of Marginal Likelihood Optimisation for Iterative Gaussian Processes

Marginal likelihood optimisation for iterative GPs consists of bi-level optimisation, where the outer loop maximises the marginal likelihood (4) using stochastic estimates of its gradient (5). Computing these gradient estimates requires the solution to systems of linear equations. These solutions are obtained using an iterative solver in the inner loop. Figure 2 illustrates this three-level hierarchy.

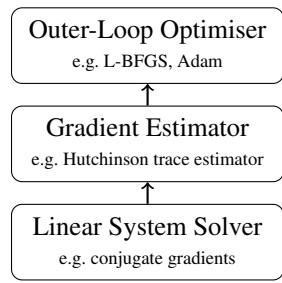

**Outer-Loop Optimiser** The outer-loop optimiser maximises the marginal likelihood $\mathcal{L}$ (4) using its gradient (5). Common choices are L-BFGS [3], when exact gradients are available, and Adam [14] in the large-data setting, when stochastic approximation is required. We consider the case where gradients are stochastic and use Adam.

Figure 2: Marginal likelihood optimisation for iterative GPs.

**Gradient Estimator** The gradient (5) involves two computationally expensive components: linear solves against the targets $\mathbf{H}_{\boldsymbol{\theta}}^{-1} \boldsymbol{y}$ and the trace term $\mathrm{tr} \left( \mathbf{H}_{\boldsymbol{\theta}}^{-1} \partial \mathbf{H}_{\boldsymbol{\theta}} / \partial \theta_k \right)$. An unbiased estimate of the latter can be obtained using $s$ probe vectors and Hutchinson's trace estimator [13],

$$\mathrm{tr} \left( \mathbf{H}_{\boldsymbol{\theta}}^{-1} \frac{\partial \mathbf{H}_{\boldsymbol{\theta}}}{\partial \theta_k} \right) = \mathbb{E}_{\boldsymbol{z}} \left[ \boldsymbol{z}^{\mathsf{T}} \mathbf{H}_{\boldsymbol{\theta}}^{-1} \frac{\partial \mathbf{H}_{\boldsymbol{\theta}}}{\partial \theta_k} \boldsymbol{z} \right] \approx \frac{1}{s} \sum_{j=1}^{s} \boldsymbol{z}_j^{\mathsf{T}} \mathbf{H}_{\boldsymbol{\theta}}^{-1} \frac{\partial \mathbf{H}_{\boldsymbol{\theta}}}{\partial \theta_k} \boldsymbol{z}_j, \tag{6}$$

where the probe vectors $\boldsymbol{z}_j \in \mathbb{R}^n$ satisfy $\forall j : \mathbb{E}[\boldsymbol{z}_j \boldsymbol{z}_j^{\mathsf{T}}] = \mathbf{I}$, and $\boldsymbol{z}_j^{\mathsf{T}} \mathbf{H}_{\boldsymbol{\theta}}^{-1}$ is obtained using a linear solve. We refer to this as the *standard estimator* and set $s = 64$, unless otherwise specified.

**Linear System Solver** Substituting the trace estimator (6) back into the gradient (5), we obtain an unbiased gradient estimate in terms of the solution to a batch of systems of linear equations,

$$\mathbf{H}_{\boldsymbol{\theta}} \left[ \boldsymbol{v_y}, \boldsymbol{v}_1, \ldots, \boldsymbol{v}_s \right] = \left[ \boldsymbol{y}, \boldsymbol{z}_1, \ldots, \boldsymbol{z}_s \right], \tag{7}$$

which share the same coefficient matrix $\mathbf{H}_{\boldsymbol{\theta}}$. Since $\mathbf{H}_{\boldsymbol{\theta}}$ is positive-definite, the solution $\boldsymbol{v} = \mathbf{H}_{\boldsymbol{\theta}}^{-1} \boldsymbol{b}$ to the system $\mathbf{H}_{\boldsymbol{\theta}} \boldsymbol{v} = \boldsymbol{b}$ can be obtained by finding the unique minimiser of the quadratic objective

$$\boldsymbol{v} = \arg\min_{\boldsymbol{u}} \ \frac{1}{2} \boldsymbol{u}^{\mathsf{T}} \mathbf{H}_{\boldsymbol{\theta}} \boldsymbol{u} - \boldsymbol{u}^{\mathsf{T}} \boldsymbol{b}, \tag{8}$$

facilitating the use of iterative solvers. Most popular in the GP literature are conjugate gradients (CG) [9, 30–32], alternating projections (AP) [23, 28, 33] and stochastic gradient descent (SGD) [15, 16]. We consider these in our study and provide detailed descriptions of them in Appendix B. Solvers are often run until the relative residual norm $\|\boldsymbol{b} - \mathbf{H}_{\boldsymbol{\theta}} \boldsymbol{u}\| / \|\boldsymbol{b}\|$ reaches a certain tolerance $\tau$ [30, 18, 33]. We set $\tau = 0.01$, following Maddox et al. [18]. The linear system solver in the inner loop dominates the computational costs of marginal likelihood optimisation for iterative GPs, as shown in Figure 1. Therefore, improving linear system solvers is the main focus of our work.

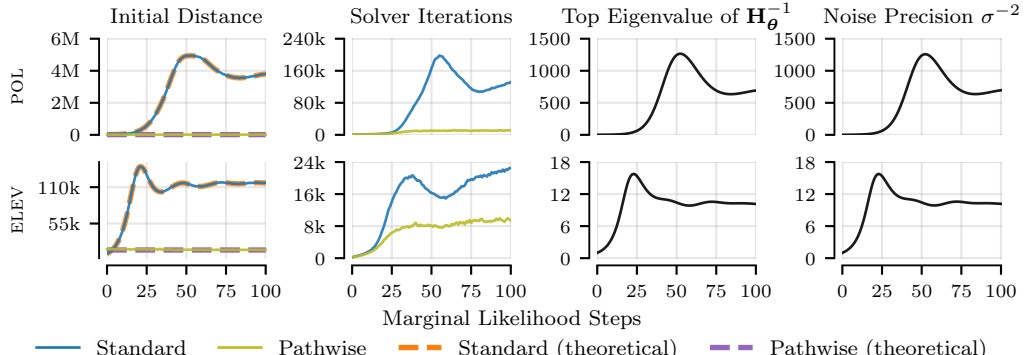

Figure 3: On the POL and ELEVATORS datasets, the pathwise estimator results in a lower RKHS distance (12) between solver initialisation and solution, as predicted by theory (14,15) (left). This results in fewer AP iterations until reaching the tolerance (left middle). When using the standard estimator, the initial distance follows the top eigenvalue of $\mathbf{H}_{\boldsymbol{\theta}}^{-1}$ (right middle), which is strongly related to the noise precision (right). The latter tends to increase during marginal likelihood optimisation when fitting the data. The effects are greater on POL due to the higher noise precision.

## 3 Pathwise Estimation of Marginal Likelihood Gradients

We introduce the *pathwise estimator*, an alternative to the standard estimator (6) which reduces the required number of linear system solver iterations until convergence (see Figure 3). Additionally, the estimator simultaneously provides us with posterior function samples via pathwise conditioning, hence the name *pathwise* estimator. This facilitates predictions without further linear solves.

We modify the standard estimator (6) to absorb $\mathbf{H}_{\boldsymbol{\theta}}^{-1}$ into the distribution of the probe vectors [2],

$$\mathrm{tr}\left(\mathbf{H}_{\boldsymbol{\theta}}^{-1}\frac{\partial\mathbf{H}_{\boldsymbol{\theta}}}{\partial\theta_k}\right) = \mathrm{tr}\left(\mathbf{H}_{\boldsymbol{\theta}}^{-\frac{1}{2}}\frac{\partial\mathbf{H}_{\boldsymbol{\theta}}}{\partial\theta_k}\mathbf{H}_{\boldsymbol{\theta}}^{-\frac{1}{2}}\right) = \mathbb{E}_{\hat{\boldsymbol{z}}}\left[\hat{\boldsymbol{z}}^{\mathsf{T}}\frac{\partial\mathbf{H}_{\boldsymbol{\theta}}}{\partial\theta_k}\hat{\boldsymbol{z}}\right] \approx \frac{1}{s}\sum_{j=1}^{s}\hat{\boldsymbol{z}}_j^{\mathsf{T}}\frac{\partial\mathbf{H}_{\boldsymbol{\theta}}}{\partial\theta_k}\hat{\boldsymbol{z}}_j, \quad (9)$$

where $\forall j : \mathbb{E}[\hat{\boldsymbol{z}}_j\hat{\boldsymbol{z}}_j^{\mathsf{T}}] = \mathbf{H}_{\boldsymbol{\theta}}^{-1}$. Probe vectors $\hat{\boldsymbol{z}}$ with the desired second moment can be obtained as

$$\begin{aligned}f(\boldsymbol{x}) &\sim \mathcal{N}(\mathbf{0}, k(\boldsymbol{x},\boldsymbol{x};\boldsymbol{\vartheta})) \\ \boldsymbol{\epsilon} &\sim \mathcal{N}(\mathbf{0}, \sigma^2\mathbf{I})\end{aligned} \implies \boldsymbol{\xi} \sim \mathcal{N}(\mathbf{0}, \mathbf{H}_{\boldsymbol{\theta}}) \implies \hat{\boldsymbol{z}} = \mathbf{H}_{\boldsymbol{\theta}}^{-1}\boldsymbol{\xi} \sim \mathcal{N}(\mathbf{0}, \mathbf{H}_{\boldsymbol{\theta}}^{-1}), \quad (10)$$

where $\boldsymbol{\xi} = f(\boldsymbol{x}) + \boldsymbol{\epsilon}$. Akin to the standard estimator in Section 2.1, we obtain $\boldsymbol{v_y}$ and $\hat{\boldsymbol{z}}_j$ by solving

$$\mathbf{H}_{\boldsymbol{\theta}}\left[\boldsymbol{v_y}, \hat{\boldsymbol{z}}_1, \ldots, \hat{\boldsymbol{z}}_s\right] = \left[\boldsymbol{y}, \boldsymbol{\xi}_1, \ldots, \boldsymbol{\xi}_s\right]. \quad (11)$$

**Initial Distance to the Linear System Solution**   Under realistic conditions, the pathwise estimator moves the solution of the linear system closer to the origin. To show this, we consider the generic linear system $\mathbf{H}_{\boldsymbol{\theta}}\boldsymbol{u} = \boldsymbol{b}$ and measure the RKHS distance between the initialisation $\boldsymbol{u}_{\mathrm{init}}$ and the solution $\boldsymbol{u} = \mathbf{H}_{\boldsymbol{\theta}}^{-1}\boldsymbol{b}$ as $\|\boldsymbol{u}_{\mathrm{init}} - \boldsymbol{u}\|_{\mathbf{H}_{\boldsymbol{\theta}}}$. With $\boldsymbol{u}_{\mathrm{init}} = \mathbf{0}$, which is standard [9, 30, 1, 15, 33, 16],

$$\|\boldsymbol{u}_{\mathrm{init}} - \boldsymbol{u}\|_{\mathbf{H}_{\boldsymbol{\theta}}}^2 = \|\boldsymbol{u}\|_{\mathbf{H}_{\boldsymbol{\theta}}}^2 = \boldsymbol{u}^{\mathsf{T}}\mathbf{H}_{\boldsymbol{\theta}}\boldsymbol{u} = \boldsymbol{b}^{\mathsf{T}}\mathbf{H}_{\boldsymbol{\theta}}^{-1}\mathbf{H}_{\boldsymbol{\theta}}\mathbf{H}_{\boldsymbol{\theta}}^{-1}\boldsymbol{b} = \boldsymbol{b}^{\mathsf{T}}\mathbf{H}_{\boldsymbol{\theta}}^{-1}\boldsymbol{b}. \quad (12)$$

Since $\boldsymbol{b}$ is a random vector ($\boldsymbol{z}$ in (7) and $\boldsymbol{\xi}$ in (11)), we analyse the expected squared distance

$$\mathbb{E}\left[\boldsymbol{b}^{\mathsf{T}}\mathbf{H}_{\boldsymbol{\theta}}^{-1}\boldsymbol{b}\right] = \mathbb{E}\left[\mathrm{tr}\left(\boldsymbol{b}^{\mathsf{T}}\mathbf{H}_{\boldsymbol{\theta}}^{-1}\boldsymbol{b}\right)\right] = \mathbb{E}\left[\mathrm{tr}\left(\boldsymbol{b}\boldsymbol{b}^{\mathsf{T}}\mathbf{H}_{\boldsymbol{\theta}}^{-1}\right)\right] = \mathrm{tr}\left(\mathbb{E}\left[\boldsymbol{b}\boldsymbol{b}^{\mathsf{T}}\right]\mathbf{H}_{\boldsymbol{\theta}}^{-1}\right). \quad (13)$$

For the standard estimator (6), we substitute $\boldsymbol{b} := \boldsymbol{z}$ with $\mathbb{E}\left[\boldsymbol{z}\boldsymbol{z}^{\mathsf{T}}\right] = \mathbf{I}$, yielding

$$\mathbb{E}\left[\|\boldsymbol{u}_{\mathrm{init}} - \boldsymbol{u}\|_{\mathbf{H}_{\boldsymbol{\theta}}}^2\right] = \mathrm{tr}\left(\mathbb{E}\left[\boldsymbol{z}\boldsymbol{z}^{\mathsf{T}}\right]\mathbf{H}_{\boldsymbol{\theta}}^{-1}\right) = \mathrm{tr}\left(\mathbf{I}\,\mathbf{H}_{\boldsymbol{\theta}}^{-1}\right) = \mathrm{tr}\left(\mathbf{H}_{\boldsymbol{\theta}}^{-1}\right). \quad (14)$$

For the pathwise estimator (9), we substitute $\boldsymbol{b} := \boldsymbol{\xi}$ with $\mathbb{E}\left[\boldsymbol{\xi}\boldsymbol{\xi}^{\mathsf{T}}\right] = \mathbf{H}_{\boldsymbol{\theta}}$, yielding

$$\mathbb{E}\left[\|\boldsymbol{u}_{\mathrm{init}} - \boldsymbol{u}\|_{\mathbf{H}_{\boldsymbol{\theta}}}^2\right] = \mathrm{tr}\left(\mathbb{E}\left[\boldsymbol{\xi}\boldsymbol{\xi}^{\mathsf{T}}\right]\mathbf{H}_{\boldsymbol{\theta}}^{-1}\right) = \mathrm{tr}\left(\mathbf{H}_{\boldsymbol{\theta}}\,\mathbf{H}_{\boldsymbol{\theta}}^{-1}\right) = \mathrm{tr}\left(\mathbf{I}\right) = n. \quad (15)$$

The initial distance for the standard estimator is equal to the trace of $\mathbf{H}_{\boldsymbol{\theta}}^{-1}$, whereas it is constant for the pathwise estimator. Figure 3 illustrates that this trace follows the top eigenvalue, which roughly matches the noise precision. As the model fits the data better, the noise precision increases, increasing the initial distance for the standard but not for the pathwise estimator. In practice, the latter leads to faster solver convergence, especially for problems with high noise precision (see Table 1).

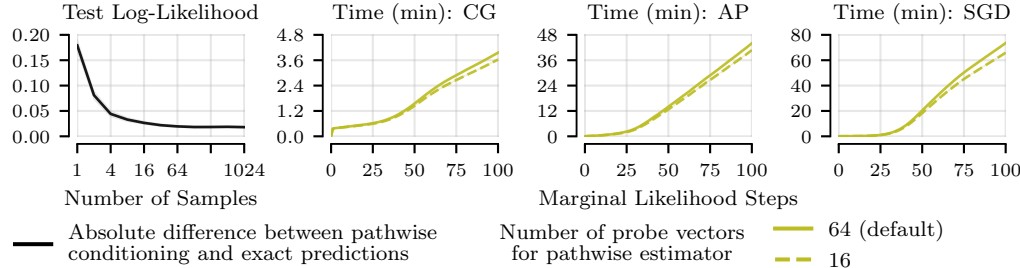

Figure 4: On the POL dataset, increasing the number of posterior samples improves the performance of pathwise conditioning until diminishing returns start to manifest with more than 64 samples (left). Furthermore, with $4\times$ as many probe vectors, the total cumulative runtime only increases by around 10% because the computational costs are dominated by shared kernel function evaluations (right).

**Amortising Linear Solves for Optimisation and Prediction**    The name of the *pathwise* estimator comes from the fact that solving the linear systems (11) provides us with all of the terms we need to construct a set of $s$ posterior samples via pathwise conditioning (3). Each of these is given by

$$(f|\boldsymbol{y})(\cdot) = f(\cdot) + k(\cdot, \boldsymbol{x}; \boldsymbol{\vartheta}) \, \mathbf{H}_{\boldsymbol{\theta}}^{-1}(\boldsymbol{y} - \boldsymbol{\xi}) = f(\cdot) + k(\cdot, \boldsymbol{x}; \boldsymbol{\vartheta})(\boldsymbol{v_y} - \hat{\boldsymbol{z}}). \tag{16}$$

We can use these to make predictions without requiring any additional linear system solves.

**How Many Probe Vectors and Posterior Samples Do We Need?**    In the literature [9, 18, 2, 33], it is common to use $s \leq 16$ probe vectors for marginal likelihood optimisation. However, a larger number of posterior samples, around $s = 64$, is necessary to make accurate predictions [2, 15, 16] (see Figure 4). Thus, to amortise linear system solves across marginal likelihood optimisation and prediction, we must use the same number of probes for both. Interestingly, as shown in Figure 4, using 64 instead of 16 probe vectors only increases the runtime by around 10% because the computational costs are dominated by kernel function evaluations, which are shared among probe vectors.

**Estimator Variance**    The standard estimator with Gaussian probe vectors and the pathwise estimator have the same variance if $\mathbf{H}_{\boldsymbol{\theta}}$ and $\partial \mathbf{H}_{\boldsymbol{\theta}} / \partial \theta_k$ commute with each other (see Appendix A.1). There has been work developing trace estimators with lower variance [19, 8], however, we did not pursue these as we find variance to be sufficiently low, even when relying on only $s = 16$ probe vectors.

**Approximate Prior Function Samples Using Random Features**    In practice, the pathwise estimator requires samples from the prior $f \sim \mathrm{GP}(0, k)$, which are intractable for large datasets without the use of random features [31, 32]. In Figure 5, we show that, despite using random features, most of the time the marginal likelihood optimisation trajectory of the pathwise estimator matches the trajectory of exact optimisation using Cholesky factorisation and backpropagation. Further, we confirm that deviations of the pathwise estimator are indeed due to the use of random features by demonstrating that we can remove these deviations using exact samples from the prior instead.

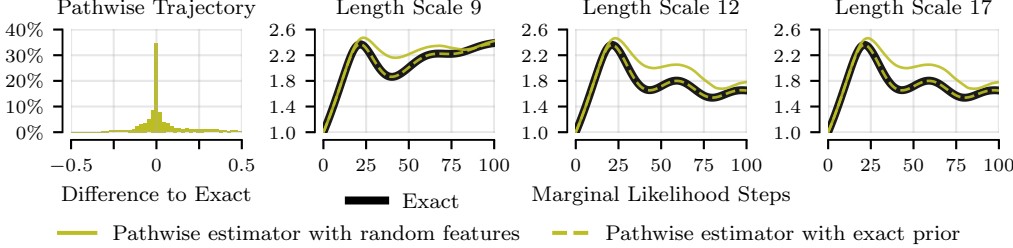

Figure 5: Across all datasets and marginal likelihood steps, most hyperparameter trajectories of the pathwise estimator rarely differ from exact optimisation, as shown by the histogram illustrating the differences between hyperparameters (left). On selected length scales of the ELEVATORS dataset, the pathwise estimator deviates due to the use of random features to approximate prior function samples. With exact samples from the prior, the pathwise estimator matches exact optimisation again (right).

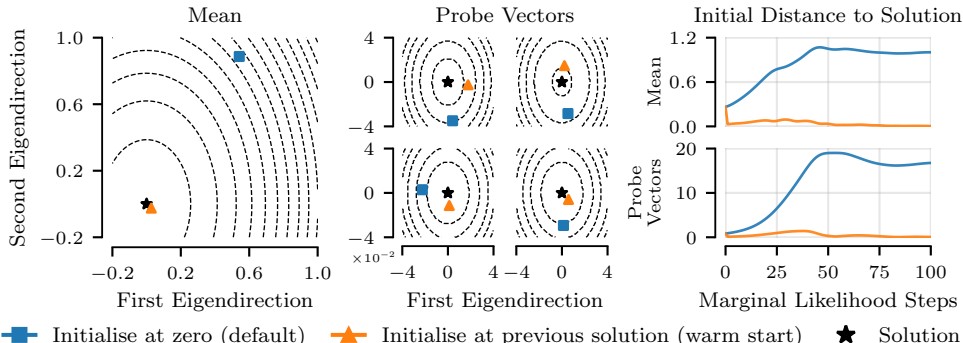

Figure 6: Two-dimensional cross-sections along top eigendirections of the inner-loop quadratic objective after 20 marginal likelihood steps on the POL dataset. The current solution is placed at the origin of coordinates (left and middle). Warm starting significantly reduces the initial root-mean-square RKHS distance to the solution throughout marginal likelihood optimisation (right).

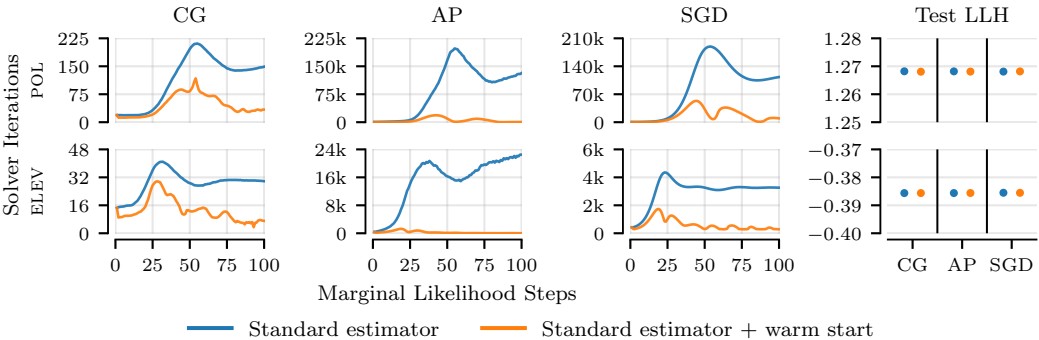

Figure 7: Required number of solver iterations to the tolerance $\tau = 0.01$ during marginal likelihood optimisation on the POL and ELEVATORS datasets. Warm starting with the previous solution reduces the required number of iterations to reach the tolerance without sacrificing predictive performance.

## 4   Warm Starting Linear System Solvers

Linear system solvers are typically initialised at zero [9, 30, 1, 15, 33, 16].[1] However, because the outer-loop marginal likelihood optimisation does not change the hyperparameters much between consecutive steps, we expect that the solution to inner-loop linear systems also does not change much between consecutive steps (see Appendix A.2 for a more formal argument). Therefore, we suggest to *warm start* linear system solvers by initialising them at the solution of the previous [17]. This requires that the targets of the linear systems, $z_j$ or $\xi_j$, are not resampled throughout optimisation, which can introduce bias [5]. However, we find that warm starting consistently provides gains across all linear system solvers for both the standard and the pathwise estimator, and that the bias is negligible.

**Visualising Warm Starts**   Figure 6 visualises the two top eigendirections of the inner-loop quadratic objective on POL. Throughout training, warm starting solvers at the solution to the previous linear system results in a substantially smaller initial distance to the current solution.

**Effects on Linear System Solver Convergence**   Reducing the initial RKHS distance to the solution reduces the required number of solver iterations until the tolerance $\tau = 0.01$ is reached for all solvers and all five datasets, as shown in Figure 7, Table 1 and Appendix C. However, the effectiveness depends on the solver type. CG is more sensitive to the direction of descent rather than the distance to the solution because it uses line searches to take big steps. It only obtains a $2.1\times$ speed-up on average. AP and SGD benefit more, with average speed-ups of $18.9\times$ and $5.1\times$, respectively.

---

[1]Notable exceptions are Artemev et al. [3], who warm start $v_y$ in a sparse lower bound on $\mathcal{L}$, and Antorán et al. [2], who warm start a stochastic gradient descent solver for finite-dimensional linear models.

Table 1: Test log-likelihoods, total training times, and average speed-up among datasets for CG, AP, and SGD after 100 outer-loop marginal likelihood steps with learning rate of 0.1. We consider five datasets with $n < 50$k, which allows us to solve to tolerance, and report the mean over 10 data splits.

| | path wise | warm start | Test Log-Likelihood | | | | | Total Time (min) | | | | | Average Speed-Up |
|---|---|---|---|---|---|---|---|---|---|---|---|---|---|
| | | | POL | ELEV | BIKE | PROT | KEGG | POL | ELEV | BIKE | PROT | KEGG | |
| CG | | | 1.27 | -0.39 | 2.15 | -0.59 | 1.08 | 4.83 | 1.58 | 5.08 | 29.9 | 28.0 | — |
| | ✓ | | 1.27 | -0.39 | 2.07 | -0.62 | 1.08 | 3.96 | 1.49 | 4.41 | 20.0 | 26.4 | **1.2 ×** |
| | | ✓ | 1.27 | -0.39 | 2.15 | -0.59 | 1.08 | 2.28 | 1.03 | 2.74 | 11.5 | 12.8 | **2.1 ×** |
| | ✓ | ✓ | 1.27 | -0.39 | 2.06 | -0.62 | 1.08 | 2.47 | 1.00 | 3.07 | 13.7 | 13.0 | **1.9 ×** |
| AP | | | 1.27 | -0.39 | 2.15 | -0.59 | — | 493. | 77.8 | 302. | 131. | > 24 h | — |
| | ✓ | | 1.27 | -0.39 | 2.07 | -0.62 | 1.08 | 27.9 | 1.67 | 19.9 | 16.4 | 211. | **> 5.4 ×** |
| | | ✓ | 1.27 | -0.39 | 2.15 | -0.59 | 1.08 | 44.0 | 36.4 | 35.1 | 55.8 | 491. | **> 18.9 ×** |
| | ✓ | ✓ | 1.27 | -0.39 | 2.06 | -0.62 | 1.08 | 3.90 | 1.21 | 5.40 | 12.3 | 14.0 | **> 72.1 ×** |
| SGD | | | 1.27 | -0.39 | 2.15 | -0.59 | 1.08 | 139. | 5.54 | 412. | 75.2 | 620. | — |
| | ✓ | | 1.27 | -0.39 | 2.07 | -0.63 | 1.08 | 73.6 | 4.58 | 156. | 24.0 | 412. | **2.1 ×** |
| | | ✓ | 1.27 | -0.39 | 2.15 | -0.59 | 1.08 | 26.5 | 1.22 | 74.3 | 11.2 | 168. | **5.1 ×** |
| | ✓ | ✓ | 1.27 | -0.39 | 2.06 | -0.62 | 1.07 | 17.9 | 1.14 | 64.2 | 11.9 | 58.7 | **7.2 ×** |

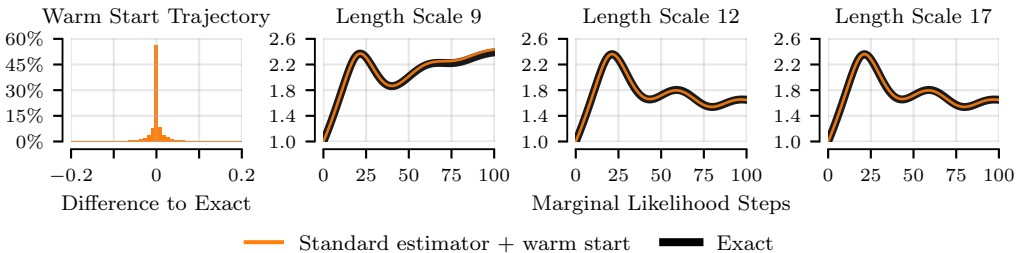

Standard estimator + warm start       Exact

Figure 8: Across all marginal likelihood steps and datasets, warm starting results in hyperparameter trajectories which barely differ from exact optimisation, as shown by the histogram (left). On the same selected length scales from Figure 5, warm starting matches exact optimisation (right).

**Does Warm Starting Introduce Bias?**    A potential concern when warm starting is that the latter introduces bias into the optimisation trajectory because the linear system targets are not resampled throughout optimisation. Although individual gradient estimates are unbiased, estimates are correlated along the optimisation trajectory. In fact, after fixing the targets, gradients become deterministic and it is unclear whether the induced optimum converges to the true optimum.[2] Fortunately, one can show that the marginal likelihood at the optimum implied by these gradients will converge in probability to the marginal likelihood of the true optimum.

**Theorem 1.** *(informal) Under reasonable assumptions, the marginal likelihood $\mathcal{L}$ of the hyperparameters obtained by maximising the objective implied by the warm-started gradients $\tilde{\boldsymbol{\theta}}^*$ will converge in probability to the marginal likelihood of a true maximum $\boldsymbol{\theta}^*$: $\mathcal{L}(\tilde{\boldsymbol{\theta}}^*) \xrightarrow{p} \mathcal{L}(\boldsymbol{\theta}^*)$ as $s \to \infty$.*

See Appendices A.3 and A.4 for details. In practice, a small number of samples seems to be sufficient. In Appendix C, we illustrate that optimisation trajectories of warm-started solvers are almost identical to trajectories obtained by non-warm-started solvers across solver types and datasets.

**Warm Starting the Pathwise Estimator**    One advantage of the pathwise estimator from Section 3 is the reduced RKHS distance between the origin and the solution. When warm starting, the inner-loop solver no longer initialises at the origin, and thus, one may be concerned that we lose this advantage. However, empirically, this is not the case. As shown in Table 1, combining both techniques further accelerates AP and SGD, reaching $72.1\times$ and $7.2\times$ average speed-ups across our datasets relative to the standard estimator without warm starting. Furthermore, since we run solvers until reaching the tolerance, the predictive performance is almost identical among all methods and solvers.

---

[2]The concern might be likened to how pointwise convergence of integrable functions does not always imply convergence of the integrals of those functions, potentially biasing the optima of the limit of the integrals.

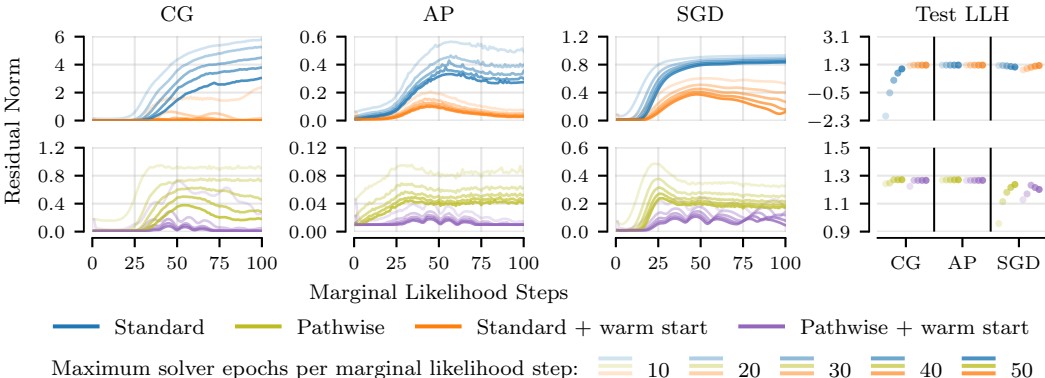

Figure 9: Relative residual norms of the probe vector linear systems at each marginal likelihood step on the POL dataset when solving until the tolerance or a maximum number of solver epochs is reached. Increasing the compute budget generally reduces the residual norm. Given the same compute budget, the pathwise estimator reaches lower residual norms than the standard estimator. Adding warm starts further reduces the residual norm for both estimators. However, the final test log-likelihood does not always match the residual norm. Surprisingly, good predictive performance can be obtained even if the residual norm is much higher than the tolerance $\tau = 0.01$.

## 5    Solving Linear Systems on a Limited Compute Budget

Our experiments so far have only considered relatively small datasets with $n < 50k$, such that inner-loop solvers can reach the tolerance in a reasonable amount of time. However, on large datasets, where linear system conditioning may be poor, reaching a low relative residual norm can become computationally infeasible. Instead, linear system solvers are commonly given a limited compute budget. Gardner et al. [9] limit the number of CG iterations to 20, Wu et al. [33] use 11 epochs of AP, Antorán et al. [2] run SGD for 50k iterations, and Lin et al. [15, 16] run SGD for 100k iterations. While effective for managing computational costs, it is not well understood how early stopping before reaching the tolerance affects different solvers and marginal likelihood optimisation. Furthermore, it is unclear whether a certain tolerance is required to obtain good downstream predictive performance.

**The Effects of Early Stopping**    We repeat the experiments from Table 1 but introduce limited compute budgets: 10, 20, 30, 40 or 50 solver epochs, where one epoch refers to computing each value in $\mathbf{H}_{\boldsymbol{\theta}}$ once (see Appendix B for details).[3] In this setting, solvers terminate upon either reaching the relative residual norm tolerance or when the compute budget is exhausted, whichever occurs first.

In Figure 9, we illustrate the relative residual norms reached for each compute budget on the POL dataset (see Figures 14 to 17 in Appendix C for other datasets). In general, the residual norms increase as $\mathbf{H}_{\boldsymbol{\theta}}$ becomes more ill-conditioned during optimisation, and as the compute budget is decreased. The increase in residual norms is much larger for CG than the other solvers, which is consistent with previous reports of CG not being amenable to early stopping [15]. AP seems to behave slightly better than SGD under a limited compute budget. Both the pathwise estimator and warm starting combine well with early stopping, reaching lower residual norms when using a budget of 10 solver epochs than the standard estimator without warm starting using a budget of 50 solver epochs.

In terms of predictive performance, we see that CG with the standard estimator and no warm starting suffers the most from early stopping. Changing to the pathwise estimator and warm starting recovers good performance most of the time. SGD also shows some sensitivity to early stopping, but there seems to be a stronger correlation between invested compute and final performance. Surprisingly, AP generally achieves good predictive performance even on the smallest compute budget, despite not reaching the tolerance of $\tau = 0.01$. Overall, the relationship between reaching a low residual norm and obtaining good predictive performance seems to be weak. This is an unexpected yet interesting observation, and future research should investigate the suitability of the relative residual norm as a metric to determine solver convergence.

---

[3]Because kernel function evaluations dominate the computational costs of linear system solvers, this results in similar time budgets across methods while preventing compute wastage due to time-based stopping.

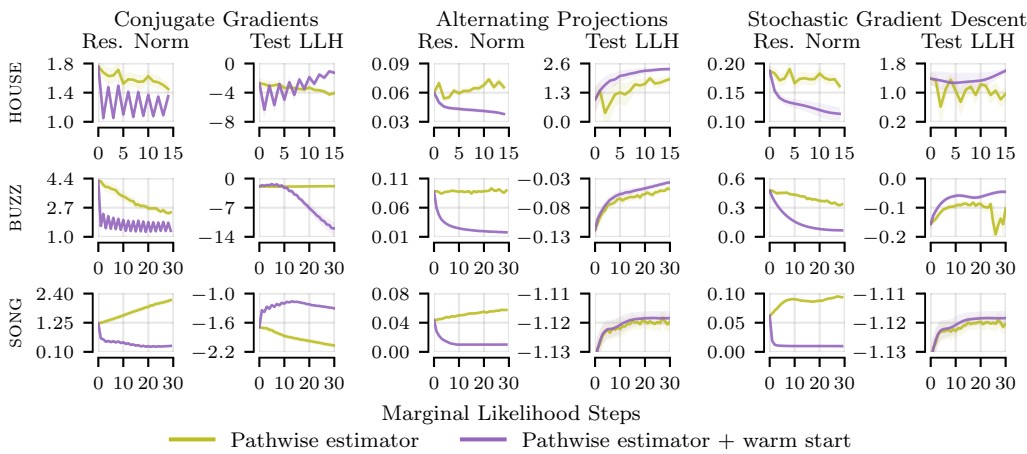

Figure 10: Relative residual norms and test log-likelihoods during marginal likelihood optimisation on large datasets using the pathwise estimator. Warm starting allows solver progress to accumulate over multiple marginal likelihood steps, leading to decreasing residual norms. Without warm starting, residual norms tend to remain similar or increase during optimisation. Despite reaching significantly lower residual norms, the predictive performance does not always improve, akin to Figure 9.

**Demonstration on Large Datasets** After analysing early stopping on small datasets, we now turn to larger UCI datasets $391k < n < 1.8M$, where solving until reaching the tolerance becomes computationally infeasable. Thus, we introduce a compute budget of 10 solver epochs per marginal likelihood step. Hyperparameters are initialised with the heuristic of Lin et al. [15] and optimised using a learning rate of 0.03 for 30 Adam steps (15 for HOUSEELECTRIC due to high computational costs). We use the pathwise estimator because it accelerates solver convergence (see Section 3), and it enables efficient tracking of predictive performance during optimisation. See Appendix B for details.

Figure 10 visualises the evolution of the relative residual norm of the probe vector linear systems and the predictive test log-likelihood during marginal likelihood optimisation. A full set of results is in Appendix C. For all solvers, warm starting leads to lower residual norms throughout outer-loop steps. This suggests a synergistic behaviour between early stopping and warm starting: the latter allows solver progress to accumulate across marginal likelihood steps. This can be interpreted as amortising the inner-loop linear system solve over multiple outer-loop steps. Despite the lower residual norm, CG is brittle under early stopping, obtaining significantly worse performance than AP and SGD on BUZZ and HOUSEELECTRIC. AP and SGD seem to be more robust to early stopping. However, lower residual norms do not always translate to improved predictive performance. Furthermore, we find that SGD can suffer due to the optimal learning rate changing as the hyperparameters change.

## 6    Conclusion

Building upon a hierarchical view of marginal likelihood optimisation, this paper consolidates several iterative GP techniques into a common framework, analysing them and showing their applicability across different linear system solvers. Overall, these provide speed-ups of up to $72\times$ when solving until a specified tolerance is reached, and decrease the average relative residual norm by up to $7\times$ under a limited compute budget. Additionally, our analyses lead to the following findings: Firstly, the pathwise gradient estimater accelerates linear system solvers by moving solutions closer to the origin, and also provides amortised predictions as an added benefit by turning probe vectors into posterior samples via pathwise conditioning. Secondly, warm starting solvers at previous solutions during marginal likelihood optimisation reduces the number of solver iterations to tolerance at the cost of introducing negligible bias into the optimisation trajectory. Furthermore, warm starting combines well with pathwise gradient estimation. Finally, stopping linear system solvers after exhausting a limited compute budget generally increases the relative residual norm. However, when paired with warm starting, solver progress accumulates, amortising inner-loop linear system solves over multiple outer-loop steps. Nonetheless, we observe that low relative residual norms are not always necessary to obtain good predictive performance, which presents an interesting avenue for future research.

## Acknowledgments

Jihao Andreas Lin and Shreyas Padhy were supported by the University of Cambridge Harding Distinguished Postgraduate Scholars Programme. José Miguel Hernández-Lobato acknowledges support from a Turing AI Fellowship under grant EP/V023756/1. We thank Runa Eschenhagen for helpful discussions. This work was performed using resources provided by the Cambridge Service for Data Driven Discovery (CSD3) operated by the University of Cambridge Research Computing Service (www.csd3.cam.ac.uk), provided by Dell EMC and Intel using Tier-2 funding from the Engineering and Physical Sciences Research Council (capital grant EP/T022159/1), and DiRAC funding from the Science and Technology Facilities Council (www.dirac.ac.uk). This work was also supported with Cloud TPUs from Google's TPU Research Cloud (TRC).

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

## A  Mathematical Derivations

In this appendix, we provide mathematical derivations for claims in the main paper.

### A.1  Variance of Standard and Pathwise Gradient Estimator

To compare the variances of the standard estimator (6) and the pathwise estimator (9), we calculate

$$\text{Var}\left( \boldsymbol{z}^\mathsf{T} \mathbf{H}_{\boldsymbol{\theta}}^{-1} \frac{\partial \mathbf{H}_{\boldsymbol{\theta}}}{\partial \theta_k} \boldsymbol{z} \right) \text{ with } \boldsymbol{z} \sim \mathcal{N}(\mathbf{0}, \mathbf{I}) \text{ and } \text{Var}\left( \hat{\boldsymbol{z}}^\mathsf{T} \frac{\partial \mathbf{H}_{\boldsymbol{\theta}}}{\partial \theta_k} \hat{\boldsymbol{z}} \right) \text{ with } \hat{\boldsymbol{z}} \sim \mathcal{N}(\mathbf{0}, \mathbf{H}_{\boldsymbol{\theta}}^{-1}). \quad (17)$$

The variance of the standard estimator is given by

$$\text{Var}\left( \boldsymbol{z}^\mathsf{T} \mathbf{H}_{\boldsymbol{\theta}}^{-1} \frac{\partial \mathbf{H}_{\boldsymbol{\theta}}}{\partial \theta_k} \boldsymbol{z} \right) = \text{tr}\left( \mathbf{H}_{\boldsymbol{\theta}}^{-1} \frac{\partial \mathbf{H}_{\boldsymbol{\theta}}}{\partial \theta_k} \left( \mathbf{H}_{\boldsymbol{\theta}}^{-1} \frac{\partial \mathbf{H}_{\boldsymbol{\theta}}}{\partial \theta_k} + \frac{\partial \mathbf{H}_{\boldsymbol{\theta}}}{\partial \theta_k} \mathbf{H}_{\boldsymbol{\theta}}^{-1} \right) \right), \quad (18)$$

$$= \text{tr}\left( \mathbf{H}_{\boldsymbol{\theta}}^{-1} \frac{\partial \mathbf{H}_{\boldsymbol{\theta}}}{\partial \theta_k} \mathbf{H}_{\boldsymbol{\theta}}^{-1} \frac{\partial \mathbf{H}_{\boldsymbol{\theta}}}{\partial \theta_k} \right) + \text{tr}\left( \mathbf{H}_{\boldsymbol{\theta}}^{-1} \frac{\partial \mathbf{H}_{\boldsymbol{\theta}}}{\partial \theta_k} \frac{\partial \mathbf{H}_{\boldsymbol{\theta}}}{\partial \theta_k} \mathbf{H}_{\boldsymbol{\theta}}^{-1} \right). \quad (19)$$

The variance of the pathwise estimator is given by

$$\text{Var}\left( \hat{\boldsymbol{z}}^\mathsf{T} \frac{\partial \mathbf{H}_{\boldsymbol{\theta}}}{\partial \theta_k} \hat{\boldsymbol{z}} \right) = 2\, \text{tr}\left( \frac{\partial \mathbf{H}_{\boldsymbol{\theta}}}{\partial \theta_k} \mathbf{H}_{\boldsymbol{\theta}}^{-1} \frac{\partial \mathbf{H}_{\boldsymbol{\theta}}}{\partial \theta_k} \mathbf{H}_{\boldsymbol{\theta}}^{-1} \right), \quad (20)$$

$$= \text{tr}\left( \mathbf{H}_{\boldsymbol{\theta}}^{-1} \frac{\partial \mathbf{H}_{\boldsymbol{\theta}}}{\partial \theta_k} \mathbf{H}_{\boldsymbol{\theta}}^{-1} \frac{\partial \mathbf{H}_{\boldsymbol{\theta}}}{\partial \theta_k} \right) + \text{tr}\left( \mathbf{H}_{\boldsymbol{\theta}}^{-1} \frac{\partial \mathbf{H}_{\boldsymbol{\theta}}}{\partial \theta_k} \mathbf{H}_{\boldsymbol{\theta}}^{-1} \frac{\partial \mathbf{H}_{\boldsymbol{\theta}}}{\partial \theta_k} \right). \quad (21)$$

Therefore, the variances of both estimators share the first trace term and only differ in the second trace term. Hence, their variances will be identical if

$$\text{tr}\left( \mathbf{H}_{\boldsymbol{\theta}}^{-1} \frac{\partial \mathbf{H}_{\boldsymbol{\theta}}}{\partial \theta_k} \frac{\partial \mathbf{H}_{\boldsymbol{\theta}}}{\partial \theta_k} \mathbf{H}_{\boldsymbol{\theta}}^{-1} \right) \overset{!}{=} \text{tr}\left( \mathbf{H}_{\boldsymbol{\theta}}^{-1} \frac{\partial \mathbf{H}_{\boldsymbol{\theta}}}{\partial \theta_k} \mathbf{H}_{\boldsymbol{\theta}}^{-1} \frac{\partial \mathbf{H}_{\boldsymbol{\theta}}}{\partial \theta_k} \right), \quad (22)$$

which is the case if $\mathbf{H}_{\boldsymbol{\theta}}^{-1}$ and $\partial \mathbf{H}_{\boldsymbol{\theta}} / \partial \theta_k$ commute with each other.

For example, consider the derivative with respect to the noise scale $\sigma$,

$$\frac{\partial \mathbf{H}_{\boldsymbol{\theta}}}{\partial \sigma} = \frac{\partial}{\partial \sigma}\left( k(\boldsymbol{x}, \boldsymbol{x}; \boldsymbol{\vartheta}) + \sigma^2 \mathbf{I} \right) = 2\sigma \mathbf{I}. \quad (23)$$

In this case, $\mathbf{H}_{\boldsymbol{\theta}}^{-1}$ and $\partial \mathbf{H}_{\boldsymbol{\theta}} / \partial \sigma$ commute with each other, such that both estimators have the same variance. In general, a sufficient condition for matrix multiplication to be commutative is simultaneous diagonalisability of two matrices.

### A.2  Taylor Approximation View of Warm Start

At iterations $t$ and $t + 1$ of the outer-loop marginal likelihood optimiser, associated with $\boldsymbol{\theta}^{(t)}$ and $\boldsymbol{\theta}^{(t+1)}$, the linear system solver must solve two batches of linear systems, namely

$$\mathbf{H}_{\boldsymbol{\theta}}^{(t)} \left[ \boldsymbol{v}_{\boldsymbol{y}}^{(t)}, \boldsymbol{v}_1^{(t)}, \ldots, \boldsymbol{v}_s^{(t)} \right] = \left[ \boldsymbol{y}, \boldsymbol{z}_1^{(t)}, \ldots, \boldsymbol{z}_s^{(t)} \right] \quad \text{and} \quad (24)$$

$$\mathbf{H}_{\boldsymbol{\theta}}^{(t+1)} \left[ \boldsymbol{v}_{\boldsymbol{y}}^{(t+1)}, \boldsymbol{v}_1^{(t+1)}, \ldots, \boldsymbol{v}_s^{(t+1)} \right] = \left[ \boldsymbol{y}, \boldsymbol{z}_1^{(t+1)}, \ldots, \boldsymbol{z}_s^{(t+1)} \right], \quad (25)$$

where $\mathbf{H}_{\boldsymbol{\theta}}^{(t)}$ and $\mathbf{H}_{\boldsymbol{\theta}}^{(t+1)}$ are related through the change from $\boldsymbol{\theta}^{(t)}$ to $\boldsymbol{\theta}^{(t+1)}$ and $\boldsymbol{v}_{\boldsymbol{y}}^{(t)}$ and $\boldsymbol{v}_{\boldsymbol{y}}^{(t+1)}$ are further related through sharing the same right-hand side $\boldsymbol{y}$ in the linear system. In such a setting, where the coefficient matrix only changes slightly and the right-hand side remains fixed, we can approximate $\boldsymbol{v}^{(t+1)}$ using a first-order Taylor expansion of $\mathbf{H}_{\boldsymbol{\theta}}^{(t+1)}$,

$$\left( \mathbf{H}_{\boldsymbol{\theta}}^{(t+1)} \right)^{-1} \approx \left( \mathbf{H}_{\boldsymbol{\theta}}^{(t)} \right)^{-1} - \left( \mathbf{H}_{\boldsymbol{\theta}}^{(t)} \right)^{-1} \left( \mathbf{H}_{\boldsymbol{\theta}}^{(t+1)} - \mathbf{H}_{\boldsymbol{\theta}}^{(t)} \right) \left( \mathbf{H}_{\boldsymbol{\theta}}^{(t)} \right)^{-1}, \quad (26)$$

$$\boldsymbol{v}^{(t+1)} \approx \boldsymbol{v}^{(t)} - \left( \mathbf{H}_{\boldsymbol{\theta}}^{(t)} \right)^{-1} \left( \mathbf{H}_{\boldsymbol{\theta}}^{(t+1)} - \mathbf{H}_{\boldsymbol{\theta}}^{(t)} \right) \boldsymbol{v}^{(t)}. \quad (27)$$

If $\Delta = \mathbf{H}_{\boldsymbol{\theta}}^{(t+1)} - \mathbf{H}_{\boldsymbol{\theta}}^{(t)}$ is small then $\boldsymbol{v}^{(t)}$ will be close to $\boldsymbol{v}^{(t+1)}$, such that we can reuse $\boldsymbol{v}^{(t)}$ to initialise the linear system solver when solving for $\boldsymbol{v}^{(t+1)}$. To satisfy the condition of fixed right-hand sides, we must set $\boldsymbol{z}_j^{(t)} = \boldsymbol{z}_j$ at the cost of introducing some bias throughout optimisation, which turns out to be negligible in practice (see Section 4 and Appendices A.3 and A.4 for details).

## A.3 Convergence of Warm Starting Marginal Likelihood Optimisation

Recall the gradient of the marginal likelihood objective:

$$\frac{\partial \mathcal{L}(\boldsymbol{\theta})}{\partial \theta_k} = \frac{1}{2}(\mathbf{H}_{\boldsymbol{\theta}}^{-1}\boldsymbol{y})^{\mathsf{T}}\frac{\partial \mathbf{H}_{\boldsymbol{\theta}}}{\partial \theta_k}\mathbf{H}_{\boldsymbol{\theta}}^{-1}\boldsymbol{y} - \frac{1}{2}\mathrm{tr}\left(\mathbf{H}_{\boldsymbol{\theta}}^{-1}\frac{\partial \mathbf{H}_{\boldsymbol{\theta}}}{\partial \theta_k}\right) \qquad k \in \{1, \ldots, d_{\boldsymbol{\theta}}\},$$

where $\boldsymbol{H}_\theta \in \mathbb{R}^{n\times n}$ is a positive semi-definite symmetric matrix, $\boldsymbol{y} \in \mathbb{R}^n$ is a real vector, $n$ is the number of "data" examples, and $d_{\boldsymbol{\theta}}$ is the number of hyperparameters. Also, recall the the warm start estimator $\tilde{g}_k(\boldsymbol{\theta})$ to the gradient $\partial \mathcal{L}(\boldsymbol{\theta})/\partial \theta_k$:

$$\tilde{g}_k(\boldsymbol{\theta}) = \frac{1}{s}\sum_{j=1}^{s} \boldsymbol{z}_j^{\mathsf{T}}\mathbf{H}_{\boldsymbol{\theta}}^{-1}\frac{\partial \mathbf{H}_{\boldsymbol{\theta}}}{\partial \theta_k}\boldsymbol{z}_j,$$

where the *probe vectors* $\boldsymbol{z}_j$ are random variables with identity second moments: $\mathbb{E}[\boldsymbol{z}_j\boldsymbol{z}_j^{\mathsf{T}}] = I$, and $s$ is the number of probe vectors in the trace estimator.

**Notation** We will write $\mathcal{S}^{n-1} \overset{\text{def}}{=} \{\boldsymbol{x} \in \mathbb{R}^n : \|\boldsymbol{x}\|_2 = 1\}$ for a sphere in $\mathbb{R}^n$. For a real matrix $\mathbf{A} \in \mathbb{R}^{m\times n}$, we will denote the operator (spectral) norm $\sup_{\boldsymbol{x}\in\mathcal{S}^{n-1}}\sup_{\boldsymbol{y}\in\mathcal{S}^{m-1}}\boldsymbol{y}^{\mathsf{T}}\mathbf{A}\boldsymbol{x}$ with $\|\mathbf{A}\|_{\mathrm{op}}$.

**Definition 2** (*Sub-gaussian norm*). *The **sub-gaussian norm** of a sub-gaussian random variable $X$ is defined as:*

$$\|X\|_{\psi_2} = \inf\{t > 0 : \mathbb{E}\left[X^2/t^2\right] \leq 2\}$$

**Definition 3** (Sub-exponential norm). *The **sub-exponential norm** of a sub-exponential random variable $X$ is defined as:*

$$\|X\|_{\psi_2} = \inf\{t > 0 : \mathbb{E}\left[|X|/t\right] \leq 2\}$$

We will denote the optimisation domain for the hyperparameters as $\Theta$, where we assume $\Theta \subseteq \mathbb{R}^{d_{\boldsymbol{\theta}}}$.

**Theorem 4.** *Assume that the probe vectors $(\boldsymbol{z}_1, \ldots, \boldsymbol{z}_s)$ are zero mean, coordinate-wise independent, and that elements of $\boldsymbol{z}_j$ are sub-gaussian with norm $\|z_{ji}\|_{\psi_2} = \sigma \forall i \in \{1, \ldots, n\}, j \in \{1, \ldots, s\}$. Assume that the sum of the singular values of $\mathbf{H}_{\boldsymbol{\theta}}^{-1}\frac{\partial \mathbf{H}_{\boldsymbol{\theta}}}{\partial \theta_k}$ is, for all $k \in \{1, \ldots, d_{\boldsymbol{\theta}}\}$, upper-bounded on the domain of $\boldsymbol{\theta}$ by $\lambda^{\max}$. Then, for all $\delta > 0$:*

$$\mathbb{P}\left[\left\|\tilde{\boldsymbol{g}}(\boldsymbol{\theta}) - \frac{\partial \mathcal{L}}{\partial \boldsymbol{\theta}}(\boldsymbol{\theta})\right\|_{\infty} < \max\left(\sqrt{\frac{n}{s}C_1\log\left(\frac{9d_{\boldsymbol{\theta}}}{2\delta}\right)}, \frac{n}{s}C_1\log\left(\frac{9d_{\boldsymbol{\theta}}}{2\delta}\right)\right)C_2\sigma\lambda^{\max}\right] > 1 - \delta,$$

The crux of the proof of Theorem 4 comes from bounding the spectral norm of the difference $\left(\left(\sum_{j=1}^{s}\boldsymbol{z}_j\boldsymbol{z}_j^{\mathsf{T}}\right) - \mathbf{I}\right)$. To do so, it is useful to introduce the following definitions and lemmas.

**Lemma 5** (**Computing the operator norm on a net [29, Exercise 4.4.3]**). *Let $\mathbf{A}$ be an $n \times n$ matrix and $\varepsilon \in [0, 1)$. Then, for any $\varepsilon$-net $\Sigma_{\varepsilon}$ of the unit sphere $\mathcal{S}^{n-1}$, we have:*

$$\sup_{\boldsymbol{x}\in\Sigma_{\varepsilon}}\boldsymbol{x}^{\mathsf{T}}\mathbf{A}\boldsymbol{x} \leq \|\mathbf{A}\|_{\mathrm{op}} \leq \frac{1}{1-2\varepsilon}\sup_{\boldsymbol{x}\in\Sigma_{\varepsilon}}\boldsymbol{x}^{\mathsf{T}}\mathbf{A}\boldsymbol{x}.$$

**Lemma 6** (Size of $\varepsilon$-net on $\mathcal{S}^{n-1}$ [29, Corollary 4.2.13]). *There exists an $\varepsilon$-net on $\mathcal{S}^{n-1}$ with cardinality at most $\left(\frac{2}{\varepsilon} + 1\right)^n$.*

**Lemma 7.** *Let $\mathbf{M} = \frac{1}{s}\sum_{j=1}^{s}\boldsymbol{z}_j\boldsymbol{z}_j^{\mathsf{T}} - I$ be an $n \times n$ random matrix, where $z_{ji}$ are independent and identically distributed sub-gaussian random variables with sub-gaussian norm $\|z_{ij}\|_{\psi_2} = \sigma$, and $\boldsymbol{x} \in \mathcal{S}^{n-1}$ any unit vector in $\mathbb{R}^n$. Then:*

$$\mathbb{P}[\boldsymbol{x}^{\mathsf{T}}\mathbf{M}\boldsymbol{x} \geq \beta] \leq 2e^{-C_1\min\left(\frac{\beta^2}{C_2^2\sigma^2}, \frac{\beta}{C_2\sigma}\right)s},$$

*where $C_1, C_2$ are absolute constants.*

*Proof.* We can rewrite:

$$\boldsymbol{x}^\mathsf{T}\mathbf{M}\boldsymbol{x} = \boldsymbol{x}^\mathsf{T}\frac{1}{s}\sum_{j=1}^{s}\left(\boldsymbol{z}_j\boldsymbol{z}_j^\mathsf{T} - I\right)\boldsymbol{x} = \frac{1}{s}\sum_{j=1}^{s}\left(\boldsymbol{x}^\mathsf{T}\boldsymbol{z}_j\right)^2 - \boldsymbol{x}^\mathsf{T}\boldsymbol{x} = \frac{1}{s}\sum_{j=1}^{s}\left(\sum_{i=1}^{n}x_i z_{ij}\right)^2 - 1.$$

Since $\sum_{i=1}^{n}x_i z_{ij}$ is a weighted sum of independent sub-gaussian random variables, it is also sub-gaussian with squared norm:

$$\left\|\sum_{i=1}^{n}x_i z_{ij}\right\|_{\psi_2}^2 \le C\sum_{i=1}^{n}\|x_i z_{ij}\|_{\psi_2}^2 = C\sum_{i=1}^{n}x_i^2\|z_{ij}\|_{\psi_2}^2 = C\sigma^2,$$

where $C$ is an absolute constant [29, Proposition 2.6.1]. Hence, since $\left(\sum_{i=1}^{n}x_i z_{ij}\right)^2$ is the square of a sub-gaussian random variable, it must be sub-exponential with the sub-exponential norm (see [29, Lemma 2.7.6]):

$$\left\|\left(\sum_{i=1}^{n}x_i z_{ij}\right)^2\right\|_{\psi_1} = \left\|\sum_{i=1}^{n}x_i z_{ij}\right\|_{\psi_2}^2 \le C\sigma^2,$$

Lastly, since $\left(\sum_{i=1}^{n}x_i z_{ij}\right)^2$ is sub-exponential, so will the mean-centered counterpart $\left(\sum_{i=1}^{n}x_i z_{ij}\right)^2 - \mathbb{E}[(\sum_{i=1}^{n}x_i z_{ij})^2] = \left(\sum_{i=1}^{n}x_i z_{ij}\right)^2 - 1$ [29, Exercise 2.7.10] with sub-exponential norm:

$$\left\|\left(\sum_{i=1}^{n}x_i z_{ij}\right)^2 - 1\right\|_{\psi_1} \le C_2\sigma^2,$$

where $C_2$ is another absolute constant. Hence, since $\left(\sum_{i=1}^{n}x_i z_{ij}\right)^2 - 1$ for $j \in \{1,\ldots,s\}$ are sub-exponential, zero-mean and independent, we can apply Bernstein's inequality to bound the tail probability of $\boldsymbol{x}^\mathsf{T}\mathbf{M}\boldsymbol{x}$:

$$\mathbb{P}[\boldsymbol{x}^\mathsf{T}\mathbf{M}\boldsymbol{x} > \beta] = \mathbb{P}\left[\frac{1}{s}\sum_{j=1}^{s}\left(\left(\sum_{i=1}^{n}x_i z_{ij}\right)^2 - 1\right) > \beta\right] \le 2e^{-C_1 \min\left(\frac{\beta^2}{C_2^2\sigma^2}, \frac{\beta}{C_2\sigma}\right)s}$$

holds for any $\beta \ge 0$, where $C_1, C_2$ are absolute constants. $\qquad\square$

**Lemma 8.** *Let $\mathbf{M} = \frac{1}{s}\sum_{j=1}^{s}\boldsymbol{z}_j\boldsymbol{z}_j^\mathsf{T} - I$ be an $n \times n$ random matrix, where $z_{ji}$ are independent and identically distributed sub-Gaussian random variables with sub-gaussian norm $\|z_{ij}\|_{\psi_2} = \sigma$. Then:*

$$\mathbb{P}\left[\|\mathbf{M}\|_{\text{op}} \ge \max\left(\sqrt{\frac{n}{s}C_1 \log\left(\frac{9}{2\delta}\right)}, \frac{n}{s}C_1 \log\left(\frac{9}{2\delta}\right)\right)C_4\sigma\right] \le \delta \qquad \forall \delta > 0,$$

*where $C_1, C_4$ are absolute constants.*

*Proof.* Pick an $\varepsilon$-net $\Sigma_\varepsilon$ on $\mathcal{S}^{n-1}$ of size at most $\left(\frac{2}{\varepsilon} + 1\right)^n$ (Lemma 6). Then, we can bound the tail probability of the operator norm as:

$$\mathbb{P}\left[\|\mathbf{M}\|_{\text{op}} \ge \beta\right] \le \mathbb{P}\left[\sup_{\boldsymbol{x}\in\Sigma_\varepsilon}\boldsymbol{x}^\mathsf{T}\mathbf{M}\boldsymbol{x} \ge (1-2\varepsilon)\beta\right] \qquad\qquad \triangle \text{ By Lemma 5}$$

$$\le \sum_{\boldsymbol{x}\in\Sigma_\varepsilon}\mathbb{P}\left[\boldsymbol{x}^\mathsf{T}\mathbf{M}\boldsymbol{x} \ge (1-2\varepsilon)\beta\right] \qquad\qquad \triangle \text{ Union bound}$$

$$\le \|\Sigma_\varepsilon\|2e^{-C_1 \min\left(\frac{\beta^2(1-2\varepsilon)^2}{C_2^2\sigma^2}, \frac{\beta(1-2\varepsilon)}{C_2\sigma}\right)s} \qquad\qquad \triangle \text{ By Lemma 7}$$

$$\le \left(\frac{2}{\varepsilon} + 1\right)^n 2e^{-C_1 \min\left(\frac{\beta^2(1-2\varepsilon)^2}{C_2^2\sigma^2}, \frac{\beta(1-2\varepsilon)}{C_2\sigma}\right)s}$$

Setting $\varepsilon = \frac{1}{4}$ gives:

$$\mathbb{P}\left[\|\mathbf{M}\|_{\text{op}} \ge \beta\right] \le 9^n 2e^{-C_1 \min\left(\frac{\beta^2}{C_4^2\sigma^2}, \frac{\beta}{C_4\sigma}\right)s},$$

where $C_4$ is an absolute constant.

Lastly, to get the bound into the form $\mathbb{P}\left[\|\mathbf{M}\|_{\mathrm{op}} \geq f(\delta, n, s)\right] \leq \delta$, we can note that:

$$9^n 2e^{-C_1 \min\left(\frac{\beta^2}{C_4^2\sigma^2}, \frac{\beta}{C_4\sigma}\right)s} \leq \delta \quad \Leftrightarrow \quad \min\left(\frac{\beta^2}{C_4^2\sigma^2}, \frac{\beta}{C_4\sigma}\right) \geq \frac{n}{s}C_1 \log\left(\frac{9}{2\delta}\right)$$

$$\Leftrightarrow \quad \frac{\beta}{C_4\sigma} \geq \max\left(\sqrt{\frac{n}{s}C_1 \log\left(\frac{9}{2\delta}\right)}, \frac{n}{s}C_1 \log\left(\frac{9}{2\delta}\right)\right),$$

and so by setting $\beta = \max\left(\sqrt{\frac{n}{s}C_1 \log\left(\frac{9}{2\delta}\right)}, \frac{n}{s}C_1 \log\left(\frac{9}{2\delta}\right)\right)C_4\sigma$ we get the desired result:

$$\mathbb{P}\left[\|\mathbf{M}\|_{\mathrm{op}} \geq \max\left(\sqrt{\frac{n}{s}C_1 \log\left(\frac{9}{2\delta}\right)}, \frac{n}{s}C_1 \log\left(\frac{9}{2\delta}\right)\right)C_4\sigma\right] \leq \delta$$

$\square$

*Proof of Theorem 4.* Let $\sum_{i=1}^n \boldsymbol{q}_i \lambda_i \boldsymbol{p}_i^\mathsf{T}$ be a singular value decomposition (SVD) of $\mathbf{A} \stackrel{\text{def}}{=} \mathbf{H}_{\boldsymbol{\theta}}^{-1} \frac{\partial \mathbf{H}_{\boldsymbol{\theta}}}{\partial \theta_k}$, where $\{\boldsymbol{q}_i\}_{i=1}^n$ and $\{\boldsymbol{p}_i\}_{i=1}^n$ are two sets of orthonormal vectors. First, note that we can rewrite:

$$\begin{aligned}
\tilde{g}_k(\boldsymbol{\theta}) - \frac{\partial \mathcal{L}(\boldsymbol{\theta})}{\partial \theta_k} &= \sum_{j=1}^s \boldsymbol{z}_j^\mathsf{T} \mathbf{A} \boldsymbol{z}_j - \mathbb{E}_{\boldsymbol{z}}\left[\boldsymbol{z}^\mathsf{T} \mathbf{A} \boldsymbol{z}\right] \\
&= \sum_{j=1}^s \boldsymbol{z}_j^\mathsf{T} \left(\sum_{i=1}^n \lambda_i \boldsymbol{q}_i \boldsymbol{p}_i^\mathsf{T}\right) \boldsymbol{z}_j - \mathbb{E}_{\boldsymbol{z}}\left[\boldsymbol{z}^\mathsf{T} \left(\sum_{i=1}^n \lambda_i \boldsymbol{q}_i \boldsymbol{p}_i^\mathsf{T}\right) \boldsymbol{z}\right] \quad (28) \\
&= \sum_{i=1}^n \lambda_i \sum_{j=1}^s \boldsymbol{z}_j^\mathsf{T} \boldsymbol{q}_i \boldsymbol{p}_i^\mathsf{T} \boldsymbol{z}_j - \sum_{i=1}^n \lambda_i \mathbb{E}_{\boldsymbol{z}}\left[\boldsymbol{z}^\mathsf{T} \boldsymbol{q}_i \boldsymbol{p}_i^\mathsf{T} \boldsymbol{z}\right] \\
&= \sum_{i=1}^n \lambda_i \left(\sum_{j=1}^s \boldsymbol{z}_j^\mathsf{T} \boldsymbol{q}_i \boldsymbol{p}_i^\mathsf{T} \boldsymbol{z}_j - \mathbb{E}_{\boldsymbol{z}}\left[\boldsymbol{z}^\mathsf{T} \boldsymbol{q}_i \boldsymbol{p}_i^\mathsf{T} \boldsymbol{z}\right]\right) \\
&= \sum_{i=1}^n \lambda_i \left(\boldsymbol{q}_i^\mathsf{T} \left(\sum_{j=1}^s \boldsymbol{z}_j \boldsymbol{z}_j^\mathsf{T}\right) \boldsymbol{p}_i - \boldsymbol{q}_i^\mathsf{T} \underbrace{\mathbb{E}_{\boldsymbol{z}}\left[\boldsymbol{z}\boldsymbol{z}^\mathsf{T}\right]}_{\mathbf{I}} \boldsymbol{p}_i\right) \\
&= \sum_{i=1}^n \lambda_i \boldsymbol{q}_i^\mathsf{T} \underbrace{\left(\left(\sum_{j=1}^s \boldsymbol{z}_j \boldsymbol{z}_j^\mathsf{T}\right) - \mathbf{I}\right)}_{\mathbf{M}} \boldsymbol{p}_i. \quad (29)
\end{aligned}$$

Therefore, we can bound the norm of the difference as

$$|\tilde{g}_k(\boldsymbol{\theta}) - g_k(\boldsymbol{\theta})| \leq \sum_{i=1}^n |\lambda_i| \left|\boldsymbol{q}_i^\mathsf{T} \mathbf{M} \boldsymbol{p}_i\right| \quad (30)$$

$$\leq \sum_{i=1}^n |\lambda_i| \|\mathbf{M}\|_{\mathrm{op}} = \lambda^{\mathrm{max}} \|\mathbf{M}\|_{\mathrm{op}}. \quad (31)$$

By Lemma 8, with probability at least $1 - \delta$ we can bound the operator norm of $\mathbf{M}$ as:

$$|\tilde{g}_k(\boldsymbol{\theta}) - g_k(\boldsymbol{\theta})| \leq \lambda^{\mathrm{max}} \|\mathbf{M}\|_{\mathrm{op}} < \max\left(\sqrt{\frac{n}{s}C_1 \log\left(\frac{9}{2\delta}\right)}, \frac{n}{s}C_1 \log\left(\frac{9}{2\delta}\right)\right)C_2\sigma\lambda^{\mathrm{max}},$$

with $C_1, C_2$ absolute constants.

We can apply a union bound over all $k \in \{1, \ldots, d_{\boldsymbol{\theta}}\}$ to bound the probability of the $\ell_\infty$-norm of the gradient deviating by a certain amount:

$$\mathbb{P}\left[ \left\| \tilde{\boldsymbol{g}}(\boldsymbol{\theta}) - \frac{\partial \mathcal{L}}{\partial \boldsymbol{\theta}}(\boldsymbol{\theta}) \right\|_\infty < \max\left( \sqrt{\frac{n}{s} C_1 \log\left(\frac{9}{2\delta}\right)}, \frac{n}{s} C_1 \log\left(\frac{9}{2\delta}\right) \right) C_2 \sigma \lambda^{\max} \right] > 1 - d_{\boldsymbol{\theta}}\delta,$$

or:

$$\mathbb{P}\left[ \left\| \tilde{\boldsymbol{g}}(\boldsymbol{\theta}) - \frac{\partial \mathcal{L}}{\partial \boldsymbol{\theta}}(\boldsymbol{\theta}) \right\|_\infty < \max\left( \sqrt{\frac{n}{s} C_1 \log\left(\frac{9d_{\boldsymbol{\theta}}}{2\delta}\right)}, \frac{n}{s} C_1 \log\left(\frac{9d_{\boldsymbol{\theta}}}{2\delta}\right) \right) C_2 \sigma \lambda^{\max} \right] > 1 - \delta.$$

$\square$

Now, if $\tilde{\boldsymbol{g}}(\boldsymbol{\theta})$ is a conservative field, and so is implicitly a gradient of some (approximate) objective $\tilde{\mathcal{L}} : \Theta \to \mathbb{R}$, the above result allows us to bound the error on the solution found when optimising using the approximate gradient $\tilde{\boldsymbol{g}}$ instead of the actual gradient $\boldsymbol{g} = \nabla \mathcal{L}$. However, in general, $\tilde{\boldsymbol{g}}(\boldsymbol{\theta})$ need not be strictly conservative. In practice, since $\tilde{\boldsymbol{g}}(\boldsymbol{\theta})$ converges to a conservative field the more samples we take, we may assume that it is close enough to being conservative for the purposes of optimisation on hardware with finite numerical precision. Assuming that $\tilde{\boldsymbol{g}}(\boldsymbol{\theta})$ is conservative allows us to show the following bound on the optimum found when optimising using $\tilde{\boldsymbol{g}}(\boldsymbol{\theta})$, which is a restatement of Theorem 1:

**Theorem 9.** *Let $\tilde{\boldsymbol{g}}$ and $\mathcal{L}$ be defined as in Theorem 4. Assume $\tilde{\boldsymbol{g}} : \Theta \to \mathbb{R}$ is a conservative field. Assume the optimisation domain $\Theta$ is convex, closed and bounded. Then, with probability at least $1 - \delta$:*

$$\mathcal{L}(\tilde{\boldsymbol{\theta}}^*) \geq \mathcal{L}(\boldsymbol{\theta}^*) - \max\left( \sqrt{\frac{n}{s} C_1 \log\left(\frac{9d_{\boldsymbol{\theta}}}{2\delta}\right)}, \frac{n}{s} C_1 \log\left(\frac{9d_{\boldsymbol{\theta}}}{2\delta}\right) \right) C_2 \sigma \lambda^{\max} \Delta\Theta \sqrt{d_{\boldsymbol{\theta}}},$$

*where $\Delta\Theta \stackrel{\text{def}}{=} \sup_{\boldsymbol{\theta},\boldsymbol{\theta}' \in \Theta} \|\boldsymbol{\theta}' - \boldsymbol{\theta}\|$ is the maximum distance between two elements in $\Theta$.*

*Proof.* Let $\tilde{\mathcal{L}} : \Theta \to \mathbb{R}$ be an approximate objective implied by the gradient field $\tilde{\boldsymbol{g}}$, namely a scalar field such that $\nabla \tilde{\mathcal{L}} = \tilde{\boldsymbol{g}}$. Such a scalar field exists if $\tilde{\boldsymbol{g}}$ is a conservative field, and is unique up to a constant (which does not affect the optimum).

For any two points $\boldsymbol{\theta}, \boldsymbol{\theta}' \in \Theta$, with $\Delta\boldsymbol{\theta} \stackrel{\text{def}}{=} \boldsymbol{\theta}' - \boldsymbol{\theta}$, we have that

$$\left| (\mathcal{L}(\boldsymbol{\theta}') - \mathcal{L}(\boldsymbol{\theta})) + \left( \tilde{\mathcal{L}}(\boldsymbol{\theta}') - \tilde{\mathcal{L}}(\boldsymbol{\theta}) \right) \right|$$

$\triangle$ Replace difference in values with integral along path from $\boldsymbol{\theta}$ to $\boldsymbol{\theta}'$

$$= \left| \int_0^1 \frac{\partial}{\partial t} \mathcal{L}\left(\boldsymbol{\theta} + \Delta\boldsymbol{\theta}t\right) dt - \int_0^1 \frac{\partial}{\partial t} \tilde{\mathcal{L}}\left(\boldsymbol{\theta} + \Delta\boldsymbol{\theta}t\right) dt \right|$$

$$= \left| \int_0^1 \Delta\boldsymbol{\theta} \cdot \nabla \mathcal{L}\left(\boldsymbol{\theta} + \Delta\boldsymbol{\theta}t\right) dt - \int_0^1 \Delta\boldsymbol{\theta} \cdot \nabla \tilde{\mathcal{L}}\left(\boldsymbol{\theta} + \Delta\boldsymbol{\theta}t\right) dt \right|$$

$$= \left| \int_0^1 \Delta\boldsymbol{\theta} \cdot \left( \nabla \mathcal{L}\left(\boldsymbol{\theta} + \Delta\boldsymbol{\theta}t\right) - \nabla \tilde{\mathcal{L}}\left(\boldsymbol{\theta} + \Delta\boldsymbol{\theta}t\right) \right) dt \right|$$

$$\leq \int_0^1 \left| \Delta\boldsymbol{\theta} \cdot \left( \nabla \mathcal{L}\left(\boldsymbol{\theta} + \Delta\boldsymbol{\theta}t\right) - \nabla \tilde{\mathcal{L}}\left(\boldsymbol{\theta} + \Delta\boldsymbol{\theta}t\right) \right) \right| dt$$

$$= \int_0^1 \|\Delta\boldsymbol{\theta}\| \left\| \left( \nabla \mathcal{L}\left(\boldsymbol{\theta} + \Delta\boldsymbol{\theta}t\right) - \nabla \tilde{\mathcal{L}}\left(\boldsymbol{\theta} + \Delta\boldsymbol{\theta}t\right) \right) \right\| dt$$

$$\leq \int_0^1 \|\Delta\boldsymbol{\theta}\| \left\| \left( \nabla \mathcal{L}\left(\boldsymbol{\theta} + \Delta\boldsymbol{\theta}t\right) - \nabla \tilde{\mathcal{L}}\left(\boldsymbol{\theta} + \Delta\boldsymbol{\theta}t\right) \right) \right\|_\infty \sqrt{d_{\boldsymbol{\theta}}} dt$$

$\triangle$ Bound $\ell_2$-norm by the $\ell_\infty$-norm

$$\leq \int_0^1 \|\Delta\boldsymbol{\theta}\| \max\left( \sqrt{\frac{n}{s} C_1 \log\left(\frac{9d_{\boldsymbol{\theta}}}{2\delta}\right)}, \frac{n}{s} C_1 \log\left(\frac{9d_{\boldsymbol{\theta}}}{2\delta}\right) \right) C_2 \sigma \lambda^{\max} \sqrt{d_{\boldsymbol{\theta}}} dt$$

△ Difference of gradients bounded with probability at least $(1 - \delta)$ by Theorem 4

$$\leq \max\left( \sqrt{\frac{n}{s} C_1 \log\left(\frac{9 d_{\boldsymbol{\theta}}}{2\delta}\right)}, \frac{n}{s} C_1 \log\left(\frac{9 d_{\boldsymbol{\theta}}}{2\delta}\right) \right) C_2 \sigma \lambda^{\max} \Delta\Theta \sqrt{d_{\boldsymbol{\theta}}}.$$

The above inequality holds with probability at least $(1 - \delta)$. Hence,

$$\mathcal{L}(\boldsymbol{\theta}^*) - \mathcal{L}(\tilde{\boldsymbol{\theta}}^*) \leq \mathcal{L}(\boldsymbol{\theta}^*) - \mathcal{L}(\tilde{\boldsymbol{\theta}}^*) - \overbrace{\left( \tilde{\mathcal{L}}(\boldsymbol{\theta}^*) - \tilde{\mathcal{L}}(\tilde{\boldsymbol{\theta}}^*) \right)}^{\substack{\text{Negative because } \tilde{\boldsymbol{\theta}}^* \text{ is} \\ \text{a maximum of } \tilde{\mathcal{L}}}},$$

$$\leq \left| \mathcal{L}(\boldsymbol{\theta}^*) - \mathcal{L}(\tilde{\boldsymbol{\theta}}^*) - \left( \tilde{\mathcal{L}}(\boldsymbol{\theta}^*) - \tilde{\mathcal{L}}(\tilde{\boldsymbol{\theta}}^*) \right) \right|$$

$$\leq \max\left( \sqrt{\frac{n}{s} C_1 \log\left(\frac{9 d_{\boldsymbol{\theta}}}{2\delta}\right)}, \frac{n}{s} C_1 \log\left(\frac{9 d_{\boldsymbol{\theta}}}{2\delta}\right) \right) C_2 \sigma \lambda^{\max} \Delta\Theta \sqrt{d_{\boldsymbol{\theta}}}.$$

□

**Remark 10.** *Theorem 9 above implies that the objective of the optimum $\tilde{\boldsymbol{\theta}}^*$ obtained with the approximate gradients converges to the objective of the true optimum $\boldsymbol{\theta}^*$ in probability:*

$$\forall \alpha > 0: \quad \mathbb{P}\left[ \left| \mathcal{L}(\tilde{\boldsymbol{\theta}}^*) - \mathcal{L}(\boldsymbol{\theta}^*) \right| > \alpha \right] \to 0 \text{ as } s \to \infty \tag{32}$$

*which trivially follows from the implication of Theorem 9 that for every $\alpha, \delta > 0$, we can find an $s \in \mathbb{N}$ such that $\mathcal{L}(\tilde{\boldsymbol{\theta}}^*) \geq \mathcal{L}(\boldsymbol{\theta}^*) - \alpha$ with probability at least $1 - \delta$.*

**Remark 11** (Convexity of $\Theta$). *We also note that the convexity of the hyperparameter domain $\Theta$ is a fairly mild assumption which is satisfied in the majority of practical settings. For example, optimising kernel length scales and noise scale on bounded intervals $(10^{-10}, 10^{10})$ falls within the assumptions, but introducing a "hole" into the domain (e.g. introducing a constraint like $\|\boldsymbol{\vartheta} - \mathbf{1}\| \geq 0.5$) would break the assumption. In particular, this assumption is* NOT *a statement about the convexity of the objective $\mathcal{L}$ — in the proof, we allow for the objective to be arbitrarily non-convex, and only assume its differentiability.*

### A.4  Convergence of Warm Starting the Pathwise Estimator

The result in Appendix A.3 can be trivially extended for the pathwise estimator in (9) for any pairwise independent probe vectors $\hat{\boldsymbol{z}}_j$ (with second moment $\mathbf{H}_{\boldsymbol{\theta}}^{-1}$) that upon rescaling by $\mathbf{H}_{\boldsymbol{\theta}}^{\frac{1}{2}}$ will be zero-mean with independent coordinates. This is true for probe vectors $\hat{\boldsymbol{z}}_j$ that are either *i.i.d.* $\mathcal{N}(\mathbf{0}, \mathbf{H}_{\boldsymbol{\theta}}^{-1})$-distributed or obtained by transforming Radamacher random variables by $\mathbf{H}_{\boldsymbol{\theta}}^{-\frac{1}{2}}$.

## B  Implementation and Experiment Details

In this appendix, we provide details about our implementation and experiments.

**General**  Our implementation uses the JAX library [4]. All reported experiments were conducted on internal NVIDIA A100-SXM4-80GB GPUs using double floating point precision. Some additional experiments and ablations were performed on Google Cloud TPUs (v4). The total compute time, including preliminary and failed experiments, and evaluation is around 4500 hours. The compute time of individual runs is reported in Tables 2 to 10. The source code is available here.

**Datasets**  Our experiments are conducted using the datasets and data splits from the popular UCI regression benchmark [7]. They consist of various high-dimensional, multivariate regression tasks and are available under the Creative Commons Attribution 4.0 International (CC BY 4.0) license. In particular, we used the POL ($n = 13500, d = 26$), ELEVATORS ($n = 14940, d = 18$), BIKE ($n = 15642, d = 17$), PROTEIN ($n = 41157, d = 9$), KEGGDIRECTED ($n = 43945, d = 20$), 3DROAD ($n = 391387, d = 3$), SONG ($n = 463811, d = 90$), BUZZ ($n = 524925, d = 77$), and HOUSEELECTRIC ($n = 1844352, d = 11$) datasets.

**Kernel Function and Random Features** In all experiments, we used the Matérn-³⁄₂ kernel, parameterised by a scalar signal scale and a length scale per input dimension. For pathwise conditioning (3) and the pathwise gradient estimator (9), we used random Fourier features [21, 25] (1000 sin/cos pairs, 2000 features in total) to draw approximate samples from the Gaussian process prior. For an explanation about how to efficiently sample prior functions from a Gaussian process using random features, we refer to existing literature [31, 32, 15]. However, we want to discuss some details in terms of using this technique for the pathwise estimator from Section 3.

For pathwise gradient estimation, the linear system solver must solve linear systems of the form

$$\mathbf{H}_{\boldsymbol{\theta}} \left[ \boldsymbol{v_y}, \hat{\boldsymbol{z}}_1, \ldots, \hat{\boldsymbol{z}}_s \right] = \left[ \boldsymbol{y}, \boldsymbol{\xi}_1, \ldots, \boldsymbol{\xi}_s \right], \tag{33}$$

with $\boldsymbol{\xi} = f(\boldsymbol{x}) + \boldsymbol{\varepsilon}$, where $f(\boldsymbol{x}) \sim \mathcal{N}(\mathbf{0}, k(\boldsymbol{x}, \boldsymbol{x}; \boldsymbol{\vartheta}))$ is a prior function $f$ sample evaluated at the training data $\boldsymbol{x}$, and $\boldsymbol{\epsilon} \sim \mathcal{N}(\mathbf{0}, \sigma^2 \mathbf{I})$ is a Gaussian random vector. Both quantities are resampled in each outer-loop marginal likelihood step if the pathwise estimator is used without warm starting. With warm starting enabled, the right-hand sides of the linear system must not be resampled. In this case, $f$ and $\boldsymbol{\epsilon}$ are sampled once and fixed afterwards. However, $f$ depends on $\boldsymbol{\vartheta}$ and $\boldsymbol{\epsilon}$ depends on $\sigma$, and both $\boldsymbol{\vartheta}$ and $\sigma$ are hyperparameters which change in each outer-loop step. Therefore, what does it mean to keep $f$ and $\boldsymbol{\epsilon}$ fixed?

For $\boldsymbol{\epsilon}$, this amounts to the reparameterisation $\boldsymbol{\epsilon} = \sigma \, \boldsymbol{w}$, where $\boldsymbol{w} \sim \mathcal{N}(\mathbf{0}, \mathbf{I})$ is sampled once and fixed afterwards, such that $\boldsymbol{\epsilon}$ becomes deterministic. For $f$, this refers to fixing the parameters of the random features, for example the frequencies in the case of random Fourier features. Intuitively, this corresponds to selecting a particular instance of a prior sample, although the distribution of the sample can change due to changes in the hyperparameters. In each outer-loop step, the random features are evaluated using the fixed random feature parameters and the updated kernel hyperparameters, and the prior function sample is then evaluated at the training data using the updated random features. Both of these operations are $\mathcal{O}(n)$ and efficient as long as the number of random features is reasonable.

**Iterative Optimiser** To optimise hyperparameters $\boldsymbol{\theta}$ given an estimate of $\nabla \mathcal{L}$, we used the Adam optimiser [14] with default settings except for the learning rate. For all small datasets ($n < 50$k), we initialised the hyperparameters at 1.0 and used a learning rate of 0.1 to perform 100 steps of Adam. For all large datasets ($n > 50$k), we initialised the hyperparameters using a heuristic and used a learning rate of 0.03 to perform 30 steps of Adam (15 for HOUSEELECTRIC due to high computational costs). The heuristic to obtain initial hyperparameters for the large datasets consists of:

1. Select a centroid data example uniformly at random from the training data.
2. Find the 10k data examples with the smallest Euclidean distance to the centroid.
3. Obtain hyperparameters by maximising the exact marginal likelihood using this subset.
4. Repeat the procedure with 10 different centroids and average the hyperparameters.

This heuristic has previously been used by Lin et al. [15, 16] to avoid aliasing bias.

To enforce positive value constraints during hyperparameter optimisation, we used the `softplus` function. In particular, we reparameterise each hyperparameter $\theta_k \in \mathbb{R}_{>0}$ as $\theta_k = \log(1 + \exp(\nu_k))$ and apply optimiser steps to $\nu_k \in \mathbb{R}$ instead, to facilitate unconstrained optimisation.

**Gradient Estimator** For all experiments, unless otherwise specified, the number of probe vectors was set to $s = 64$ for both the standard and the pathwise estimator. The distributions of the probe vectors are $\boldsymbol{z} \sim \mathcal{N}(\mathbf{0}, \mathbf{I})$ for the standard estimator and $\hat{\boldsymbol{z}} := \mathbf{H}_{\boldsymbol{\theta}}^{-1} \boldsymbol{\xi} \sim \mathcal{N}(\mathbf{0}, \mathbf{H}_{\boldsymbol{\theta}}^{-1})$ for the pathwise estimator. See Section 3 for details about how to generate samples from $\mathcal{N}(\mathbf{0}, \mathbf{H}_{\boldsymbol{\theta}}^{-1})$.

The probe vectors used by Gardner et al. [9] have conceptual similarities but the motivation is different. They used probe vectors $\boldsymbol{z} \sim \mathcal{N}(\mathbf{0}, \mathbf{P})$, where $\mathbf{P}$ is constructed using a low-rank pivoted Cholesky decomposition to implement the preconditioner. In contrast, our pathwise probe vectors are sampled using random features, and, for CG, we used the pivoted Cholesky preconditioner *in addition* to the pathwise probe vectors. Furthermore, we used the solution of the pathwise probe vector systems to construct posterior samples via pathwise conditioning, which has not been done by Gardner et al. [9].

The name of the *pathwise* estimator can be related to the reparameterisation trick by viewing a sample from the GP posterior as a deterministic transformation of a sample from the GP prior, and observing that the latter itself is an affine transformation of a standard normal random variable.

**Linear System Solver**  We conducted two sets of experiments which only differ in the termination criterion of the linear system solver. In the first set, we stop linear system solvers once they reach a relative residual norm tolerance of $\tau = 0.01$. In the second set, we also restrict the maximum number of solver epochs to 10, 20, 30, 40 or 50, such that most of the time the residual norm does not reach $\tau$.

For a generic system of linear equations $\mathbf{H}_{\boldsymbol{\theta}}\, \boldsymbol{u} = \boldsymbol{b}$, the residual is defined as $\boldsymbol{r} = \boldsymbol{b} - \mathbf{H}_{\boldsymbol{\theta}}\, \boldsymbol{u}$ and the relative residual norm is defined as $\|\boldsymbol{r}\|/\|\boldsymbol{b}\|$. In practice, to improve numerical stability, the relative residual norm tolerance is implemented by solving the system $\mathbf{H}_{\boldsymbol{\theta}}\, \tilde{\boldsymbol{u}} = \tilde{\boldsymbol{b}}$, where $\tilde{\boldsymbol{b}} := \boldsymbol{b}/(\|\boldsymbol{b}\| + \epsilon)$, until $\|\tilde{\boldsymbol{r}}\| := \|\tilde{\boldsymbol{b}} - \mathbf{H}_{\boldsymbol{\theta}}\, \tilde{\boldsymbol{u}}\| \leq \tau$ and then returning $\boldsymbol{u} := (\|\boldsymbol{b}\| + \epsilon)\, \tilde{\boldsymbol{u}}$, where epsilon is set to a small constant value to prevent division by zero. Since we are solving batches of systems of linear equations of the form $\mathbf{H}_{\boldsymbol{\theta}}\, [\, \boldsymbol{v_y}, \boldsymbol{v}_1, \ldots, \boldsymbol{v}_s\,] = [\, \boldsymbol{y}, \boldsymbol{z}_1, \ldots, \boldsymbol{z}_s\,]$, we track the residuals of each individual system and calculate separate residual norms for the mean and for the probe vectors, where the residual norm for the mean $\|\boldsymbol{r_y}\|$ corresponds to the system $\mathbf{H}_{\boldsymbol{\theta}}\, \boldsymbol{v_y} = \boldsymbol{y}$ and the residual norm for the probe vectors $\|\boldsymbol{r_z}\|$ is defined as the arithmetic average over residual norms corresponding to the systems $\mathbf{H}_{\boldsymbol{\theta}}\, [\, \boldsymbol{v}_1, \ldots, \boldsymbol{v}_s\,] = [\, \boldsymbol{z}_1, \ldots, \boldsymbol{z}_s\,]$. Both relative residual norms must reach the tolerance $\tau$ to satisfy the termination criterion. We use separate residual norms because $\|\boldsymbol{r_y}\|$ typically converges faster than $\|\boldsymbol{r_z}\|$, such that an average other all systems tends to dilute the latter (see Figures 14 to 17).

**Conjugate Gradients**  The conjugate gradients algorithm [9, 30] computes necessary residuals as part of the algorithm. In terms of counting solver epochs, every conjugate gradient iteration counts as one solver epoch because in every iteration each value of $\mathbf{H}_{\boldsymbol{\theta}}$ is computed once. Following previous work, we used a pivoted Cholesky preconditioner of rank 100 for all experiments [30]. We initialised conjugate gradients either at zero (no warm start) or at the previous solution (warm start). Otherwise, conjugate gradients does not have any other parameters. Pseudocode is provided in Algorithm 1.

---

**Algorithm 1** Conjugate gradients for solving $\mathbf{H}_{\boldsymbol{\theta}}\, [\, \boldsymbol{v_y}, \boldsymbol{v}_1, \ldots, \boldsymbol{v}_s\,] = [\, \boldsymbol{y}, \boldsymbol{z}_1, \ldots, \boldsymbol{z}_s\,]$

**Require:** Linear operator $\mathbf{H}_{\boldsymbol{\theta}}(\cdot)$, targets $\boldsymbol{b} = [\, \boldsymbol{y}, \boldsymbol{z}_1, \ldots, \boldsymbol{z}_s\,]$, tolerance $\tau$, maximum epochs $T$
**Require:** Preconditioner $\mathbf{P}(\cdot)$
1: Let $(\cdot)_*$ denote parallel execution over $(\cdot)_{\boldsymbol{y}}, (\cdot)_1, \ldots, (\cdot)_s$
2: $\boldsymbol{v}_* \leftarrow \mathbf{0}$ (or previous solution if warm start)
3: $\boldsymbol{r}_* \leftarrow \boldsymbol{b}_* - \mathbf{H}_{\boldsymbol{\theta}}(\boldsymbol{v}_*)$
4: $\boldsymbol{p}_* \leftarrow \mathbf{P}(\boldsymbol{r}_*)$
5: $\boldsymbol{d}_* \leftarrow \boldsymbol{p}_*$
6: $\gamma_* \leftarrow \boldsymbol{r}_*^{\mathsf{T}} \boldsymbol{p}_*$
7: $t \leftarrow 0$
8: **while** $t < T$ **and** $\|\boldsymbol{r_y}\| > \tau$ **and** $\frac{1}{s}\sum_{j=1}^{s} \|\boldsymbol{r}_j\| = \|\boldsymbol{r_z}\| > \tau$ **do**
9:     $\alpha_* \leftarrow \gamma_*/\boldsymbol{d}_*^{\mathsf{T}} \mathbf{H}_{\boldsymbol{\theta}}(\boldsymbol{d}_*)$
10:     $\boldsymbol{v}_* \leftarrow \boldsymbol{v}_* + \alpha_* \boldsymbol{d}_*$
11:     $\boldsymbol{r}_* \leftarrow \boldsymbol{r}_* - \alpha_* \mathbf{H}_{\boldsymbol{\theta}}(\boldsymbol{d}_*)$
12:     $\boldsymbol{p}_* \leftarrow \mathbf{P}(\boldsymbol{r}_*)$
13:     $\beta_* \leftarrow \boldsymbol{r}_*^{\mathsf{T}} \boldsymbol{p}_*/\gamma_*$
14:     $\gamma_* \leftarrow \boldsymbol{r}_*^{\mathsf{T}} \boldsymbol{p}_*$
15:     $\boldsymbol{d}_* \leftarrow \boldsymbol{p}_* + \beta_* \boldsymbol{d}_*$
16:     $t \leftarrow t + 1$
17: **end while**
18: **return** $[\, \boldsymbol{v_y}, \boldsymbol{v}_1, \ldots, \boldsymbol{v}_s\,]$

---

**Alternating Projections**  The alternating projections algorithm [33] also keeps track of the residuals as part of the algorithm. In terms of counting solver epochs, we convert the number of maximum solver epochs to a maximum number of solver iterations by multiplying with $n/b$, where $b$ is the block size, because every iteration of alternating projections computes $b/n$ of all entries of $\mathbf{H}_{\boldsymbol{\theta}}$. We used a block size of $b = 1000$ for all datasets, except PROTEIN and KEGGDIRECTED, where we used $b = 2000$ instead. We initialised alternating projections either at zero (no warm start) or at the previous solution (warm start). During each marginal likelihood step, the Cholesky factorisation of every block is computed once and cached afterwards (although, in practice, the Cholesky factorisation does not dominate the computational costs). In each iteration of alternating projections, the block with largest residual norm is selected to be processed. Pseudocode is provided in Algorithm 2.

---

**Algorithm 2** Alternating projections for solving $\mathbf{H}_{\boldsymbol{\theta}}\,[\,\boldsymbol{v_y}, \boldsymbol{v}_1, \ldots, \boldsymbol{v}_s\,] = [\,\boldsymbol{y}, \boldsymbol{z}_1, \ldots, \boldsymbol{z}_s\,]$

---

**Require:** Linear operator $\mathbf{H}_{\boldsymbol{\theta}}(\cdot)$, targets $\boldsymbol{b} = [\,\boldsymbol{y}, \boldsymbol{z}_1, \ldots, \boldsymbol{z}_s\,]$, tolerance $\tau$, maximum epochs $T$
**Require:** Block size $b$, block partitions $[1], [2], \ldots, [\lceil \frac{n}{b} \rceil]$
  1: Let $(\cdot)_*$ denote parallel execution over $(\cdot)_{\boldsymbol{y}}, (\cdot)_1, \ldots, (\cdot)_s$
  2: $\boldsymbol{v}_* \leftarrow \mathbf{0}$ (or previous solution if warm start)
  3: $\boldsymbol{r}_* \leftarrow \boldsymbol{b}_* - \mathbf{H}_{\boldsymbol{\theta}}(\boldsymbol{v}_*)$
  4: $t \leftarrow 0$
  5: **while** $t < \frac{n}{b}T$ **and** $\|\boldsymbol{r_y}\| > \tau$ **and** $\frac{1}{s}\sum_{j=1}^{s}\|\boldsymbol{r}_j\| = \|\boldsymbol{r_z}\| > \tau$ **do**
  6: $\quad [i] \leftarrow \texttt{arg\_max}(\|\boldsymbol{r_y}[1] + \sum_{j=1}^{s}\boldsymbol{r}_j[1]\|, \ldots, \|\boldsymbol{r_y}[\lceil \frac{n}{b} \rceil] + \sum_{j=1}^{s}\boldsymbol{r}_j[\lceil \frac{n}{b} \rceil]\|)$
  7: $\quad \boldsymbol{v}_*[i] \leftarrow \boldsymbol{v}_*[i] + \texttt{chol\_solve}(\mathbf{H}_{\boldsymbol{\theta}}[i,i], \boldsymbol{r}_*[i])$
  8: $\quad \boldsymbol{r}_* \leftarrow \boldsymbol{r}_* - \mathbf{H}_{\boldsymbol{\theta}}[:,i](\texttt{chol\_solve}(\mathbf{H}_{\boldsymbol{\theta}}[i,i], \boldsymbol{r}_*[i]))$
  9: $\quad t \leftarrow t + 1$
10: **end while**
11: **return** $[\,\boldsymbol{v_y}, \boldsymbol{v}_1, \ldots, \boldsymbol{v}_s\,]$

---

**Stochastic Gradient Descent**  The stochastic gradient descent algorithm [15, 16] does not compute residuals as part of the algorithm. Therefore, we estimate the current residual by keeping a residual vector in memory and updating it sparsely whenever we compute the gradient on a batch of data, leveraging the property that the negative gradient is equal to the residual. In practice, we find that this estimates an approximate upper bound on the true residual, which becomes fairly accurate after a few iterations. In terms of counting solver epochs, we apply the same procedure as for alternating projections. The number of maximum solver epochs is converted to a maximum number of solver iterations by multiplying with $n/b$, where $b$ is the batch size. We used a batch size of $b = 500$, momentum of $\rho = 0.9$, and no Polyak averaging, because averaging is not strictly necessary [16] and would interfere with our residual estimation heuristic. We use learning rates of 30, 20, 30, 20, and 20 respectively for the POL, ELEVATORS, BIKE, KEGGDIRECTED and PROTEIN datasets, picking the largest learning rate from a grid $[5, 10, 20, 30, 50, 60, 70, 80, 90, 100]$ that does not cause the inner linear system solver to diverge on the very first outer marginal likelihood loop. For the larger datasets, we use learning rates of 10, 10, 50, and 50 for 3DROAD, BUZZ, SONG and HOUSEELECTRIC, picking half of the largest learning rate as above. We find that the larger datasets are more sensitive to diverging when the hyperparameters change, and therefore we choose half of the largest learning rate possible at initialisation. Pseudocode is provided in Algorithm 3.

---

**Algorithm 3** Stochastic gradient descent for solving $\mathbf{H}_{\boldsymbol{\theta}}\,[\,\boldsymbol{v_y}, \boldsymbol{v}_1, \ldots, \boldsymbol{v}_s\,] = [\,\boldsymbol{y}, \boldsymbol{z}_1, \ldots, \boldsymbol{z}_s\,]$

---

**Require:** Linear operator $\mathbf{H}_{\boldsymbol{\theta}}(\cdot)$, targets $\boldsymbol{b} = [\,\boldsymbol{y}, \boldsymbol{z}_1, \ldots, \boldsymbol{z}_s\,]$, tolerance $\tau$, maximum epochs $T$
**Require:** Batch size $b$, learning rate $\gamma$, momentum $\rho$
  1: Let $(\cdot)_*$ denote parallel execution over $(\cdot)_{\boldsymbol{y}}, (\cdot)_1, \ldots, (\cdot)_s$
  2: $\boldsymbol{v}_* \leftarrow \mathbf{0}$ (or previous solution if warm start)
  3: $\boldsymbol{r}_* \leftarrow \boldsymbol{b}_*$
  4: $\boldsymbol{m}_* \leftarrow \mathbf{0}$
  5: $t \leftarrow 0$
  6: **while** $t < \frac{n}{b}T$ **and** $\|\boldsymbol{r_y}\| > \tau$ **and** $\frac{1}{s}\sum_{j=1}^{s}\|\boldsymbol{r}_j\| = \|\boldsymbol{r_z}\| > \tau$ **do**
  7: $\quad [i] \leftarrow \texttt{uniform\_batch\_sample}(1, \ldots, n)$
  8: $\quad \boldsymbol{g}_* \leftarrow \mathbf{0}$
  9: $\quad \boldsymbol{g}_*[i] \leftarrow \mathbf{H}_{\boldsymbol{\theta}}[i,:](\boldsymbol{v}_*) - \boldsymbol{b}_*[i]$
10: $\quad \boldsymbol{m}_* \leftarrow \rho\,\boldsymbol{m}_* - \frac{\gamma}{b}\,\boldsymbol{g}_*$
11: $\quad \boldsymbol{v}_* = \boldsymbol{v}_* + \boldsymbol{m}_*$
12: $\quad \boldsymbol{r}_*[i] \leftarrow \boldsymbol{g}_*[i]$
13: $\quad t \leftarrow t + 1$
14: **end while**
15: **return** $[\,\boldsymbol{v_y}, \boldsymbol{v}_1, \ldots, \boldsymbol{v}_s\,]$

---

## C  Additional Empirical Results

In this appendix, we provide additional result from our experiments.

For our first experiment (solving until reaching the relative residual norm tolerance), we present the predictive performance, time taken, and speed-ups in Tables 2 to 6. Hyperparameter trajectories are illustrated in Figures 11 to 13. Required number of solver iterations are shown in Figure 21.

For our second experiment (solving until reaching the tolerance or exhausting the compute budget), we tabulate the predictive performance, time taken, and average residual norms in Tables 7 to 10. The behaviour of residual norms is visualised in Figures 14 to 17. The evolution of residual norms and predictive performance via pathwise conditioning is depicted in Figures 18 to 20.

Table 2: Results on POL when solving until convergence (mean $\pm$ standard error over 10 splits).

| | path wise | warm start | Test RMSE | Test LLH | Total Time (min) | Solver Time (min) | Speed-Up |
|---|---|---|---|---|---|---|---|
| | | | \multicolumn{4}{c}{POL ($n = 13\,500$, $d = 26$)} | | |
| CG | | | $0.0750 \pm 0.0010$ | $1.2682 \pm 0.0084$ | $4.8263 \pm 0.0356$ | $4.6138 \pm 0.0355$ | — |
| CG | ✓ | | $0.0754 \pm 0.0010$ | $1.2716 \pm 0.0077$ | $3.9567 \pm 0.0204$ | $3.7148 \pm 0.0205$ | $1.2 \times$ |
| CG | | ✓ | $0.0750 \pm 0.0010$ | $1.2681 \pm 0.0084$ | $2.2844 \pm 0.0131$ | $2.0755 \pm 0.0131$ | $2.1 \times$ |
| CG | ✓ | ✓ | $0.0758 \pm 0.0010$ | $1.2666 \pm 0.0074$ | $2.4652 \pm 0.0133$ | $2.2513 \pm 0.0133$ | $2.0 \times$ |
| AP | | | $0.0750 \pm 0.0010$ | $1.2682 \pm 0.0084$ | $493.05 \pm 3.0738$ | $492.84 \pm 3.0737$ | — |
| AP | ✓ | | $0.0754 \pm 0.0010$ | $1.2715 \pm 0.0077$ | $44.006 \pm 0.0722$ | $43.765 \pm 0.0725$ | $11.2 \times$ |
| AP | | ✓ | $0.0750 \pm 0.0010$ | $1.2681 \pm 0.0084$ | $27.915 \pm 0.2707$ | $27.706 \pm 0.2708$ | $17.7 \times$ |
| AP | ✓ | ✓ | $0.0758 \pm 0.0010$ | $1.2666 \pm 0.0074$ | $3.8962 \pm 0.0174$ | $3.6831 \pm 0.0173$ | $126.6 \times$ |
| SGD | | | $0.0750 \pm 0.0010$ | $1.2681 \pm 0.0084$ | $138.63 \pm 0.8169$ | $138.60 \pm 0.8170$ | — |
| SGD | ✓ | | $0.0754 \pm 0.0010$ | $1.2708 \pm 0.0074$ | $73.546 \pm 0.2661$ | $73.434 \pm 0.2661$ | $1.9 \times$ |
| SGD | | ✓ | $0.0750 \pm 0.0010$ | $1.2682 \pm 0.0084$ | $26.483 \pm 0.1002$ | $26.461 \pm 0.1001$ | $5.2 \times$ |
| SGD | ✓ | ✓ | $0.0757 \pm 0.0010$ | $1.2678 \pm 0.0076$ | $17.938 \pm 0.1083$ | $17.854 \pm 0.1083$ | $7.7 \times$ |

Table 3: Results on ELEV when solving until convergence (mean $\pm$ standard error over 10 splits).

| | path wise | warm start | Test RMSE | Test LLH | Total Time (min) | Solver Time (min) | Speed-Up |
|---|---|---|---|---|---|---|---|
| | | | \multicolumn{4}{c}{ELEVATORS ($n = 14\,940$, $d = 18$)} | | |
| CG | | | $0.3550 \pm 0.0034$ | $-0.3856 \pm 0.0065$ | $1.5811 \pm 0.0063$ | $1.3661 \pm 0.0062$ | — |
| CG | ✓ | | $0.3562 \pm 0.0033$ | $-0.3868 \pm 0.0065$ | $1.4940 \pm 0.0039$ | $1.2484 \pm 0.0039$ | $1.1 \times$ |
| CG | | ✓ | $0.3550 \pm 0.0034$ | $-0.3856 \pm 0.0065$ | $1.0308 \pm 0.0037$ | $0.8191 \pm 0.0036$ | $1.5 \times$ |
| CG | ✓ | ✓ | $0.3558 \pm 0.0034$ | $-0.3856 \pm 0.0066$ | $0.9977 \pm 0.0029$ | $0.7835 \pm 0.0028$ | $1.6 \times$ |
| AP | | | $0.3550 \pm 0.0034$ | $-0.3856 \pm 0.0065$ | $77.787 \pm 0.2694$ | $77.572 \pm 0.2693$ | — |
| AP | ✓ | | $0.3562 \pm 0.0033$ | $-0.3868 \pm 0.0065$ | $36.346 \pm 0.0821$ | $36.100 \pm 0.0820$ | $2.1 \times$ |
| AP | | ✓ | $0.3550 \pm 0.0034$ | $-0.3856 \pm 0.0065$ | $1.6658 \pm 0.0081$ | $1.4541 \pm 0.0081$ | $46.7 \times$ |
| AP | ✓ | ✓ | $0.3558 \pm 0.0034$ | $-0.3856 \pm 0.0066$ | $1.2065 \pm 0.0032$ | $0.9928 \pm 0.0031$ | $64.5 \times$ |
| SGD | | | $0.3550 \pm 0.0034$ | $-0.3855 \pm 0.0066$ | $5.5408 \pm 0.0101$ | $5.4091 \pm 0.0101$ | — |
| SGD | ✓ | | $0.3562 \pm 0.0033$ | $-0.3868 \pm 0.0065$ | $4.5758 \pm 0.0086$ | $4.4146 \pm 0.0086$ | $1.2 \times$ |
| SGD | | ✓ | $0.3550 \pm 0.0034$ | $-0.3855 \pm 0.0066$ | $1.2221 \pm 0.0042$ | $1.0927 \pm 0.0041$ | $4.5 \times$ |
| SGD | ✓ | ✓ | $0.3558 \pm 0.0034$ | $-0.3854 \pm 0.0068$ | $1.1437 \pm 0.0033$ | $1.0105 \pm 0.0032$ | $4.8 \times$ |

Table 4: Results on BIKE when solving until convergence (mean ± standard error over 10 splits).

| | path wise | warm start | Test RMSE | Test LLH | Total Time (min) | Solver Time (min) | Speed-Up |
|---|---|---|---|---|---|---|---|
| | | | | | BIKE ($n = 15\,642$, $d = 17$) | | |
| CG | | | $0.0326 \pm 0.0031$ | $2.1500 \pm 0.0180$ | $5.0797 \pm 0.0280$ | $4.8412 \pm 0.0280$ | — |
| | ✓ | | $0.0326 \pm 0.0030$ | $2.0674 \pm 0.0167$ | $4.4147 \pm 0.0225$ | $4.1410 \pm 0.0221$ | $1.2 \times$ |
| | | ✓ | $0.0327 \pm 0.0031$ | $2.1508 \pm 0.0181$ | $2.7392 \pm 0.0137$ | $2.5039 \pm 0.0135$ | $1.9 \times$ |
| | ✓ | ✓ | $0.0329 \pm 0.0030$ | $2.0615 \pm 0.0151$ | $3.0658 \pm 0.0168$ | $2.8268 \pm 0.0167$ | $1.7 \times$ |
| AP | | | $0.0326 \pm 0.0031$ | $2.1504 \pm 0.0180$ | $302.26 \pm 1.7735$ | $302.03 \pm 1.7735$ | — |
| | ✓ | | $0.0325 \pm 0.0030$ | $2.0668 \pm 0.0167$ | $35.081 \pm 0.0773$ | $34.811 \pm 0.0772$ | $8.6 \times$ |
| | | ✓ | $0.0326 \pm 0.0031$ | $2.1503 \pm 0.0181$ | $19.892 \pm 0.2146$ | $19.657 \pm 0.2146$ | $15.2 \times$ |
| | ✓ | ✓ | $0.0330 \pm 0.0030$ | $2.0616 \pm 0.0150$ | $5.4041 \pm 0.0394$ | $5.1653 \pm 0.0393$ | $56.0 \times$ |
| SGD | | | $0.0326 \pm 0.0031$ | $2.1535 \pm 0.0181$ | $412.17 \pm 10.460$ | $412.03 \pm 10.450$ | — |
| | ✓ | | $0.0324 \pm 0.0030$ | $2.0692 \pm 0.0174$ | $156.24 \pm 2.2113$ | $156.05 \pm 2.2109$ | $2.6 \times$ |
| | | ✓ | $0.0327 \pm 0.0031$ | $2.1524 \pm 0.0179$ | $74.341 \pm 1.2532$ | $74.184 \pm 1.2524$ | $5.5 \times$ |
| | ✓ | ✓ | $0.0332 \pm 0.0030$ | $2.0562 \pm 0.0144$ | $64.145 \pm 1.1086$ | $63.983 \pm 1.1083$ | $6.4 \times$ |

Table 5: Results on PROT when solving until convergence (mean ± standard error over 10 splits).

| | path wise | warm start | Test RMSE | Test LLH | Total Time (min) | Solver Time (min) | Speed-Up |
|---|---|---|---|---|---|---|---|
| | | | | | PROTEIN ($n = 41\,157$, $d = 9$) | | |
| CG | | | $0.5024 \pm 0.0036$ | $-0.5871 \pm 0.0096$ | $29.849 \pm 0.2463$ | $28.208 \pm 0.2463$ | — |
| | ✓ | | $0.4909 \pm 0.0035$ | $-0.6210 \pm 0.0082$ | $19.984 \pm 0.1429$ | $18.317 \pm 0.1430$ | $1.5 \times$ |
| | | ✓ | $0.5026 \pm 0.0036$ | $-0.5871 \pm 0.0096$ | $11.542 \pm 0.1058$ | $9.8816 \pm 0.1059$ | $2.6 \times$ |
| | ✓ | ✓ | $0.4912 \pm 0.0034$ | $-0.6214 \pm 0.0079$ | $13.744 \pm 0.0959$ | $12.085 \pm 0.0956$ | $2.2 \times$ |
| AP | | | $0.5024 \pm 0.0036$ | $-0.5871 \pm 0.0096$ | $130.93 \pm 1.0450$ | $129.29 \pm 1.0451$ | — |
| | ✓ | | $0.4907 \pm 0.0035$ | $-0.6214 \pm 0.0082$ | $55.790 \pm 0.2054$ | $54.125 \pm 0.2053$ | $2.3 \times$ |
| | | ✓ | $0.5027 \pm 0.0036$ | $-0.5871 \pm 0.0097$ | $16.425 \pm 0.1816$ | $14.765 \pm 0.1818$ | $8.0 \times$ |
| | ✓ | ✓ | $0.4912 \pm 0.0034$ | $-0.6213 \pm 0.0079$ | $12.341 \pm 0.0347$ | $10.682 \pm 0.0346$ | $10.6 \times$ |
| SGD | | | $0.5026 \pm 0.0036$ | $-0.5871 \pm 0.0096$ | $75.205 \pm 1.4358$ | $72.932 \pm 1.4357$ | — |
| | ✓ | | $0.4894 \pm 0.0035$ | $-0.6268 \pm 0.0077$ | $24.030 \pm 0.1364$ | $21.726 \pm 0.1364$ | $3.1 \times$ |
| | | ✓ | $0.5027 \pm 0.0037$ | $-0.5878 \pm 0.0096$ | $11.226 \pm 0.1430$ | $8.9541 \pm 0.1433$ | $6.7 \times$ |
| | ✓ | ✓ | $0.4911 \pm 0.0034$ | $-0.6217 \pm 0.0078$ | $11.939 \pm 0.0700$ | $9.6617 \pm 0.0700$ | $6.3 \times$ |

Table 6: Results on KEGG when solving until convergence (mean ± standard error over 10 splits).

| | path wise | warm start | Test RMSE | Test LLH | Total Time (min) | Solver Time (min) | Speed-Up |
|---|---|---|---|---|---|---|---|
| | | | | | KEGGDIRECTED ($n = 43\,945$, $d = 20$) | | |
| CG | | | $0.0837 \pm 0.0016$ | $1.0818 \pm 0.0170$ | $27.974 \pm 0.3172$ | $25.543 \pm 0.3120$ | — |
| | ✓ | | $0.0837 \pm 0.0016$ | $1.0818 \pm 0.0169$ | $26.362 \pm 0.2851$ | $23.897 \pm 0.2804$ | $1.1 \times$ |
| | | ✓ | $0.0837 \pm 0.0016$ | $1.0816 \pm 0.0171$ | $12.754 \pm 0.1314$ | $10.326 \pm 0.1266$ | $2.2 \times$ |
| | ✓ | ✓ | $0.0836 \pm 0.0016$ | $1.0819 \pm 0.0166$ | $12.998 \pm 0.1383$ | $10.559 \pm 0.1330$ | $2.2 \times$ |
| AP | | | — | — | $> 24\,\text{h}$ | $> 24\,\text{h}$ | — |
| | ✓ | | $0.0837 \pm 0.0016$ | $1.0820 \pm 0.0166$ | $491.41 \pm 0.4624$ | $488.95 \pm 0.4657$ | $> 2.9 \times$ |
| | | ✓ | $0.0837 \pm 0.0016$ | $1.0818 \pm 0.0172$ | $211.28 \pm 1.9504$ | $208.85 \pm 1.9556$ | $> 6.8 \times$ |
| | ✓ | ✓ | $0.0836 \pm 0.0016$ | $1.0817 \pm 0.0166$ | $14.013 \pm 0.0768$ | $11.574 \pm 0.0711$ | $> 102.8 \times$ |
| SGD | | | $0.0837 \pm 0.0016$ | $1.0816 \pm 0.0173$ | $620.07 \pm 6.3224$ | $617.35 \pm 6.3194$ | — |
| | ✓ | | $0.0837 \pm 0.0016$ | $1.0822 \pm 0.0164$ | $411.58 \pm 6.1065$ | $408.86 \pm 6.0874$ | $1.5 \times$ |
| | | ✓ | $0.0837 \pm 0.0016$ | $1.0821 \pm 0.0170$ | $168.38 \pm 1.6669$ | $165.66 \pm 1.6636$ | $3.7 \times$ |
| | ✓ | ✓ | $0.0839 \pm 0.0016$ | $1.0725 \pm 0.0136$ | $58.679 \pm 0.5457$ | $55.952 \pm 0.5418$ | $10.6 \times$ |

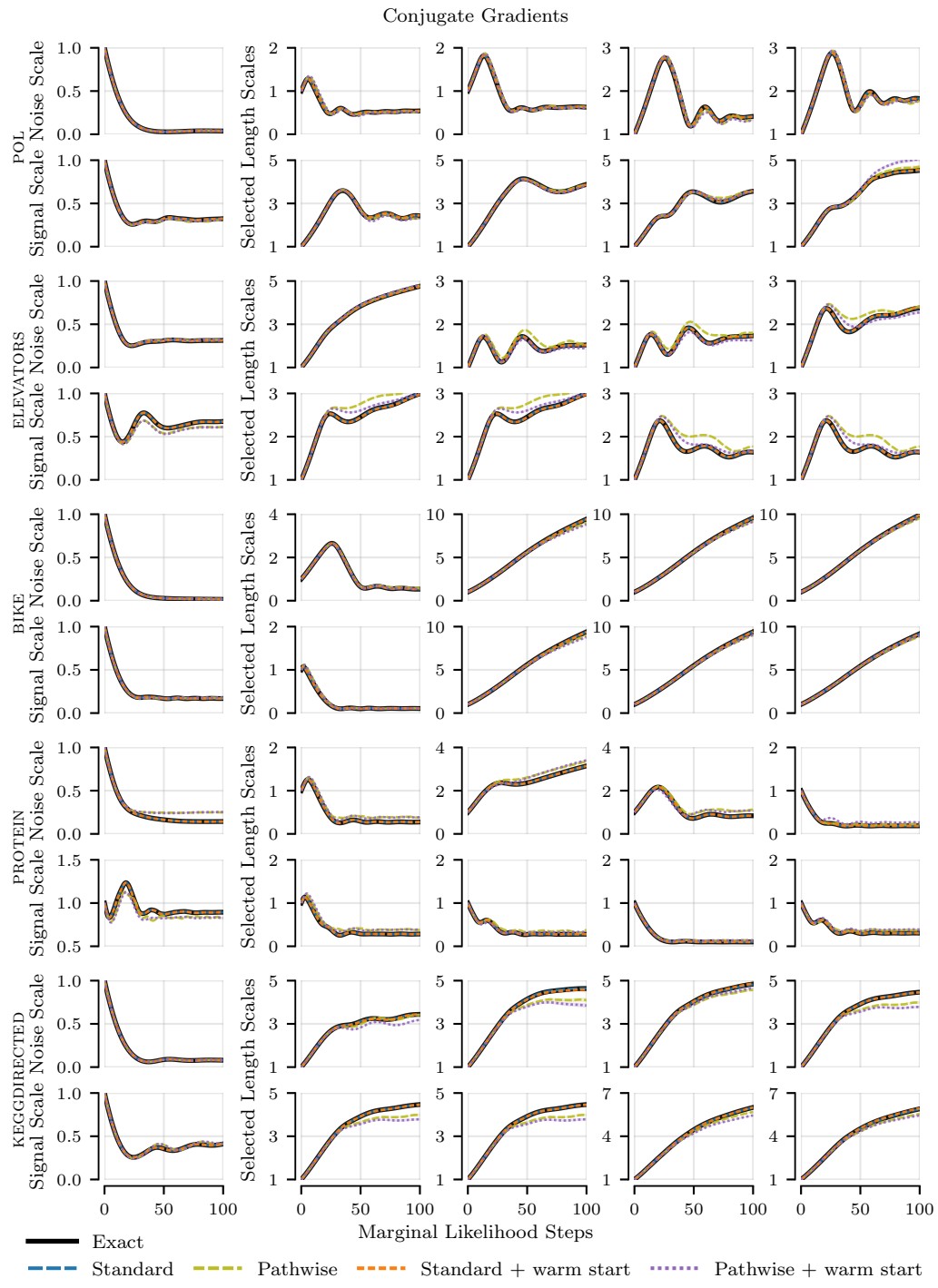

Figure 11: Evolution of hyperparameters during marginal likelihood optimisation on different datasets using conjugate gradients as linear system solver. Most of the time, the behaviour of exact gradient computation using Cholesky factorisation is resembled.

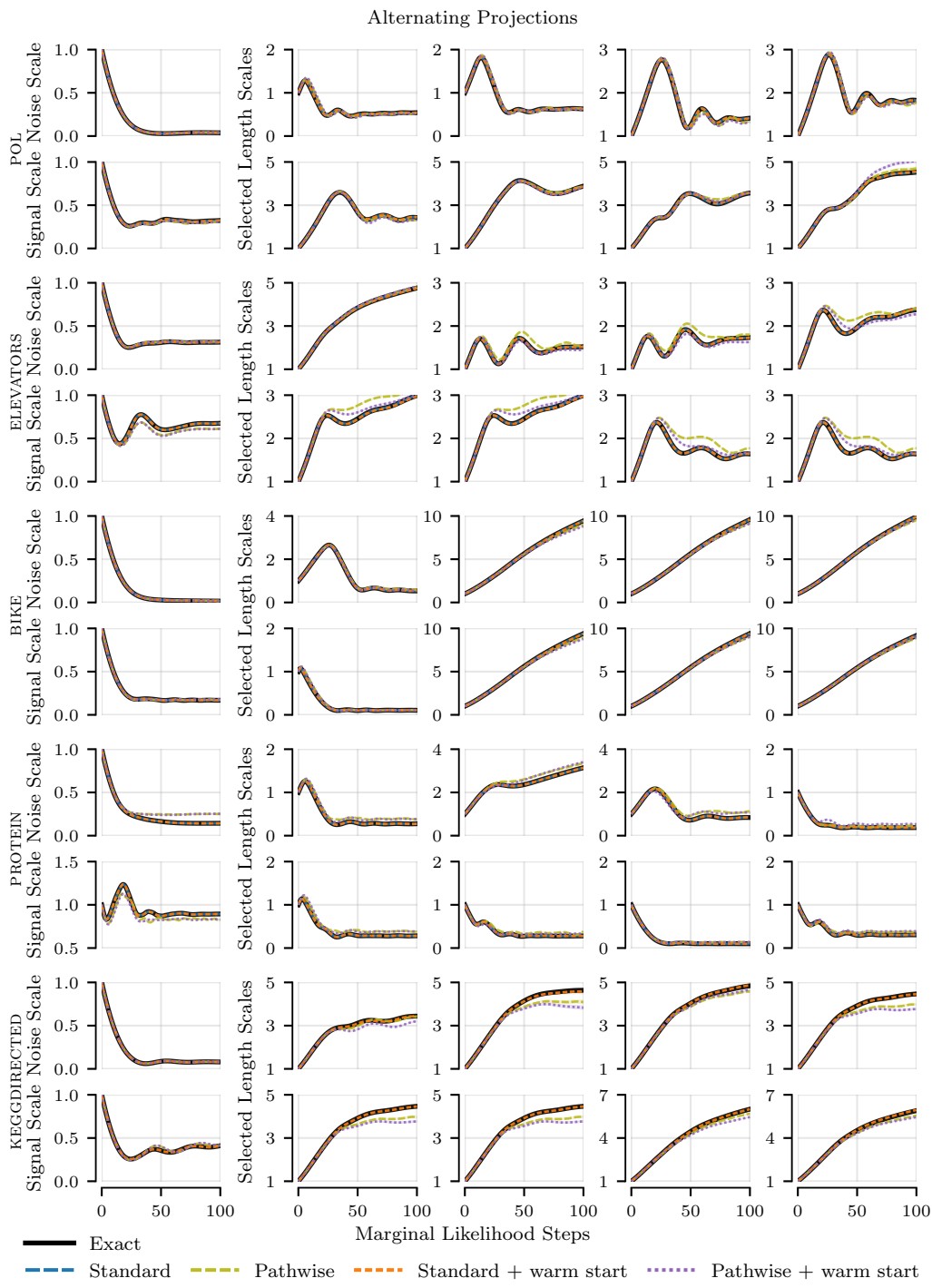

Figure 12: Evolution of hyperparameters during marginal likelihood optimisation on different datasets using alternating projections as linear system solver. Most of the time, the behaviour of exact gradient computation using Cholesky factorisation is resembled.

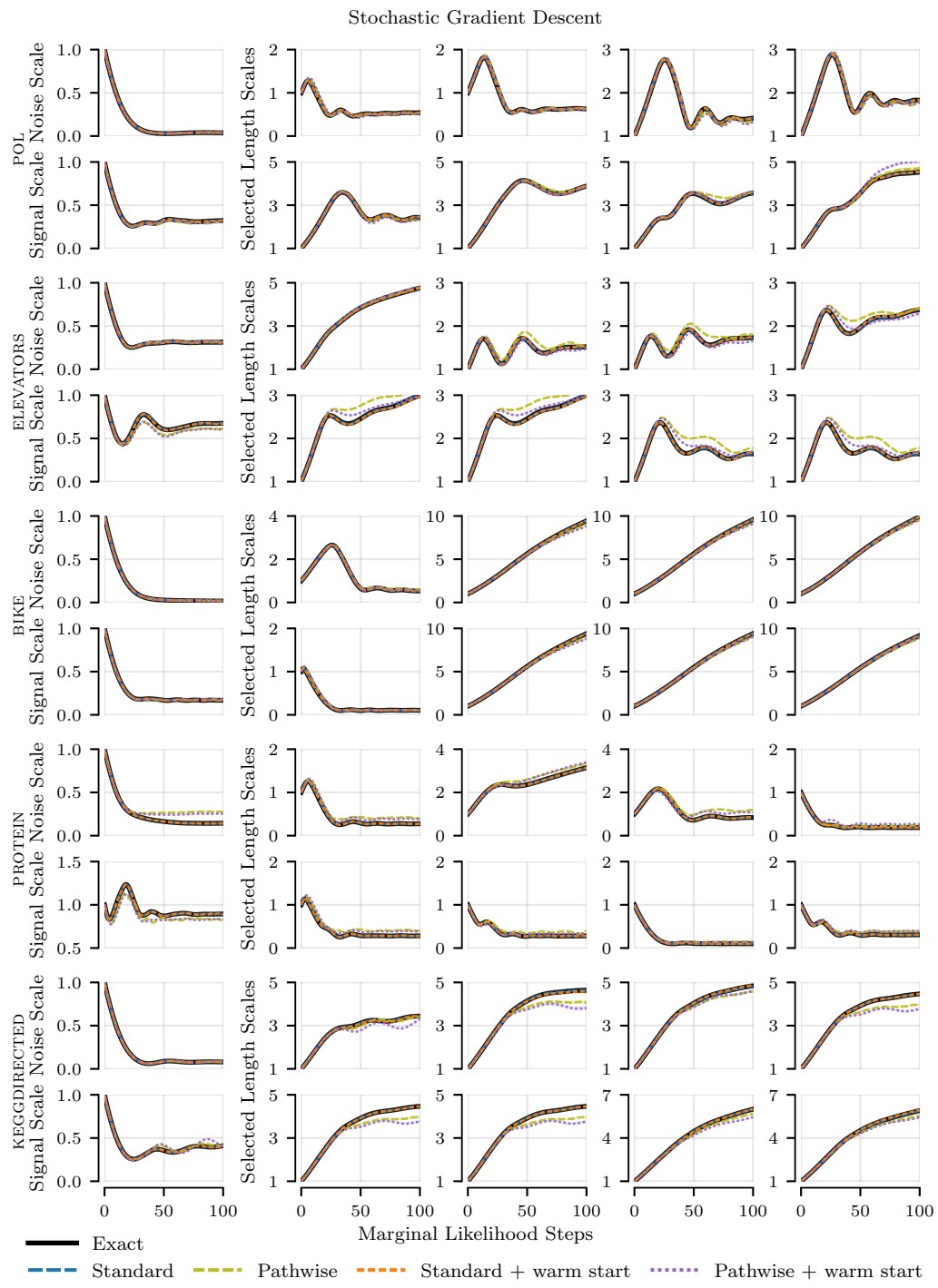

Figure 13: Evolution of hyperparameters during marginal likelihood optimisation on different datasets using stochastic gradient descent as linear system solver. Most of the time, the behaviour of exact gradient computation using Cholesky factorisation is resembled.

Table 7: Results on 3DROAD with 10 maximum solver epochs (mean ± standard error over 10 splits).

| | warm start | 3DROAD ($n = 391\,387, d = 3$) | | | Average Residual Norm | |
|---|---|---|---|---|---|---|
| | | Test RMSE | Test LLH | Total Time (h) | of Mean | of Probe Vectors |
| CG | | $1.2177 \pm 0.0296$ | $-1.5463 \pm 0.0187$ | $1.9308 \pm 0.0073$ | $0.9815 \pm 0.0217$ | $0.9174 \pm 0.0106$ |
| CG | ✓ | $0.5371 \pm 0.0598$ | $-0.9143 \pm 0.0336$ | $1.9411 \pm 0.0047$ | $0.6530 \pm 0.0152$ | $0.9003 \pm 0.0077$ |
| AP | | $0.1042 \pm 0.0017$ | $0.8237 \pm 0.0172$ | $2.1782 \pm 0.0059$ | $0.0950 \pm 0.0017$ | $0.0541 \pm 0.0010$ |
| AP | ✓ | $0.0563 \pm 0.0009$ | $0.9309 \pm 0.0186$ | $2.1805 \pm 0.0047$ | $0.0469 \pm 0.0010$ | $0.0651 \pm 0.0010$ |
| SGD | | $0.1430 \pm 0.0014$ | $-0.4662 \pm 0.0399$ | $1.2717 \pm 0.0014$ | $0.1130 \pm 0.0017$ | $0.0633 \pm 0.0020$ |
| SGD | ✓ | $0.0654 \pm 0.0010$ | $0.9276 \pm 0.0126$ | $1.2772 \pm 0.0031$ | $0.0561 \pm 0.0010$ | $0.0797 \pm 0.0022$ |

Table 8: Results on SONG with 10 maximum solver epochs (mean ± standard error over 10 splits).

| | warm start | SONG ($n = 463\,811, d = 90$) | | | Average Residual Norm | |
|---|---|---|---|---|---|---|
| | | Test RMSE | Test LLH | Total Time (h) | of Mean | of Probe Vectors |
| CG | | $2.1573 \pm 0.0672$ | $-2.0688 \pm 0.0303$ | $18.107 \pm 0.0602$ | $1.6528 \pm 0.0773$ | $1.6356 \pm 0.0716$ |
| CG | ✓ | $0.8698 \pm 0.0091$ | $-1.3025 \pm 0.0140$ | $17.879 \pm 0.1979$ | $0.3793 \pm 0.0171$ | $0.4147 \pm 0.0186$ |
| AP | | $0.7428 \pm 0.0019$ | $-1.1197 \pm 0.0024$ | $18.256 \pm 0.0272$ | $0.0421 \pm 0.0016$ | $0.0499 \pm 0.0011$ |
| AP | ✓ | $0.7420 \pm 0.0019$ | $-1.1184 \pm 0.0023$ | $15.114 \pm 0.2373$ | $0.0085 \pm 0.0004$ | $0.0125 \pm 0.0002$ |
| SGD | | $0.7426 \pm 0.0019$ | $-1.1205 \pm 0.0024$ | $17.160 \pm 0.0669$ | $0.0688 \pm 0.0017$ | $0.0834 \pm 0.0014$ |
| SGD | ✓ | $0.7419 \pm 0.0019$ | $-1.1184 \pm 0.0023$ | $16.756 \pm 0.0842$ | $0.0100 \pm 0.0003$ | $0.0117 \pm 0.0001$ |

Table 9: Results on BUZZ with 10 maximum solver epochs (mean ± standard error over 10 splits).

| | warm start | BUZZ ($n = 524\,925, d = 77$) | | | Average Residual Norm | |
|---|---|---|---|---|---|---|
| | | Test RMSE | Test LLH | Total Time (h) | of Mean | of Probe Vectors |
| CG | | $2.0042 \pm 0.0382$ | $-1.8249 \pm 0.0393$ | $22.012 \pm 0.0047$ | $1.9248 \pm 0.0340$ | $3.0380 \pm 0.1826$ |
| CG | ✓ | $4.5317 \pm 0.4186$ | $-11.973 \pm 1.4876$ | $21.725 \pm 0.2588$ | $2.1218 \pm 0.1021$ | $1.6968 \pm 0.0424$ |
| AP | | $0.2770 \pm 0.0030$ | $-0.0483 \pm 0.0039$ | $22.466 \pm 0.0048$ | $0.0516 \pm 0.0017$ | $0.0851 \pm 0.0009$ |
| AP | ✓ | $0.2743 \pm 0.0030$ | $-0.0366 \pm 0.0034$ | $22.470 \pm 0.0056$ | $0.0143 \pm 0.0004$ | $0.0263 \pm 0.0001$ |
| SGD | | $0.2851 \pm 0.0035$ | $-0.1029 \pm 0.0152$ | $16.722 \pm 0.0012$ | $0.2252 \pm 0.0004$ | $0.3906 \pm 0.0115$ |
| SGD | ✓ | $0.2735 \pm 0.0030$ | $-0.0457 \pm 0.0045$ | $16.702 \pm 0.0016$ | $0.0767 \pm 0.0022$ | $0.1544 \pm 0.0039$ |

Table 10: Results on HOUSE with 10 maximum solver epochs (mean ± standard error over 10 splits).

| | warm start | HOUSEELECTRIC ($n = 1\,844\,352, d = 11$) | | | Average Residual Norm | |
|---|---|---|---|---|---|---|
| | | Test RMSE | Test LLH | Total Time (h) | of Mean | of Probe Vectors |
| CG | | $0.8328 \pm 0.0215$ | $-4.0695 \pm 0.2445$ | $32.368 \pm 0.3881$ | $0.7495 \pm 0.0224$ | $1.4922 \pm 0.0826$ |
| CG | ✓ | $0.4519 \pm 0.0220$ | $-1.2631 \pm 0.2168$ | $32.770 \pm 0.0029$ | $0.6063 \pm 0.0252$ | $1.1958 \pm 0.0454$ |
| AP | | $0.0320 \pm 0.0005$ | $1.9051 \pm 0.1403$ | $29.090 \pm 0.0075$ | $0.0215 \pm 0.0005$ | $0.0598 \pm 0.0027$ |
| AP | ✓ | $0.0292 \pm 0.0007$ | $2.3509 \pm 0.0654$ | $29.058 \pm 0.0064$ | $0.0104 \pm 0.0002$ | $0.0409 \pm 0.0014$ |
| SGD | | $0.0449 \pm 0.0003$ | $0.9635 \pm 0.2071$ | $23.646 \pm 0.0135$ | $0.0409 \pm 0.0005$ | $0.1645 \pm 0.0085$ |
| SGD | ✓ | $0.0334 \pm 0.0002$ | $1.6021 \pm 0.0652$ | $23.445 \pm 0.1611$ | $0.0321 \pm 0.0006$ | $0.1230 \pm 0.0081$ |

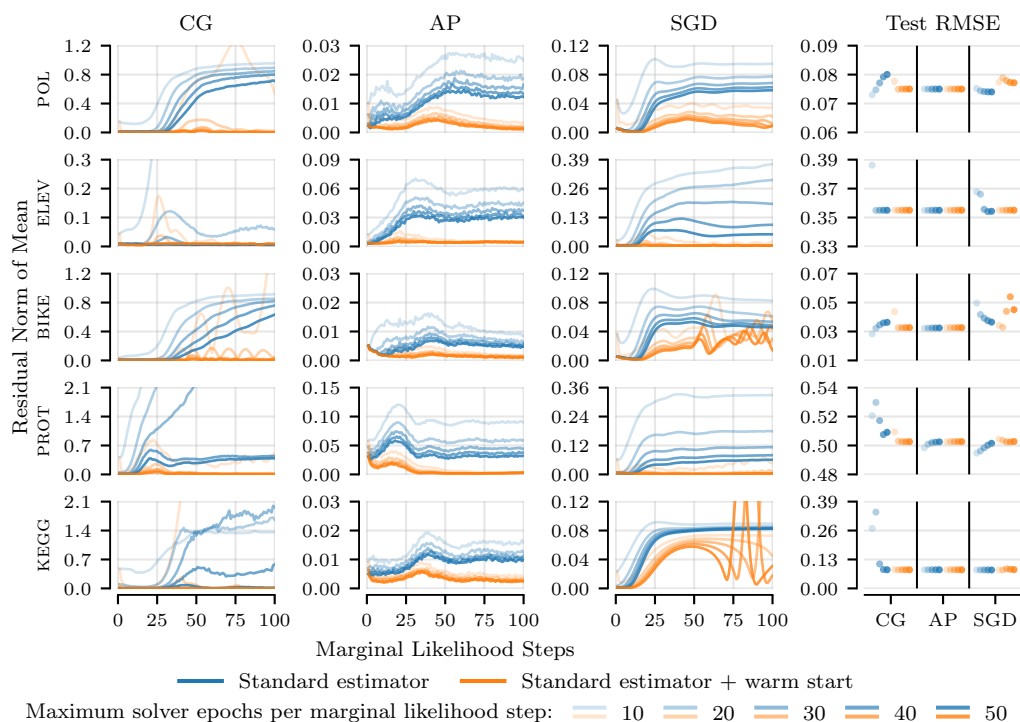

Figure 14: Relative residual norms of the mean at each marginal likelihood step and final test root-mean-square errors using the standard estimator on different datasets. The linear system solver is terminated upon either reaching the tolerance or exhausting a maximum number of solver epochs.

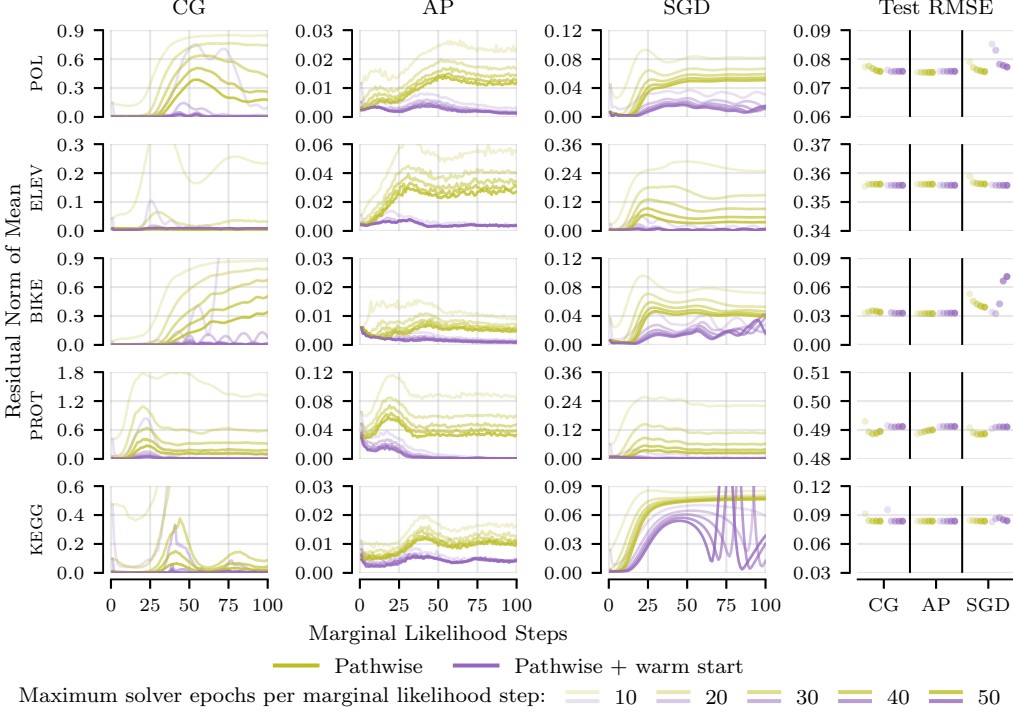

Figure 15: Relative residual norms of the mean at each marginal likelihood step and final test root-mean-square errors using the pathwise estimator on different datasets. The linear system solver is terminated upon either reaching the tolerance or exhausting a maximum number of solver epochs.

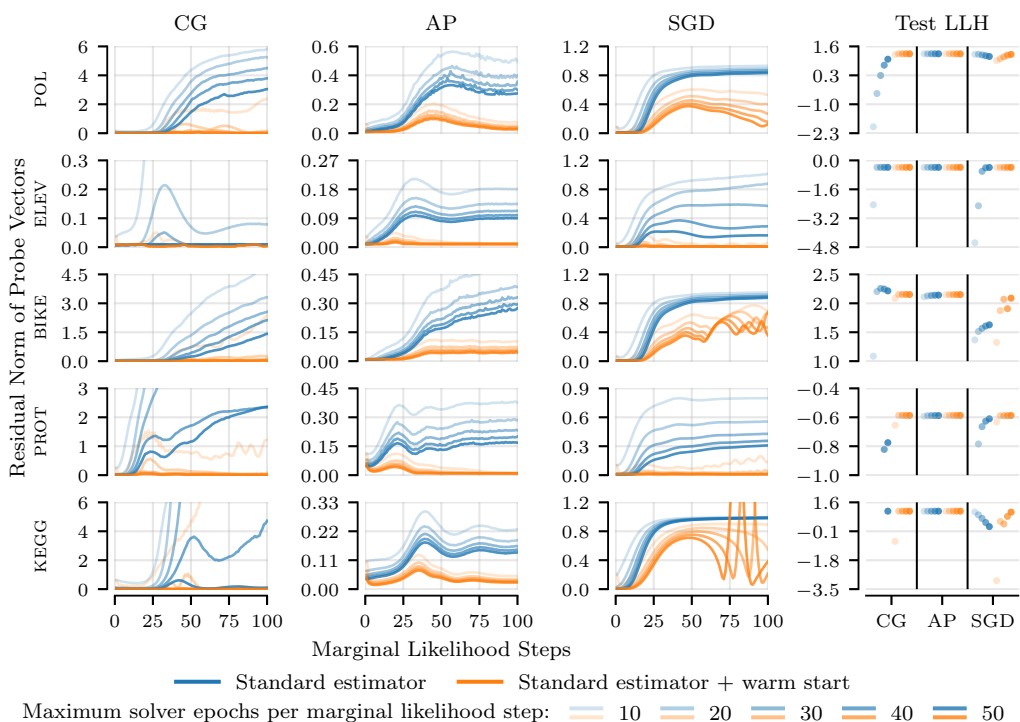

Figure 16: Relative residual norms of the probe vectors at each marginal likelihood step and final test log-likelihoods using the standard estimator on different datasets. The linear system solver is terminated upon either reaching the tolerance or exhausting a maximum number of solver epochs.

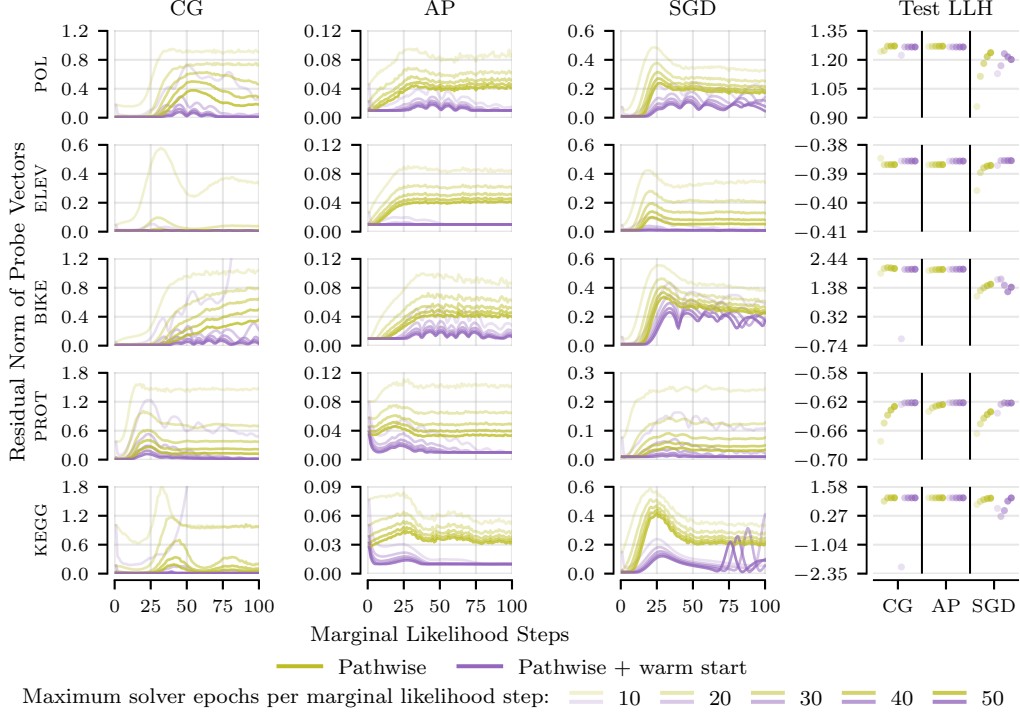

Figure 17: Relative residual norms of the probe vectors at each marginal likelihood step and final test log-likelihoods using the pathwise estimator on different datasets. The linear system solver is terminated upon either reaching the tolerance or exhausting a maximum number of solver epochs.

Conjugate Gradients

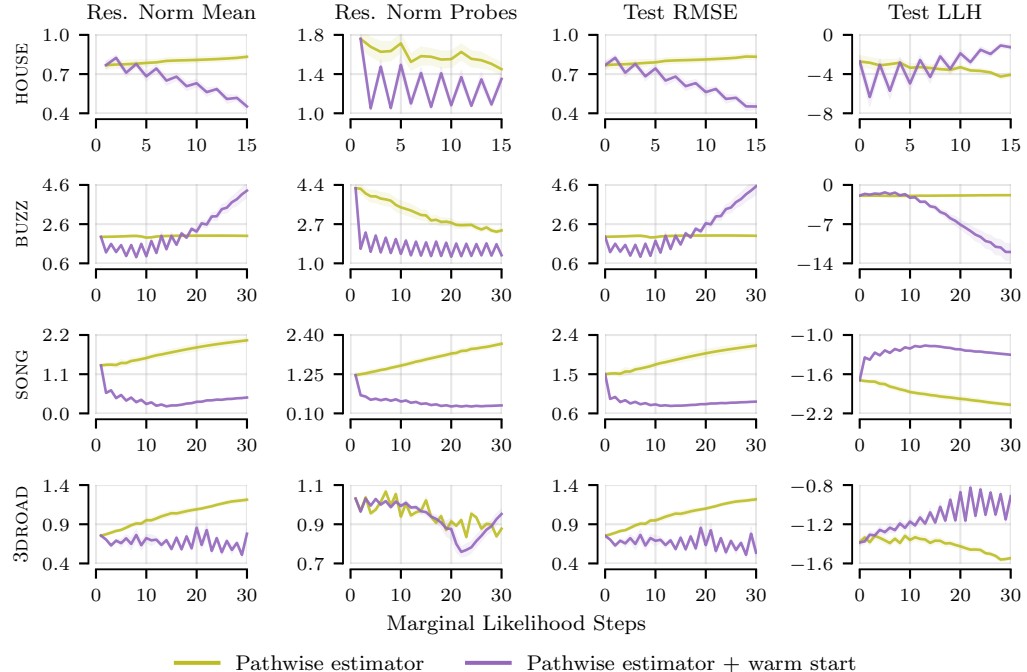

Figure 18: Relative residual norms, test root-mean-square errors and test log-likelihoods during marginal likelihood optimisation on large datasets using the pathwise gradient estimator and conjugate gradients as linear system solver.

Alternating Projections

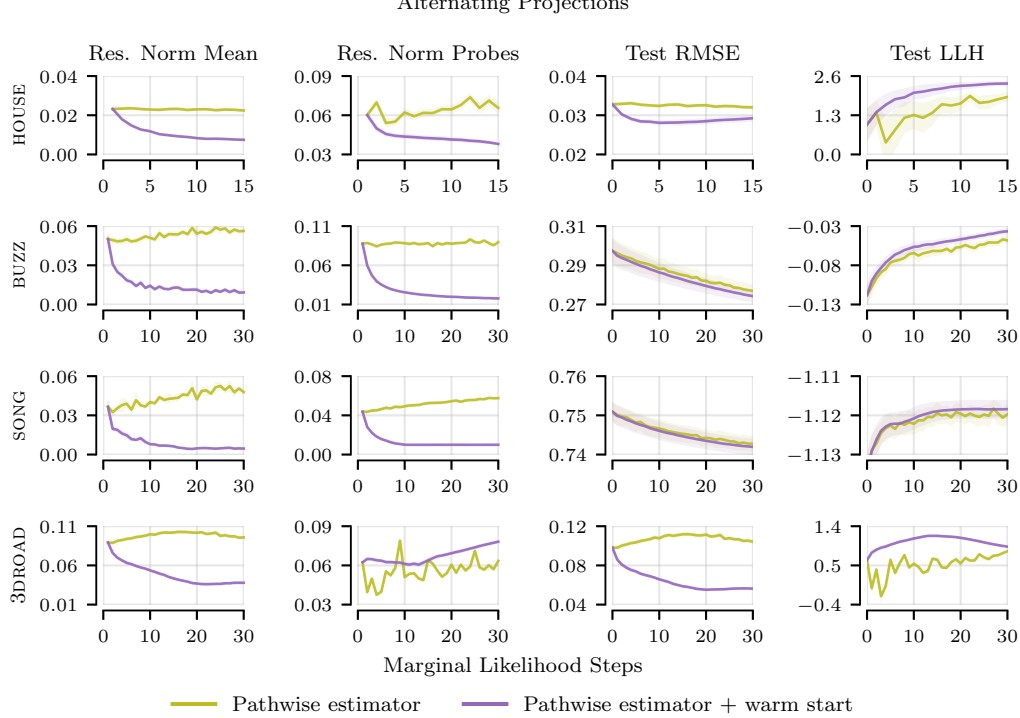

Figure 19: Relative residual norms, test root-mean-square errors and test log-likelihoods during marginal likelihood optimisation on large datasets using the pathwise gradient estimator and alternating projections as linear system solver.

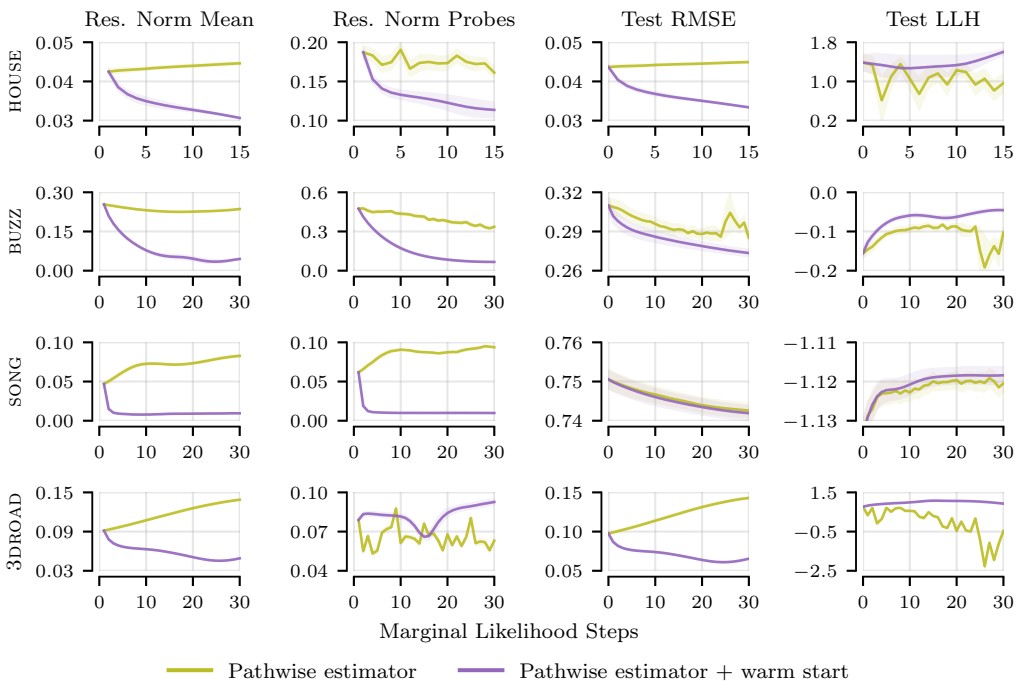

Figure 20: Relative residual norms, test root-mean-square errors and test log-likelihoods during marginal likelihood optimisation on large datasets using the pathwise gradient estimator and stochastic gradient descent as linear system solver.

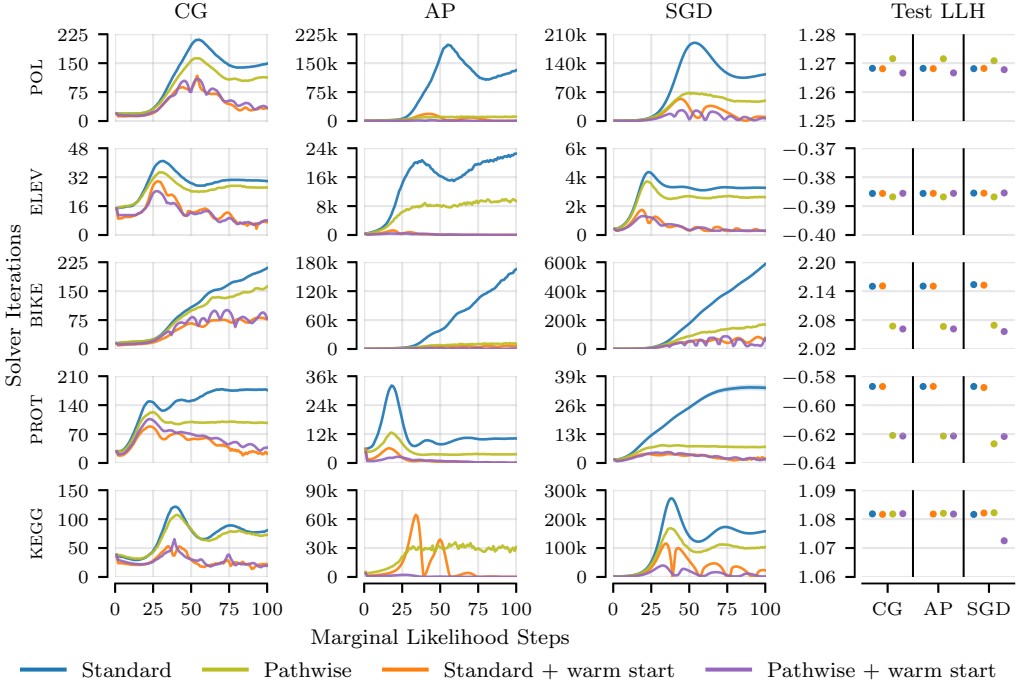

Figure 21: Required number of solver iterations until reaching the tolerance $\tau = 0.01$ at each step of marginal likelihood optimisation and final predictive test log-likelihoods on different datasets. On the KEGGDIRECTED dataset, alternating projections with the standard estimator and without warm starting did not complete the experiment within 24 hours.

