# OpenReview forum: "Improving Linear System Solvers for Hyperparameter Optimisation in Iterative Gaussian Processes"
_NeurIPS.cc/2024/Conference — NeurIPS 2024 poster_

### Official Review · Reviewer_upE1 · 2024-07-11

**Soundness:** 3
**Presentation:** 4
**Contribution:** 3
**Rating:** 6
**Confidence:** 4

**Summary:**

This paper focuses on solving linear systems arising from large-scale Gaussian process hyperparameter optimisation.

The first technique proposed in this paper is a reformulation of the linear system for log determinant gradient estimation, which is called the "pathwise estimator".
The authors argue that this reformulation makes the all-zero initialization closer to the solution.
This reformulation also allows them to reuse the linear system solutions in the prediction time by pathwise conditioning (Wilson et al., 2021).

The second technique proposed in this paper is warm starting the linear solvers.
Though, warm starts require using a set of **fixed** probe vectors, which necessarily introduces biases to the hyperparameter optimization.
The authors demonstrate empirically that the bias does not hurt in practice.

**Strengths:**

- The writing is very clear.
- By combining several techniques, this paper achieves large speed-up factors over a wide range of datasets.
Besides the speed-up, it appears that the pathwise estimator with warm start also stabilizes the linear systems solvers in some cases.
- The empirical evaluations are extensive with datasets ranging from 10 thousand data points to 1.8 million data points.
The extensive experiments with recent scalable GP methods could provide tips and tricks to practitioners.

**Weaknesses:**

The main weakness is that the techniques are not super novel per se.

There exists something similar to the pathwise estimator proposed in this paper.
For example, see Equation (7) in Gardner et al. (2018), where the probe vectors are sampled from $\mathcal{N}(\mathbf{0}, \mathbf{P})$ with $\mathbf{P} \approx \mathbf{H}_{\mathbf{\theta}}$.
Though, the original motivation in Gardner et al. (2018) is different from this paper.
Nevertheless, they do appear to be similar at least on a conceptual level.

In addition, warming starting the linear solvers has been used in practice by Artemev et al. (2021) and Antoran et al. (2023).

However, the main contribution of this paper is combining all techniques together and show their effectiveness on a wide range of datasets.
Future work could benefit from the the insights and empirical evaluations in this paper.

Minor Weaknesses:
- Theorem 1 is not very interesting.
This theorem is based on the fact that the gradient bias goes to zero as the number of probe vectors goes to infinity.
My impression is that the authors wrote this theorem to make the paper looks more "theoretical".
Frankly, I think removing this theorem makes the paper look cleaner.
A middle ground would be just presenting how fast the bias goes to zero as $s \to \infty$.

- The name "pathwise estimator" might be somewhat confusing to some readers.
This name is usually reserved for the reparameterization trick gradient, which is evidently not the case for this paper.
I suggest the authors think about alternative names for this estimator.
An possible option is to simply rephrase Section 3 as "a reformulation of the linear systems".

- Line 172: footnote should be after the period.

**Questions:**

- The conjugate gradient method estimates both the log marginal likelihood (by tridiagonalization) and its gradient.
I am wondering if using the pathwise estimator for conjugate gradient produces an estimate for the log marginal likelihood.
This is useful monitoring the progress of hyperparameter optimization.

---

> ### Author Rebuttal · Authors · 2024-08-06
>
> We thank Reviewer upE1 for their time to read and review our work, and are delighted to hear that our "writing is very clear", and "empirical evaluations are extensive". In the following, we want to address their specific concerns and questions:
>
> ---
> *There exists something similar to the pathwise estimator proposed in this paper. For example, see Equation (7) in Gardner et al. (2018), where the probe vectors are sampled from $\mathcal{N}(0, P)$ with $P \approx H$. Though, the original motivation in Gardner et al. (2018) is different from this paper. [...]*
>
> We agree that the probe vectors used by Gardner et al. (2018) have **conceptual similarities but the motivation is different**. In particular, Gardner et al. (2018) use $z \sim \mathcal{N}(0, P)$ probe vectors, where $P$ is constructed using a low-rank pivoted Cholesky decomposition, **to implement the preconditioner**, allowing their Lanczos quadrature and trace estimation to return the correct values. In contrast, our pathwise probe vectors are sampled using random features, and we used the pivoted Cholesky preconditioner *in addition* to the pathwise probe vectors.
>
> Furthermore, **we leverage the solution of the pathwise probe vector systems to construct posterior samples via pathwise conditioning** [1], which has not been done by Gardner et al. (2018). We do this by adapting the standard trace estimator to share linear solves with pathwise conditioning, effectively amortising expensive computations. In contrast, Gardner at al. (2018) solve additional linear systems to obtain the posterior predictive distribution.
>
> ---
> *In addition, warming starting the linear solvers has been used in practice by Artemev et al. (2021) and Antoran et al. (2023).*
>
> Indeed, and **we state exactly this in our footnote on page 6**. However, we also point out in our footnote that **they considered slightly different settings**.
>
> In particular, **Artemev et al. (2021) only apply warm starts to the mean system and not to any probe vectors**, because they do not perform stochastic trace estimation. Arguably, applying warm starts to the probe vectors is non-trivial due to the introduced correlations and bias, as discussed in Section 4.
>
> **Antoran et al. (2023) used warm starts for linear solvers in the context of finite-dimensional linear models with Laplace approximation rather than Gaussian processes.**
>
> ---
> *Theorem 1 is not very interesting. This theorem is based on the fact that the gradient bias goes to zero as the number of probe vectors goes to infinity. [...] Frankly, I think removing this theorem makes the paper look cleaner. A middle ground would be just presenting how fast the bias goes to zero as $s \to \infty$.*
>
> To avoid misconceptions about Theorem 1, we would like to point out that **it is not as simple as it may seem**. In particular, to obtain the result, **it is *not* sufficient to show that the error of the gradient estimator goes to zero as the number of probe vectors goes to infinity** (which would indeed not be very interesting), since the rate of convergence of the error could technically be different in different parts of the optimisation landscape, leading to asymptotically biased optima. Therefore, the actual result was quite non-trivial to obtain. The complete proof in Appendix A gives a non-asymptotic result with a convergence rate. To avoid excessive notation, we only provided a simplified asymptotic statement in Section 4.
>
> We are currently working on a simplified proof with an improved convergence rate. **In general, we believe that this is a non-trivial result and would be inclined to keep Theorem 1 in the main paper. However, we are open to keeping it in Appendix A if Reviewer upE1 feels strongly about this.**
>
> ---
> *The name "pathwise estimator" might be somewhat confusing to some readers. This name is usually reserved for the reparameterization trick gradient, which is evidently not the case for this paper. [...]*
>
> This is an interesting observation! In fact, **our pathwise estimator inherits its name from pathwise conditioning** [1], **which can be interpreted as reparameterising a sample from the GP posterior as a deterministic transformation of a sample from the GP prior** (and thus its name)! To make the connection to the reparameterisation trick more evident, note that the sample from the GP prior itself can again be reparameterised as an affine transformation of a standard normal random variable.
>
> **We will update our manuscript to include a brief version of the explanation above.**
>
> ---
> *The conjugate gradient method estimates both the log marginal likelihood (by tridiagonalization) and its gradient. I am wondering if using the pathwise estimator for conjugate gradient produces an estimate for the log marginal likelihood. This is useful monitoring the progress of hyperparameter optimization.*
>
> **Currently, the pathwise estimator does not estimate the marginal likelihood** (although this could be a future research endeavor!). However, it produces **posterior predictive samples via pathwise conditioning** [1], which can be used to **monitor the progress of hyperparameter optimisation by evaluating the predictive performance**. Conventionally, evaluating predictions is expensive because it requires additional linear solves. This becomes intractable for existing iterative methods without relying on additional approximations.
>
> ---
> *Line 172: footnote should be after the period.*
>
> **We thank Reviewer upE1 for pointing this out and will correct this for the camera-ready version.**
>
> ---
> Finally, we hope that we successfully addressed all concerns and questions, and kindly encourage Reviewer upE1 to consider increasing their score or to reach out again with follow-up questions if there are any unresolved concerns.
>
> ---
> [1] Wilson et al. (2021), "Pathwise Conditioning of Gaussian Processes", *Journal of Machine Learning Research*.

---

> > ### Comment · Reviewer_upE1 · 2024-08-09
> >
> > Thanks for the response.
> >
> > I think the paper will be in good shape provided that the authors add the promised discussions on
> > 1. The pathwise estimator and its connection with the reparameterization trick,
> > 2. The similarity and difference compared to existing work using probe vectors $\mathbf{z} \sim \mathcal{N}(\mathbf{0}, \mathbf{P})$.
> >
> > I will maintain my score for now. I think the paper presents some solid contributions, though I do understand the concerns raised by other reviewers regarding the novelty. The review scores are quite divided, and it seems that addressing the concerns in the negative reviews is in order.

---

> ### Author Response · Authors · 2024-08-13
>
> Dear Reviewer upE1,
>
> Thank you for acknowledging our rebuttal. We will make sure to include those discussions in the updated version of our manuscript.
>
> Thank you again for your time!

---

### Official Review · Reviewer_89S9 · 2024-07-13

**Soundness:** 2
**Presentation:** 2
**Contribution:** 2
**Rating:** 5
**Confidence:** 2

**Summary:**

The paper considers the problem of simultaneously fitting a Gaussian process (GP) to data along with determining the hyperparameters (kernel width, noise variance) for the GP.  The overall algorithm is standard, consisting of an outer loop, which is a simple gradient update to the hyperparameters, and an inner loop that fits the GP to the data.  Fitting the data of course requires a system solve with the kernel matrix.  Computing the gradient (or an approximation thereof) for the hyperparamters requires multiple system solves.

The paper gives some numerical experiements and some rough reasoning to argue that tweaking the trace estimator used for the gradient estimate and warm starting the linear solver saves a tangible amount of computation in this process.

**Strengths:**

Hyperparameter optimization for fitting GPs is an important and prevalent problem in machine learning.

The numerical experiments are thorough and appear to be carefully done, and lend good supporting evidence to the efficacy of the method.

**Weaknesses:**

It is hard for me to see what the really new ideas are in this paper.  The the authors say, the most critical part of this whole process is the linear system solver.  They do not actually propose any new type of solver, but rather show that by modifying the way the gradient of the hyperparameters is estimated and using a warm start, the linear solver they happen to be using will take fewer iterations.  The idea of warm starting a linear solver using previous solutions is pervasive throughout numerical analysis, so it is no surprise that it gives you some gains.  The main novelty in the paper seems to be changing the trace estimator; that is, using equation (11) in place of (7).

Something which is not really discussed, but seems important, is that the z in (7) are easier to generate than the $\xi$ in (11).  The z can be random vectors with independent entries, the $\xi$ need a prescribed covaraince structure that changes at every iteration.  Since this covariance matrix (the kernel matrix) is large, taking a random draw of the $\xi$ seems nontrivial.  How this works, and why it does not have a significant effect on the number of computations needed, needs to be spelled out more clearly.

The numerical experiements are on interesting problems, but the paper would be stronger if the mathematical arguments were more precise.

**Questions:**

How are the $\xi$ in (11) generated and what is the computational cost relative to the system solves?

---

> ### Author Rebuttal · Authors · 2024-08-06
>
> We thank Reviewer 89S9 for their time to read and review our work, and are excited to hear that our experiments are "thorough" and "carefully done". In the following, we want to address their specific concerns and questions:
>
> ---
> *It is hard for me to see what the really new ideas are in this paper. [...] They do not actually propose any new type of solver [...]. The idea of warm starting a linear solver using previous solutions is pervasive throughout numerical analysis [...]. The main novelty in the paper seems to be changing the trace estimator [...].*
>
> Indeed, **we do not propose a new solver, nor do we claim to**; the title of our paper is “Improving Linear System Solvers […]”.
>
> We want to emphasise that a major (if not our main) contribution consists of an **extensive empirical evaluation** which
> - quantifies the effectiveness of using **warm starts** and the **pathwise estimator**;
> - across **three different kinds of linear system solvers**;
> - for five small ( < 50k) datasets in the regime of **running solvers until convergence**;
> - and four large ( up to 1.8M) datasets in the regime of **limited compute budgets** of 10, 20, 30, 40, and 50 epochs.
>
> As a result, we demonstrate **speed-ups of up to 72x** when running solvers until reaching the tolerance, and achieve **up to 7x lower average relative residual norms** under a limited compute budget. We believe that this evaluation will be of **great value to future researchers and practitioners**, and **should not be underrated** when judging the novelty and contributions of our work.
>
> While we agree that warm starts are commonly used, we also believe that **warm starts have not been considered for stochastic trace estimation in the context of marginal likelihood optimisation for GPs, nor compared across various solvers before** (if yes, we would appreciate a reference to the literature). Thus, we do not claim warm starts per se as a novel contribution, but rather our application and theoretical / empirical analysis in this particular setting.
>
> ---
> *Something which is not really discussed, but seems important, is that the $z$ in (7) are easier to generate than the $\xi$ in (11). The $z$ can be random vectors with independent entries, the $\xi$ need a prescribed covaraince structure that changes at every iteration. Since this covariance matrix (the kernel matrix) is large, taking a random draw of the $\xi$ seems nontrivial. [...] How are the $\xi$ in (11) generated and what is the computational cost relative to the system solves?*
>
> We agree that the $z$ are easier to generate than the $\xi$, because the latter effectively requires a matrix square root of the kernel matrix $K$ of size $n \times n$. To draw $\xi$ efficiently, we use random features, which produce a low-rank square root with an unbiased approximation to the kernel, reducing the asymptotic time complexity from $\mathcal{O}(n^3)$ to $\mathcal{O}(nm)$, where $m$ is the number of random features. Since $m$ is constant and much smaller than $n$ (e.g. we used $m$ = 2k while $n$ > 1M), this effectively becomes an $\mathcal{O}(n)$ operation, which is faster than CG with $\mathcal{O}(n^2)$ per iteration, and similar to AP / SGD with $\mathcal{O}(n)$ per iteration. Existing work used random features for this purpose and demonstrated strong empirical performance despite this approximation [1, 2]. A discussion of alternatives can be found in Section 4 of [1].
>
> **We mention random features in Section 2, lines 70-71, and discuss them further in Section 3, lines 143-149, and Figure 5. Additionally, a detailed description of how to sample random probe vectors is provided in Appendix B, lines 486-501.**
>
> ---
> *The numerical experiements are on interesting problems, but the paper would be stronger if the mathematical arguments were more precise.*
>
> We thank Reviewer 89S9 for describing the problems in our experiments as "interesting". To clarify any potential misconceptions about our mathematical arguments, we summarise them here:
>
> In Section 3, we calculate the expected squared distance between zero initialisation and the solution of the linear solver. In particular, we show that, for the standard estimator, this distance depends on the spectrum of the regularised kernel matrix, while it is constant for the pathwise estimator. Thus, for the standard estimator, this distance tends to increase as the spectrum changes during hyperparameter optimisation, leading to substantially increased number of required solver iterations. In contrast, for the pathwise estimator, the constant distance translates to a roughly constant (and significantly lower) number of required solver iterations.
>
> In Section 4, we investigate the concern that warm starts require deterministic probe vectors which introduce correlations into subsequent gradient estimates. Despite individual estimates being unbiased, these correlations introduce bias into the optimisation. Theorem 1 shows that, under certain assumptions, the marginal likelihood after optimisation with deterministic probe vectors converges in probability to the marginal likelihood of a true maximum as the number of probe vectors goes to infinity. The proof is provided in Appendix A, and we are working on an updated, simpler proof with better constants.
>
> **After the clarifications above, we are happy to provide further details upon request and would appreciate it if Reviewer 89S9 has specific suggestions for further mathematical arguments which they think might strengthen the paper.**
>
> ---
> Finally, we hope that we successfully addressed all concerns and questions, and kindly encourage Reviewer 89S9 to increase their score or to reach out again with follow-up questions if there are any unresolved concerns.
>
> ---
> [1] Wilson et al. (2021), "Pathwise Conditioning of Gaussian Processes", *Journal of Machine Learning Research*.
>
> [2] Lin et al. (2023), "Sampling from Gaussian Process Posteriors using Stochastic Gradient Descent", *Neural Information Processing Systems*.

---

> > ### Author Response · Authors · 2024-08-10
> >
> > We thank you again for your effort put into reviewing our paper. Since there are only a few working days left for the discussion period, we would like to ask if our response resolved your concerns. If there are any remaining questions or concerns, we are happy to discuss them here. Otherwise we kindly invite you to raise your score.

---

> ### Author Response · Authors · 2024-08-13
>
> Dear Reviewer 89S9,
>
> Given that the discussion phase is soon coming to an end , this is a gentle reminder to react to our rebuttal before the deadline.
>
> Thank you!

---

> > ### Comment · Reviewer_89S9 · 2024-08-13
> >
> > Thank you for your response and detailed defense of your work.  Given the re-iteration of the magnitude of the performance gain in the numerical results, I am increasing the overall score.

---

### Official Review · Reviewer_Xgts · 2024-07-13

**Soundness:** 3
**Presentation:** 3
**Contribution:** 2
**Rating:** 7
**Confidence:** 4

**Summary:**

The paper presents approaches that can speed up solving linear systems arising in the GP regression problem. The two basic ideas are a warm start and limiting the computation budget. Another novelty is the pathwise gradient estimator, which leads to fewer iterations needed for convergence. The detailed numerical experiments confirm the performance of the suggested approaches and raise questions on the connection between the accurate solving of linear systems and the performance of the GP regression method.

**Strengths:**

1. The novel approach for gradient estimation is proposed and demonstrates significant speed-up.
2. Different techniques to accelerate solvers for linear systems are proposed, and their efficiency is confirmed via extensive experimental evaluation. This evaluation includes target solvers CG, AP, and SGD.
3. The manuscript is well-prepared, the motivation and suggestions are clear, and the results are well-explained.
4. The presented techniques can be easily incorporated into the existing packages for GP regression and used to solve practically important problems.

**Weaknesses:**

In this paper, the authors use some heuristics to speed up linear solvers and suggest a new gradient estimator based on the transformation of the normal distribution. Although the suggested tricks demonstrate improvement, their theoretical interpretation and the possibility of obtaining even better results remain unclear. So, a more detailed discussion of further research directions related to the presented approaches would be interesting. The theoretical bounds for the potential gain obtained from linear solvers while preserving test error can make the work more solid and fundamental.

**Questions:**

1. The authors mentioned in the Appendix that they use precondition based on the Cholesky factorization. Please provide more details since, typically, such preconditioners, their structure, and properties significantly affect the performance of CG.
2. Please add bold labeling the top timing and test log-likelihood values in Table 1 to simplify parsing such a large number of values
3. Why are large datasets (lines 219-235) not included in Table 1 or a similar table where the runtime gain can be observed? Please add these results, too. The large datasets can illustrate the gain from the introduced approach even better.

**Limitations:**

The authors do not provide explicit limitations of the suggested approach.

---

> ### Author Rebuttal · Authors · 2024-08-06
>
> We thank Reviewer Xgts for their time to read and review our work, and are grateful to hear that our "motivation and suggestions are clear", our manuscript is "well-prepared", and our experimental evaluation is "extensive". In the following, we want to address their specific concerns and questions:
>
> ---
> *So, a more detailed discussion of further research directions related to the presented approaches would be interesting.*
>
> At the end of Section 6, we mention that an interesting direction for future research would be to investigate why low relative residual norms do not always translate to good predictive performance. To elaborate on that, we believe that interesting research questions are:
> - Which metric, norm, or quantity actually corresponds to predictive performance (other than predictive performance itself)?
> - Can we develop a stopping criterion based on this particular quantity (instead of relative residual norm) to optimise for predictive performance?
>
> **We are happy to include this discussion at the end of Section 6 in the camera-ready version of our manuscript.**
>
> ---
> *The authors mentioned in the Appendix that they use precondition based on the Cholesky factorization. Please provide more details since, typically, such preconditioners, their structure, and properties significantly affect the performance of CG.*
>
> **We will update our manuscript to provide the following details (and references) about the pivoted Cholesky preconditioner in Appendix B:**
>
> The pivoted Cholesky factorisation [1] constructs a low-rank approximation to the full (but intractable) Cholesky factorisation by permuting rows and columns to prioritise the entries with the highest variance in a greedy way. Given the extensive and successful use of this preconditioner in the context of iterative Gaussian processes [2, 3, 4], we follow and refer to existing work.
>
> ---
> *Please add bold labeling the top timing and test log-likelihood values in Table 1 to simplify parsing such a large number of values.*
>
> **We will update our manuscript to include a summary of the following explanation in the caption of Table 1:**
>
> In Table 1, we compare different iterative methods when solving until the tolerance is reached. This results in all methods performing (nearly) identical in terms of log-likelihood (except for a few cases where the bias of the pathwise estimator due to random features leads to a slightly different outcome). Therefore, we did not bold these log-likelihood values because almost every value in the table would be bold. While the runtimes indeed differ, the comparison mainly evaluates the effects of warm starts and the pathwise estimator *per solver*. Thus, we bold the relative speed-ups per solver instead of the absolute runtimes.
>
> ---
> *Why are large datasets (lines 219-235) not included in Table 1 or a similar table where the runtime gain can be observed? Please add these results, too. The large datasets can illustrate the gain from the introduced approach even better.*
>
> **We do provide runtimes for large datasets in Tables 7-10 in Appendix C, and we are happy to include these in main paper for the camera-ready version.**
>
> Table 1 considers small datasets and presents the predictive performances of runtimes of solving until reaching the tolerance. In this setting, the final performance is basically fixed and gains in runtime are the main focus. In contrast, for the large datasets, solvers run for a fixed maximum number of epochs, implying a (roughly) fixed maximum amount of time. While in some cases solvers eventually reach the tolerance in fewer epochs and less time, this is not the main emphasis of these experiments. Instead, the main goal is to show that warm starts and the pathwise estimator lead to better performance under a fixed compute budget due to accumulation of solver progress, which is visualised in Figure 10.
>
> ---
> Finally, we hope that we successfully addressed all concerns and questions, and kindly encourage Reviewer Xgts to consider increasing their score or to reach out again with follow-up questions if there are any unresolved concerns.
>
> ---
> [1] Harbrecht et al. (2012), "On the low-rank approximation by the pivoted Cholesky decomposition", *Applied Numerical Mathematics*.
>
> [2] Gardner et al. (2018), "GPyTorch: Blackbox Matrix-Matrix Gaussian Process Inference with GPU Acceleration", *Neural Information Processing Systems*.
>
> [3] Wang et al. (2019), "Exact Gaussian Processes on a Million Data Points", *Neural Information Processing Systems*.
>
> [4] Wu et al. (2024), "Large-Scale Gaussian Processes via Alternating Projection", *Artificial Intelligence and Statistics*.

---

> > ### Comment · Reviewer_Xgts · 2024-08-09
> >
> > Dear authors,
> >
> > Thanks for the detailed response to my questions! However, a theoretical analysis of the introduced combination technique is still missing. Such theoretical analysis is important to understand why, for example, pathwise + warm start is the best combination for AP and SGD but not for CG. Please comment on what approaches can help to model the effect of the selected combinations of the subroutines. This analysis may improve your work from purely empirical to more theoretical-based and introduce more novelty.
> > Currently, I keep my score the same.

---

> > > ### Author Response · Authors · 2024-08-13
> > >
> > > Dear Reviewer Xgts,
> > >
> > > thank you for responding to our rebuttal. While we did not have enough time to compose a rigorous mathematical proof about the effectiveness of warm starts + pathwise estimator, here is some intuition about their efficacy for different solvers:
> > >
> > > Both warm starts and the pathwise estimator reduce the number of required solver iterations until reaching the tolerance by reducing the distance which the solver has to travel in the quadratic optimisation landscape.
> > >
> > > Since CG performs an exact line search during each iteration, the actual distance to the solution is less impactful than the direction of descent (for example, starting arbitrarily far away from the solution but choosing an eigenvector as direction of descent still converges in one iteration). Nonetheless, the distance to the solution still matters, because each CG iteration monotonically decreases the energy norm of the residual, such that a smaller initial residual in general translates to a smaller number of iterations until convergence.
> > >
> > > For SGD, arguably, the initial distance is more important due to the constant step size / learning rate. A "bad" initial direction of descent can be compensated via accumulated momentum. For AP, the effects of warm starts are most profound, and we believe this is the case because AP chooses the descent subspace (the next "block") based on the corresponding residual. Therefore, if a warm start initialisation is "generally close" to the solution and only "far away in certain dimensions", the residual-based selection of blocks will converge quickly.
> > >
> > > Interestingly, if the problem is ill-conditioned enough, such as in the case of continued optimisation on the BIKE dataset (as requested by Reviewer NcMH), eventually warm starts + pathwise also becomes the best combination for CG, because (i) the standard estimator becomes too sensitive to the low noise (as explained in lines 116-127 and illustrated Figure 3), whereas the pathwise estimator is (more) robust, and (ii) warm starts seem to be generally helpful. Empirically, we observe the following behaviour:
> > >
> > > | Adam iterations |  | 100 | 200 | 300 | 400 | 500 | 600 | 700 | 800 | 900 | 1000 |
> > > |:---:|---|:---:|:---:|:---:|:---:|:---:|:---:|:---:|:---:|:---:|:---:|
> > > | Solver iterations | CG + ws | 75 | 159 | 296 | 558 | 944 | 1401 | 2032 | 2861 | 3954 | 5251 |
> > > | until tolerance | CG + ws + pw | 80 | 100 | 148 | 185 | 189 | 214 | 221 | 376 | 389 | 491 |
> > >
> > > Thank you again for your time!

---

### Official Review · Reviewer_NcMH · 2024-07-21

**Soundness:** 3
**Presentation:** 3
**Contribution:** 2
**Rating:** 3
**Confidence:** 4

**Summary:**

This paper investigates several iterative techniques for solving linear systems when applied to the problem of finding GP hyperparameters. Specifically, the following modifications to the method are suggested:
- A "pathwise" sampling estimator for the Hutchinson trace estimator
- Warm starting linear systems solvers

After this, a thorough investigation is made into the effect of these choices on training GP hyperparameters.

**Strengths:**

The method follows an existing schema for training GP hyperparameters, and performs an in-depth investigation into how different choices affect the algorithm.

**Weaknesses:**

The main weakness of this paper is that it does not compare the results of the approximate methods to a reliable exact method. This makes it impossible to determine from the paper whether the predictive performances are actually good.

For datasets of these sizes (<50k), it is feasible on modern hardware to run a Cholesky on the full dataset. This should be run with BFGS optimisation. Reporting these results will give a ground-truth comparison for how good the resulted test metrics are.

If these are close, then readers will know how much to value these speedups. If they are distant, readers will be able to determine how much they value a faster approximation that does not get close to optimal predictor.

Without a comparison to any other method, the value of the overall method cannot be established. I'm happy to discuss this and significantly increase my score if this is addressed sufficiently.

**Questions:**

- Is there any information in the paper that gives a comparison to another method, or ideally a ground truth evaluation of an exact GP?

**Limitations:**

- No evaluations to any other methods, other than methods with iterative solvers.

---

> ### Author Rebuttal · Authors · 2024-08-06
>
> We thank Reviewer NcMH for their time to read and review our work. In the following, we want to address their specific concerns and questions:
>
> ---
> *The main weakness of this paper is that it does not compare the results of the approximate methods to a reliable exact method. This makes it impossible to determine from the paper whether the predictive performances are actually good.*
>
> We thank Reviewer NcMH for mentioning BFGS. Indeed, this is the preferred algorithm for GP hyperparameter optimisation when its use is computationally tractable. We initially did not consider it due to our focus on the large-scale setting. **We conducted additional experiments using the Cholesky factorisation + BFGS optimiser on the POL, ELEVATORS, and BIKE datasets**. We were not able to run BFGS on larger datasets because our A100 80GB GPUs did not support the necessary memory requirements. In particular, we used jaxopt [1], and performed optimisation for a maximum of 100 iterations, a maximum of 30 linesearch steps per iteration, and used a stopping tolerance of 1e-5.
>
> Additionally, **we provide results using the Adam optimiser + Cholesky factorisation** (instead of stochastic gradient estimation) with all other settings being identical to the iterative methods in our paper. This can be considered a pure comparison of the quality of estimated gradients. For both BFGS and Adam, all hyperparameters were initialised at the same values as for the iterative methods in our paper (namely 1.0) and also used the softplus transformation to enforce positive value constraints.
>
> Furthermore, **we also include results for SVGP** [2], a popular variational approximation, taken from Lin et al. [3], who used the same datasets and train / test splits, and performed optimisation until convergence using the Adam optimiser and 3000 inducing points.
>
> The table below lists the mean over 10 splits of test root-mean-square-errors (RMSE) and test log-likelihoods (LLH), with "Exact" referring to Cholesky factorisation + backpropagation. In summary, BFGS achieves marginally better results on the POL and ELEVATORS datasets and significantly better results on the BIKE dataset. However, we conclude that the discrepancy on the BIKE dataset is due to Adam not converging in 100 iterations, because Adam with Cholesky factorisation + backpropagation achieves nearly identical performance compared to the iterative methods (which also use Adam). Therefore, the performance gap cannot be linked to the gradient estimation via iterative methods, but rather to the different optimisers. However, in the large-scale setting, memory requirements and stochastic gradient estimation make BFGS infeasible. Therefore, we conducted all our experiments with Adam. In comparison, SVGP generally yields worse results, particularly in terms of log-likelihood.
>
> |  |  |  | RMSE |  |  |  |  |  | LLH |  |  |
> |---|:---:|:---:|:---:|:---:|:---:|:---:|:---:|:---:|:---:|:---:|:---:|
> |  | POL | ELEV | BIKE | PROT | KEGG |  | POL | ELEV | BIKE | PROT | KEGG |
> | CG | 0.0750 | 0.3550 | 0.0326 | 0.5024 | 0.0837 |  | 1.2682 | -0.3856 | 2.1500 | -0.5871 | 1.0818 |
> | CG + ws | 0.0754 | 0.3562 | 0.0326 | 0.4909 | 0.0837 |  | 1.2716 | -0.3868 | 2.0674 | -0.6210 | 1.0818 |
> | CG + pw | 0.0750 | 0.3550 | 0.0327 | 0.5026 | 0.0837 |  | 1.2681 | -0.3856 | 2.1508 | -0.5871 | 1.0816 |
> | CG + ws + pw | 0.0758 | 0.3558 | 0.0329 | 0.4912 | 0.0836 |  | 1.2666 | -0.3856 | 2.0615 | -0.6214 | 1.0819 |
> |  |  |  |  |  |  |  |  |  |  |  |  |
> | AP | 0.0750 | 0.3550 | 0.0326 | 0.5024 | - |  | 1.2682 | -0.3856 | 2.1504 | -0.5871 | - |
> | AP + ws | 0.0754 | 0.3562 | 0.0325 | 0.4907 | 0.0837 |  | 1.2715 | -0.3868 | 2.0668 | -0.6214 | 1.0820 |
> | AP + pw | 0.0750 | 0.3550 | 0.0326 | 0.5027 | 0.0837 |  | 1.2681 | -0.3856 | 2.1503 | -0.5871 | 1.0818 |
> | AP + ws + pw | 0.0758 | 0.3558 | 0.0330 | 0.4912 | 0.0836 |  | 1.2666 | -0.3856 | 2.0616 | -0.6213 | 1.0817 |
> |  |  |  |  |  |  |  |  |  |  |  |  |
> | SGD | 0.0750 | 0.3550 | 0.0326 | 0.5026 | 0.0837 |  | 1.2681 | -0.3855 | 2.1535 | -0.5871 | 1.0816 |
> | SGD + ws | 0.0754 | 0.3562 | 0.0324 | 0.4894 | 0.0837 |  | 1.2708 | -0.3868 | 2.0692 | -0.6268 | 1.0822 |
> | SGD + pw | 0.0750 | 0.3550 | 0.0327 | 0.5027 | 0.0837 |  | 1.2682 | -0.3855 | 2.1524 | -0.5878 | 1.0821 |
> | SGD + ws + pw | 0.0757 | 0.3558 | 0.0332 | 0.4911 | 0.0839 |  | 1.2678 | -0.3854 | 2.0562 | -0.6217 | 1.0725 |
> |  |  |  |  |  |  |  |  |  |  |  |  |
> | Exact (BFGS) | 0.0714 | 0.3495 | 0.0317 | - | - |  | 1.2765 | -0.3693 | 3.2685 | - | - |
> | Exact (Adam) | 0.0750 | 0.3550 | 0.0326 | 0.5025 | 0.0837 |  | 1.2683 | -0.3856 | 2.1501 | -0.5868 | 1.0813 |
> | SVGP | 0.10 | 0.37 | 0.08 | 0.57 | 0.10 |  | 0.67 | -0.43 | 1.21 | -0.85 | 0.54 |
>
> ---
> *Is there any information in the paper that gives a comparison to another method, or ideally a ground truth evaluation of an exact GP?*
>
> Currently, **Figures 5 and 8** visualise the influence of the pathwise estimator and warm starts by comparing their optimisation trajectories to exact optimisation using Cholesky factorisation + backpropagation (and otherwise identical configurations).
>
> Additionally, **Figures 11-13 in Appendix C** also compare iterative methods to exact optimisation with Cholesky factorisation + backpropagation by visualising the optimisation trajectories of selected hyperparameters. Furthermore, we are happy to include the quantitative results from the table above in the camera-ready version.
>
> ---
> Finally, we hope that we successfully addressed all concerns and questions, and kindly encourage Reviewer NcMH to increase their score or to reach out again with follow-up questions if there are any unresolved concerns.
>
> ---
> [1] Blondel et al. (2021), "Efficient and Modular Implicit Differentiation", *arXiv*.
>
> [2] Hensman et al. (2013), "Gaussian Processes for Big Data", *Uncertainty in Artificial Intelligence*.
>
> [3] Lin et al. (2024), "Stochastic Gradient Descent for Gaussian Processes Done Right", *International Conference on Learning Representations*.

---

> > ### Comment · Reviewer_NcMH · 2024-08-07
> >
> > Yes, the focus on truly large-scale datasets would make running a full Cholesky impossible. It is also fine to run these experiments on small datasets, where you can make the comparison, to give an indication of how your proposed method works across many datasets. (Provided you can discuss how the tuning of the method in the small-scale experiments is related to how the method can be tuned in the large-scale experiments where it cannot be compared to a ground truth.)
> >
> > It seems wrong that memory is a problem on Prot and Kegg. BFGS stores a Hessian approximation of size H x H, where H are the number of parameters to be optimized. Given the information in lines 58-62, Kegg should have H = 29, resulting in a minimal influence on memory?
> >
> > Your own results show that 100 iterations is not enough for some datasets (bike) to converge. Should you not then optimise for longer, since this is a trivial way to improve performance? In the neural networks community, comparing to a network that isn't trained for long enough isn't considered a proper baseline. The comparison to Adam is very helpful here. However, if bfgs can't be used with your proposed method, and this is a way to get better performance, this certainly would be a disadvantage of your method that should be discussed clearly.
> >
> > I had a look at some of the papers you cite that benchmark similar datasets, and it indeed seems like the rmse of bike can be improved by ~10x (see your reference [3]). This is a huge difference that is very impactful in places where GPs are used, e.g. in finance applications where a tiny improvement in signal from noise can be very valuable.
> >
> > In looking into this, I noticed that [3] raises an additional point about bike this paper does not investigate: The (CG) solver based method became unstable in the low noise setting, which is exactly what the dataset needs to perform well. This seems related to restricting the number of optimisation steps to 100, to avoid this behaviour from appearing.
> >
> > I'm happy to hear your thoughts on this. Overall, my current thinking is that this paper is empirical in nature (as is acknowledged by other reviewers). However, this makes good experiments crucial, and currently it seems that there are limitations in the baselines, and limitations in the method that are clear from other papers, but not properly discussed here.

---

> > > ### Author Response · Authors · 2024-08-10
> > >
> > > *It seems wrong that memory is a problem on Prot and Kegg. BFGS stores a Hessian approximation of size H x H, where H are the number of parameters to be optimized. Given the information in lines 58-62, Kegg should have H = 29, resulting in a minimal influence on memory?*
> > >
> > > After careful investigation, we realised that our implementation based on jaxopt [1] was calculating the objective value and its gradient at the same time, using more memory than necessary in order to save computations. In particular, the high memory consumption was not due to the Hessian approximation of BFGS, as you pointed out correctly. Instead, simultaneous computation of the objective value and its gradient effectively instantiated multiple matrices of size $n \times n$, where $n$ is the amount of training data, leading to exhaustive memory consumption on the PROT and KEGG datasets.
> > >
> > > **We made the change to calculate the objective value and its gradient sequentially, thus saving memory** (at the cost of introducing more computations), and added the **new BFGS results for the PROT and KEGG datasets** to the table below (while replicating some results from the previous post to ease comparison). We conclude that, on the PROT and KEGG datasets, **BFGS achieves negligible improvements compared to Adam** in terms of test root-mean-square-error (RMSE) and test log-likelihood (LLH), similar to the POL and ELEV datasets (we discuss BIKE below).
> > >
> > > |  |  |  | RMSE |  |  |  |  |  | LLH |  |  |
> > > |:---:|:---:|:---:|:---:|:---:|:---:|:---:|:---:|:---:|:---:|:---:|:---:|
> > > |  | POL | ELEV | BIKE | PROT | KEGG |  | POL | ELEV | BIKE | PROT | KEGG |
> > > | Exact (BFGS) | 0.0714 | 0.3495 | 0.0317 | 0.5023 | 0.0832 |  | 1.2765 | -0.3693 | 3.2685 | -0.5864 | 1.0818 |
> > > | Exact (Adam) | 0.0750 | 0.3550 | 0.0326 | 0.5025 | 0.0837 |  | 1.2683 | -0.3856 | 2.1501 | -0.5868 | 1.0813 |
> > >
> > > ---
> > > *The comparison to Adam is very helpful here. However, if bfgs can't be used with your proposed method, and this is a way to get better performance, this certainly would be a disadvantage of your method that should be discussed clearly.*
> > >
> > > We agree that BFGS cannot be used with our proposed method and acknowledge this as a limitation. However, this is (currently) a **general limitation of scalable iterative methods for GPs** and Adam is commonly used instead, for example, Gardner et al. (2018) [4], Wang et al. (2019) [5] (who perform 10 iterations of L-BFGS and 10 iterations of Adam on a subset of size 10k followed by 3 iterations of Adam on the full dataset, and compare those results to performing 100 iterations of Adam on the full dataset), and Wu et al. (2024) [6] (who perform 50 or 100 iterations of Adam on the full dataset, depending on the dataset).
> > >
> > > There are several obstructions to using BFGS in this setting: stochastic gradients, evaluation of the log-marginal likelihood itself (not just its gradient), and computationally expensive linesearch. Arguably, stochastic gradients can be turned into deterministic gradients by using fixed probe vectors, which is required for warm starts anyways. However, evaluating the log-marginal likelihood in a deterministic way, in particular, **calculating the log-determinant of the kernel matrix, is intractable in the large-scale setting without low-rank approximations or lower bounds**. Even if it were possible to evaluate the log-determinant, the iterative linesearch in each BFGS iterations might become prohibitively expensive.
> > >
> > > Furthermore, on the $n$ < 50k datasets, where Cholesky factorisation + BFGS is still feasible, **the performance between BFGS and Adam, as reported in the table above, is small and arguably negligible** (except for bike, which we will discuss below). Given that Adam only requires (stochastic) evaluations of the gradient and does not need to perform computationally expensive linesearch, and thus scales to larger datasets, we believe this is a reasonable trade-off.

---

> > > > ### Author Response · Authors · 2024-08-10
> > > >
> > > > *I had a look at some of the papers you cite that benchmark similar datasets, and it indeed seems like the rmse of bike can be improved by ~10x (see your reference [3]). This is a huge difference that is very impactful in places where GPs are used, e.g. in finance applications where a tiny improvement in signal from noise can be very valuable.*
> > > >
> > > > Lin et al. (2024) [3] report a best RMSE value of 0.04 for the BIKE dataset (using SDD or CG), which is actually slightly worse than the values which we report for our iterative methods (ranging from 0.0324 - 0.0332). **Could it be that you accidentally compared the wrong numbers or columns?** Please correct us if we misunderstood your statement.
> > > >
> > > > ---
> > > > *Your own results show that 100 iterations is not enough for some datasets (bike) to converge. Should you not then optimise for longer, since this is a trivial way to improve performance? In the neural networks community, comparing to a network that isn't trained for long enough isn't considered a proper baseline.*
> > > >
> > > > *In looking into this, I noticed that [3] raises an additional point about bike this paper does not investigate: The (CG) solver based method became unstable in the low noise setting, which is exactly what the dataset needs to perform well. This seems related to restricting the number of optimisation steps to 100, to avoid this behaviour from appearing.*
> > > >
> > > > **We performed additional experiments on the BIKE dataset using 1000 iterations of Adam** and report the test RMSE, test LLH, and noise variance $\sigma^2$ every 100 iterations in the table below. We observe that RMSE and LLH both continue to improve and do not appear to be fully converged even after 1000 iterations. After 10x the amount of iterations, RMSE values decreased by less than 15%, LLH values increased by more than 60%, while $\sigma^2$ decreases exponentially. **We conclude that the BIKE dataset is essentially noiseless, such that the test LLH can likely be increased indefinitely by reducing $\sigma^2$ until reaching numerical limits.**
> > > >
> > > > In particular, after 1000 iterations, $\sigma^2$ reaches ~$10^{-6}$ for iterative methods and ~$10^{-8}$ for exact optimisation, which is far below typical noise values used in this setting (Wu et al. (2024) [6] enforce $\sigma^2 > 10^{-4}$ and Wang et al. (2019) [5] use $\sigma^2 > 10^{-1}$ for large datasets), and close to the amount of jitter which is often added to the diagonal of a kernel matrix to ensure numerical stability when performing Cholesky factorisation.
> > > >
> > > > Notably, iterative methods (with warm starts and using the pathwise estimator) lead to larger $\sigma^2$ values than exact optimisation, which is expected because the pathwise estimator makes use of random features which introduce a small amount of noise. **Together with our conclusion above, the difference in $\sigma^2$ explains the performance gap in terms of test LLH.** This is supported by the fact that iterative methods actually achieve marginally better test RMSE than exact optimisation.
> > > >
> > > > We initially chose to perform 100 iterations of Adam to match previous work [5, 6] and because we mainly cared about providing a controlled environment, where all settings such as initial conditions, optimiser, learning rate, etc. are identical, such that the evaluation focuses on comparing the estimated gradients and the resulting optimisation trajectory.
> > > >
> > > > | Iterations |  | 100 | 200 | 300 | 400 | 500 | 600 | 700 | 800 | 900 | 1000 |
> > > > |:---:|---|:---:|:---:|:---:|:---:|:---:|:---:|:---:|:---:|:---:|:---:|
> > > > | RMSE | CG + ws + pw | 0.0329 | 0.0311 | 0.0304 | 0.0299 | 0.0293 | 0.0289 | 0.0286 | 0.0283 | 0.0281 | 0.0279 |
> > > > |  | AP + ws + pw | 0.0330 | 0.0311 | 0.0304 | 0.0299 | 0.0294 | 0.0289 | 0.0286 | 0.0283 | 0.0281 | 0.0279 |
> > > > |  | Exact (Adam) | 0.0326 | 0.0312 | 0.0314 | 0.0312 | 0.0307 | 0.0301 | 0.0295 | 0.0290 | 0.0285 | 0.0281 |
> > > > |  |  |  |  |  |  |  |  |  |  |  |  |
> > > > | LLH | CG + ws + pw | 2.0615 | 2.4447 | 2.6774 | 2.8406 | 2.9655 | 3.0676 | 3.1544 | 3.2306 | 3.2991 | 3.3594 |
> > > > |  | AP + ws + pw | 2.0616 | 2.4439 | 2.6758 | 2.8387 | 2.9634 | 3.0652 | 3.1515 | 3.2268 | 3.2948 | 3.3554 |
> > > > |  | Exact (Adam) | 2.1501 | 2.6182 | 2.9202 | 3.1397 | 3.3092 | 3.4466 | 3.5621 | 3.6617 | 3.7492 | 3.8273 |
> > > > |  |  |  |  |  |  |  |  |  |  |  |  |
> > > > | $\sigma^2$ | CG + ws + pw | 3.52e-4 | 1.00e-4 | 4.69e-5 | 2.91e-5 | 2.08e-5 | 1.59e-5 | 1.24e-5 | 9.92e-6 | 7.99e-6 | 6.52e-6 |
> > > > |  | AP + ws + pw | 3.52e-4 | 9.88e-5 | 4.67e-5 | 2.88e-5 | 2.05e-5 | 1.56e-5 | 1.23e-5 | 9.87e-6 | 7.99e-6 | 6.53e-6 |
> > > > |  | Exact (Adam) | 2.38e-4 | 4.89e-5 | 1.22e-5 | 3.10e-6 | 9.13e-7 | 3.34e-7 | 1.47e-7 | 7.44e-8 | 4.16e-8 | 2.51e-8 |

---

> > > > > ### Author Response · Authors · 2024-08-10
> > > > >
> > > > > Finally, we thank you for being a responsive and engaging reviewer, and hope that you find our additional results and discussion convincing. We are happy to include any of the results and discussion above in the camera-ready version. Please reach out to us again if there are further questions or concerns! Otherwise we kindly ask you to reconsider your score.
> > > > >
> > > > > ---
> > > > > [4] Gardner et al. (2018), "GPyTorch: Blackbox Matrix-Matrix Gaussian Process Inference with GPU Acceleration", *Neural Information Processing Systems*.
> > > > >
> > > > > [5] Wang et al. (2019), "Exact Gaussian Processes on a Million Data Points", *Neural Information Processing Systems*.
> > > > >
> > > > > [6] Wu et al. (2024), "Large-Scale Gaussian Processes via Alternating Projection", *Artificial Intelligence and Statistics*.

---

> ### Author Response · Authors · 2024-08-13
>
> Dear Reviewer NcMH,
>
> Given that the discussion phase will end very soon, this is a friendly reminder to respond to our latest update before the deadline. We remain excited to share these new results and clarifications with you, and we are confident they are likely to address your concerns!
>
> Thank you!

---

> > ### Author Response · Authors · 2024-08-14
> >
> > Dear Reviewer NcMH,
> >
> > We are reaching out to you one last time before the deadline of the author-reviewer discussion period to kindly encourage you to consider our latest update in your evaluation of our work. In particular, all other reviewers are now leaning towards acceptance of our submission, and we think that our new results are likely to address your concerns.
> >
> > Thank you!

---

> > > ### Comment · Reviewer_NcMH · 2024-08-14
> > >
> > > I was referring to reference [3] in your paper.
> > >
> > > The "controlled setting" you mention means that you only compare approximations to approximations. In deep learning, continuous progress on benchmark datasets is shown. Each paper gets a better accuracy on benchmark datasets like Imagenet, and are compared with the strongest previous methods. These experiments don't do that, and that worries me.
> > >
> > > Overall, I am happy to accept that you have made an improvement for iterative methods. I am not convinced that your experiments show where your approximation stands in relation to other approximation methods, and what the best possible performance is.
> > >
> > > It also worries me that important baselines are being done during the rebuttal phase, in a rush, with mistakes made along the way, when they should have been core to the paper. Overall, this leaves me undecided. Perhaps the paper could be accepted on the basis of a claim of improvement only compared to other iterative methods, on limited datasets.
> > >
> > > I will discuss with the other reviewers, and AC.

---

### Author Rebuttal · Authors · 2024-08-06

We thank all reviewers, ACs, SACs, PCs, organisers, and other volunteers for their time. NeurIPS 2024 would not be possible without their generous time commitments! In the following, we first give an overview of reviewer comments and then discuss the main concerns raised by reviewers.

---
**Overview**

- We are generally happy about the **large amount of positive feedback** received from the reviewers.
- In particular, we are glad to hear that our **"motivation and suggestions are clear"**, that our manuscript is **"well-prepared"**, and that our **"writing is very clear"**.
- In terms of our empirical contributions, we are excited to hear that reviewers described our **experimental evaluation** as **"extensive"**, **"thorough"**, and **"carefully done"**.
- In the following, we summarise the main concerns raised by reviewers, and juxtapose them with our perspective.

---
**No Comparison to Exact Optimisation**
- According to Reviewer NcMH: **"The main weakness of this paper is that it does not compare the results of the approximate methods to a reliable exact method."**
    - Therefore, we provide **new results for BFGS and Adam optimisers using Cholesky factorisation + backpropagation**, as requested by Reviewer NcMH.
    - **Tabulated results**, **experimental details**, and a **comprehensive discussion** are provided in the individual response to Reviewer NcMH.

---
**Novelty and Contributions**
- We want to emphasise that a major (if not our main) contribution consists of an **extensive empirical evaluation**.
    - Our evaluation quantifies the effectiveness of using **warm starts** and the **pathwise estimator**, separately and combined;
    - across **three different kinds of linear system solvers**;
    - for five small ($n$ < 50k) datasets in the regime of running solvers **until convergence**;
    - and four large ($n$ up to 1.8M) datasets in the regime of **limited compute budgets** of 10, 20, 30, 40, and 50 epochs.
    - As a result, we demonstrate **speed-ups of up to 72x** when running solvers until reaching the tolerance, and achieve **up to 7x lower average relative residual norms** under a limited compute budget.
    - We believe that this evaluation will be of **great value to future researchers and practitioners**, and **should not be underrated** when judging the novelty and contributions of our work.

- Some reviewers were **concerned about the novelty of warm starts**, because they are commonly used in the context of optimisation.
    - While we agree about their common use, we would like to highlight that, to the best of our knowledge, **warm starts have not been explored in the context of stochastic trace estimation for marginal likelihood optimisation in Gaussian processes**.
    - In this setting, warm starts **introduce correlations into otherwise unbiased gradient estimates**, leading to bias in the optimisation trajectory.
    - We provide a **theorem which supports the use of warm starts** in this setting, and demonstrate empirically that **warm starts are extremely effective in practice**.

- Reviewer upE1 pointed out that **concepts similar to our proposed pathwise estimator can be found in the existing literature**, such as Gardner et al. (2018) who sample probe vectors $z \sim \mathcal{N}(0, P)$ to implement a preconditioner $P^{-1}$.
    - While we also sample probe vectors from a similar distribution, we agree with Reviewer upE1 that our **motivation is different**.
    - In particular, we introduce a **novel connection based on pathwise conditioning**, which **amortises computations** between optimising hyperparameters and drawing posterior samples, and enables **efficient evaluation of predictive performance** during training.
    - Furthermore, we provide a **compelling theoretical argument** and **matching empirical evidence** which explain why the **standard estimator requires more solver iterations than the pathwise estimator** to reach the tolerance.

---
**Limitations**

- Reviewers mentioned that our **manuscript currently does not contain an explicit section which discusses the limitations** of our methods.
    - While this is true, **we do acknowledge limitations and discuss them** throughout the paper (as stated in the NeurIPS checklist).
    - In particular, we consider the **dependence on random features** to efficiently sample from a Gaussian process prior as the main limitation of the pathwise estimator
    - We briefly mention this dependence in **Section 2**, discuss its consequences in **Section 3** and **Figure 5**, and provide further details in **Appendix C**.
    - Additionally, we consider the **bias introduced into the optimisation trajectory** due to deterministic probe vectors required for warm starts as another limitation.
    - Theorem 1 shows that **optimal performance can theoretically still be achieved** despite this bias, and Figure 8 illustrates that this **bias is negligible in practice**.
    - We are happy to provide an **explicit discussion of these limitations in its own section** if the reviewers believe that this would be a valuable addition to the manuscript.

---

Last but not least, we again thank everyone involved for their time and contributions to making NeurIPS 2024 possible!

---

### Decision · Program_Chairs · 2024-09-25

**Decision:**

Accept (poster)

**Comment:**

I think this is a solid paper with some weaknesses that are generally well addressed overall in the rebuttal. To summarize my thoughts:

- The inclusion of exact results where possible clearly demonstrates that the authors' speed ups are "valid" in the sense that their approximate methods are achieving nearly exact results in terms of RMSE and LLH. Additionally, in my experience in general is that the strength of BFGS is often oversold compared to Adam provided care is taken with GPs, and the results presented here are roughly in line with what I've seen.

- I don't generally see comparing a method whose sole purpose is to speed up iterative methods with the iterative methods it speeds up as a weakness. Given that the authors' results are ultimately the same as the underlying iterative method, whether or not these methods perform better than e.g. SVGPs is a total red herring: that is probably the fault of the underlying iterative method, not the authors' techniques in this paper. I ultimately don't think that this paper is the correct venue to litigate the merit of iterative methods as a whole.

- On the weaknesses side (and I mention this in the possibly vain hope that the authors consider this for the camera ready deadline), I'd like to note that a trivial way around the issue of comparing to exact GPs would be to simply run on larger datasets. In particular, I'm surprised that -- given fairly large speedups to already relatively scalable methods -- we aren't running on much larger datasets here? Many of the original papers (e.g., [32]) run on much larger datasets. Why aren't we showing impressive 72x speedups on massive datasets that should now take very little time to train on?

Overall I think the paper is good and I'm strongly in favor of acceptance, but I *do* think the paper could be *great* if the authors would just run on larger datasets and solve point 3.